**Estimation of hydrological drought recovery based on precipitation and GRACE water storage deficit**

Alka Singh[1&2,*], John T. Reager[3], Ali Behrangi[4]

[1]Universities Space Research Association, Columbia, MD, 21046, USA.
5  [2]NASA Goddard Space Flight Center, Greenbelt, MD, 20771, USA.
[3]Jet Propulsion Laboratory, California Institute of Technology, Pasadena, CA, 91109, USA
[4]Department of hydrology and atmospheric sciences, The University of Arizona, Tucson, AZ, 85721 USA
*Correspondence to*: Alka Singh (alka.singh@nasa.gov; +1.301.286.6386)

**Abstract.** Drought is a natural climate extreme phenomenon that presents great challenges in forecasting and monitoring for water management purposes. Previous studies have examined the use of Gravity Recovery and Climate Experiment (GRACE) terrestrial water storage anomalies to measure the amount of water 'missing' from a drought-affected region, and other studies have attempted statistical
approaches to drought recovery forecasting based on joint probabilities of precipitation and soil moisture. The goal of this study is to combine GRACE data and with historical precipitation observations to quantify the amount of precipitation required to achieve normal storage conditions in order to estimate a likely drought recovery time.  First, linear relationships between terrestrial water storage anomaly (TWSA) and cumulative precipitation anomaly are established across a range of
conditions. Then, historical precipitation data are statistically modeled to develop simplistic precipitation forecast skill for two years based on climatology and long-term trend. Two additional three different precipitation scenarios i.e. wet and exceptionally wet are simulated to predict the recovery period by using respectively one and three a standard deviations of the normal precipitation forecast. Precipitation scenarios are convolved with precipitation water deficit estimates (from GRACE) to
calculate the best-estimate of a drought recovery period. The results show that in the regions of strong seasonal amplitude (like monsoon belt) drought continues even with the above-normal precipitation until its wet season. The h Historical GRACE-observed drought recovery period is used to validate the approach. The e Estimated drought for an example month demonstrated ~80% similar recovery period as observed by the GRACE for the region where storage - precipitation relationship is strong.

## 1  Introduction

Drought is a widespread recurring natural hazard with several direct and indirect impacts. The s Shortage of water in an ecosystem not only reduces water availability for human consumption but also causes extensive flora and fauna mortality. Dryland, with little vegetation on the surface, increases soil erosion, reduces water resilience time, and enhances the possibility of forest fires, leading to many
indirect disasters. Big historical droughts have affected millions of lives and cost billions of dollars in the last half a century. For example, the 1988 USA drought is estimated to cost $40 billion, 1999 drought in Asia affected 60 million people (Mishra and Singh, 2010). Severe water-crises can put society in turmoil and drive large-scale migrations particularly in the developing parts of the world for example the 2011 East African drought (Lyon and DeWitt, 2012) or the 2014-16 dry corridors of
central America (Guevara-Murua et al., 2018).
There are different definitions of drought depending on the context, including agricultural (soil moisture deficit), meteorological (eg. precipitation deficit or increase in evapotranspiration), and hydrological (storage deficit for eg. in streamflow/groundwater deficit) droughts (Behrangi et al., 2015; Mishra et al., 2006; Wilhite and Glantz, 1985). This study focusses on hydrological drought, which requires
combining both surface (snow and surface water), and subsurface (soil moisture and groundwater) hydrological information. To monitor and evaluate drought, several drought indices are available like the Palmer drought severity index (PDSI) (Palmer, 1965), standardized precipitation index (SPI) (McKee et al., 1993), standardized precipitation evaporation index (SPEI) (Vicente-Serrano et al., 2009), etc. They heavily rely on the accuracy of meteorological inputs, hence become unreliable where
ground observations are sparse (Zhao et al., 2017). Additonally, use of a consistent drought metrics for

various climatic regimes is essential for global drought studies. With the availability of different remote sensing observations, various global drought indicies are developed like Normalized differential vegetation index (NDVI) (Keshavarz et al., 2014), Evaporation stress index (ESI) (Otkin et al., 2013), Soil moisture index (SMI) (Sridhar et al., 2008), Soil water deficit index (SWDI) (Martínez-Fernández
et al., 2015). These drought monitoring indices are mostly based on a few hydrological parameters (like soil moisture, precipitation, ET) and have no information about the drought recovery period. Gravity Recovery and Climate Experiment (GRACE) mission enables us to measure the integrated water storage variation in a system, which includes surface water, soil moisture, and groundwater. Many studies have used GRACE to described the process and monitoring of drought (Awange et al.,
2016; Forootan et al., 2019; Sun et al., 2017; Thomas et al., 2014; Yirdaw et al., 2008; Zhang et al., 2015). Yirdaw et al. (2008) were foremost in exploring the potential of GRACE in the drought monitoring in the Canadian Prairie region. Houborg et al. (2012) developed a GRACE-based drought indicator by assimilating terrestrial water storage (TWS) into Catchment Land Surface Model (CLSM) over North America. Thomas et al. (2014), for the first time, used GRACE terrestrial water storage
anomaly (TWSA) as an independent global drought severity index by considering negative deviations from the monthly climatology of the time series as storage deficits. ~~This method can improve the characterization of drought because it provides both the total amount of missing water from an ecosystem and also clearly identifies the beginning and the end of a drought, on a monthly timescale. The ultimate benefit of this approach is that by quantifying the amount of water required in storage for a~~
~~region to return to historical average conditions, the method allows for the identification of an explicit hydrological drought recovery target. Furthermore, the GRACE-based drought index is independent of other drought indices and the have global spatial coverage.~~ While an increasing number of case studies have used GRACE to characterize drought in different regions, for example, Amazon (Chen et al., 2009; Frappart et al., 2012), Texas (Long et al., 2013), China (Zhao et al., 2018), a global gridded
assessment of the direct application of GRACE on drought are still a few (Gerdener et al., 2020; Li et al., 2019). Unlike other drought indices, the GRACE-based drought index is independent of the meteorological estimates and their combined uncertainties. The GRACE based index~~is method can improve the characterization of drought because it~~ not only provides ~~both~~ the total amount of missing water from an ecosystem but~~and~~ also clearly identifies the beginning and the end of a drought, on a
monthly timescale. The ultimate benefit of this approach is that by quantifying the amount of water required in storage for a region to return to historical average conditions, the method allows for the identification of an explicit hydrological drought recovery target.
Recovery time can be a critical metric of drought impact, in showing how long an ecosystem requires to revert to its pre-drought functional state (Schwalm et al., 2017). With the increasing frequency of
drought (Cook et al., 2014), it is essential for an ecosystem to recover completely before the next ~~successive~~ drought, otherwise repeated exposure to stress can degrade the ecosystem for a long-term. A tentative estimate of expected recovery can help water management authorities to regulate the water supply until a system recovers completely from drought stress. Previous studies have analyzed historical drought events and different predictors like teleconnections, local climate variables (temperature,
precipitation) for drought prediction (Behrangi et al., 2015; Maity et al., 2016; Otkin et al., 2015; Yuan et al., 2013)~~(Behrangi et al., 2015a; Maity et al., 2016; Otkin et al., 2015; Yuan et al., 2013)~~ but not much work has been done on drought recovery analysis. Many studies have analyzed causes and patterns of onset and termination of drought (Dettinger, 2013; Maxwell et al., 2013; Mo, 2011; Seager et al., 2019) but did not dwell into the statistical evolution of drought recovery. Hao et al., (2018)
reviewed ~~a~~ different kinds of droughts and its prediction methods based on statistical, dynamical, and hybrid methods. (Pan et al., (2013) were the first to develop a probabilistic drought recovery framework based on an ensemble forecast. They used a Copula model to establish a joint distribution between cumulative precipitation and a soil-moisture-based drought index to fine-tune their correlation structure. They demonstrated that drought recovery estimates typically have significant uncertainty and that a
probabilistic approach can offer better information on realized drought risk. Pan et al., approach is exclusively precipitation based. However, above-average rain in a given month may replenish surface water/soil moisture and support recovery in vegetation, but the true impact of drought continues until all hydrological storage compartments, including deep soil moisture and groundwater recover to their normal. In t~~T~~his study, we looked into ~~type of~~ an integrated hydrological drought ~~onset and~~ recovery

phenomenon, which can only be estimated by combining using integrated totalerrestrial water storage in all the hydrological compartmentsobservations. With the sparse availability of in-situ groundwater observations and limited soil moisture observations (up to top 5cm of the soil), a complete profile of the water stored in a column can only be obtained from the GRACE-based terrestrial water storage. The intellectual contribution of this paper is in the estimation drought recovery and conceptually bringing a framework for drought recovery forecast based on precipitation deficit.
Here we explored hydrological drought recovery time at a 0.5-degree gridded framework and focused on sub-decadal drought only because of the availability of GRACE data for 15 years. The study can be extended for a longer time frame with the GRACE- follow on observations. Building upon previous works, we apply GRACE-observed storage deficits as a drought indicator and provide different probabilistic scenarios for drought recovery based on historical precipitation analysis. Specifically, we estimate the required-precipitation to fill a storage deficit by deriving a linear relationship between precipitation and storage variability. Here, we focus on sub-decadal drought only within the GRACE period. Different precipitation scenarios are generated for precipitation inputs based on the distribution of historical observations. The required-precipitation estimates are validated by the duration of drought using the Global Precipitation Climatology Project (GPCP)GPCP and GRACE observations independently.

## 2    Data

### 2.1    GRACE

The GRACE mission operated from April 2002- June 2017 with a primary goal to track water redistribution on Earth and to improved our understanding of the global (Eicker et al., 2016; Fasullo et al., 2016) and regional water cycle (Singh et al., 2018; Springer et al., 2017). The GRACE-based TWSA includes integrated water mass changes in a vertical column which may consist of rivers, lakes, snow, ice, glaciers, soil moisture, permafrost, swamp, groundwater, etc. The GRACE level-3 solution is officially available from three different centers, which are produced using different approaches (spherical harmonics or mascon), different filers, smoothing factors, etc. and eventually, there can be discrepancies between different TWS estimates (Jing et al., 2019). The differences between GRACE solutions from different centers are mostly very small at a basin level and lie within the error bounds of the GRACE solution itself (Sakumura et al., 2014). However, at 0.5-degree grid, the difference between the amount of missing water estimated by the different GRACE solutions increases substantially (as seen in the supplementary material) because of different downscaling methods. This caveat can be mitigated in the future by using ensemble of GARCE products. Nevertheless, they are consistent with the detection of drought duration. Please look into the supplementary material for the comparison between water storage deficit estimated by the different GRACE solutions

For this study We downloaded the GRACE mascon (RL06) solutions from the Jet Propulsion Laboratory (JPL) website https://grace.jpl.nasa.gov, accessed on 03.03.2019 (Wiese et al., 2018). The gravity field signals of the GRACE are pre-processed to monthly-gridded equivalent water height (EWH) variations by JPL (Watkins et al., 2015; Wiese et al., 2016). The mascon GRACE solutions are provided at a 0.5-degree long-lat grid, but they represent the 3x3 degree equal-area caps. The sShape and size of the mascon caps vary with latitude. Therefore, the gridded mascon solutions are multiplied by a scaling factor grid (https://grace.jpl.nasa.gov/data/get-data/jpl_global_mascons/), to improve the interpretation of signals at sub-mascon resolution. Since 2011, the GRACE dataset has data gaps of 1-2 months in every 5-6 months due to the aging batteries of the satellites. However, to compare precipitation and storage variability, a continuous monthly TWSA time-series is required. Therefore, the data gaps in the time-series are filled by cubic convolution interpolation (Keys, 1981).

## 2.2    GPCP

Global Precipitation Climatology Project (GPCP) is a widely used global precipitation data. Most of the other observational products don't produce precipitation estimates beyond 60deg S/N for the longer historical period (1979 – present). Besides, GPCP applies gauge under catch correction to in situ precipitation measurement, which has been found important to improve snowfall measurement (Behrangi et al., 2018). The latest global monthly precipitation data is obtained from the ~~Global Precipitation Climatology Project~~ (GPCP V2.3~~,~~  from their website https://www.esrl.noaa.gov/psd/ (Adler et al., 2003) for 1979-2017. It is a combined satellite-based product, adjusted by the rain gauge analysis. The downloaded 2.5-degree resolution data is re-gridded to 0.5 degrees by using bilinear interpolation to harmonize it with the ~~grid resolution of the~~ GRACE grid~~solutions~~.

The spatial resolution of the original GRACE solution (3-degree mascon) and GPCP (2.5-degree) are comparable. Nevertheless, areas of the unit representations are different in tens of thousands km$^2$ at different locations which get worst towards the poles. However, the ~~as mascon~~ size and shape of mascons var~~y~~ies with latitude, therefore to improve the interpretation both datasets are brought to the 0.5-degree grid. We also acknowledge the possible caveat due to different methods of re-gridding of both the datasets, which can be improved in future work. However, as drought is a smooth process the impact of neighboring pixels should not affect the analysis significantly.

## 3    Methods

### 3.1    Storage deficit

It is useful to know the total amount of missing water from an ecosystem in order to characterize a drought so that an explicit target can be assumed that defines a drought recovery. Currently, global gridded total water storage variations can only be obtained from GRACE TWSA. The TWSA is first smoothed by three months moving average filter, followed by the removal of a linear trend to reduce the impact of long-term signals in the storage. A linear trend in storage variability can be caused by other continuous/long term processes than just precipitation, like upstream water abstraction, groundwater pumping, increase/decrease in snowmelt, etc. We acknowledge the caveat of the possibility of sudo-trend due to unusual signal at the beginning or end of the record in some regions. The reduced TWSA is termed as dTWSA. The deviation of storage (dTWSA) from its normal water storage cycle (i.e., its historical climatology) can give an idea of the severity of drought phenomena.  Here, we define 'recovery' as a return of dTWSA to the climatological storage state for a given month.  The climatology of the time series is estimated over the 15-year GRACE record (April 2002-March 2017) by averaging values from the same months of each year (i.e., all Januaries, all Februaries, so on). The negative residuals of the dTWSA from its climatology are considered as water storage 'deficit' in a grid cell (Thomas et al., 2014). If the duration of negative residuals is longer than three months, we designated it as a drought event. If recurring drought happens within  a month gap (i.e., recovery shorter than one-month duration), we considered it a continuation of the same drought. The green plot in Fig.1 shows the duration and severity of recurring drought in an example location in Australia (centered on 133.75°E 16.75°S). Using this approach, we produce a global gridded drought characteristics record, which includes the frequency, intensity~~severity~~, and duration of drought, for the 2002-2017 period. For any instance and location, the state of drought and its length can be identified by quantifying the water storage deficit from dTWSA. Eventually, recovery duration for each drought can also be observed, i.e., how long negative residuals from climatology continued. For instance, Figure-1 shows three major droughts and their respective recovery periods (of nearly 4~~1.5~~, 3~~.1~~ and 1 ~~0.5~~ years) for a sample location in Australia.

Figure 1

## 3.2 Estimation of the required-precipitation for storage deficit

$$dS/dt = P - ET - R \qquad\qquad Eq.\ 1$$

The water balance equation based on hydrological fluxes ( Eq. 1) shows that the change in terrestrial water storage (dS) in a region for a given month (dt) depends on is the monthly precipitation (P, mm/month); evapotranspiration (ET, mm/month) and the streamflow (R, which includes both surface water and subsurface water) (Swenson and Wahr, 2006). Assuming the relationship between precipitation and ET + R remains constant for a region, the variability in precipitation gives an idea of possible variation in the storage. The amount of required-precipitation to overcome a deficit can beis estimated using the association between precipitation and water storage anomaly (TWSA). Monthly GPCP observations are first reduced by their mean for the April 2002 – March 2017 period (i.e., the 15-year GRACE data record) to obtain precipitation anomaly. Then the relationship between precipitation and storage anomalies isare derived. For this, first, both variables are smoothed by a three-month moving average low pass filter to remove high-frequency noise. Then, their linear trends are removed to reduce the impact of other processes like groundwater, upstream abstraction, glacier melts, etc (as discussed above) and to focus our analysis on sub-decadal drought events within the GRACE period. The smoothed and detrended precipitation anomaly is then integrated in time to get storage anomaly, which is termed obtain as cumulative detrended smoothed precipitation anomaly (cdPA). and Finally, cdPA is compared with the smoothed and detrended storage anomaly (dTWSA).

An ecosystem may behave differently under stress (a deficit period) than under an excess-water situation. In this study, the storage (dTWSA) and precipitation (cdPA) linear relationship have been analyzed only during historical deficit periods as the system behaves differently under stress (Famiglietti et al., 1998; Vereecken et al., 2007). Several researchers used rainfall-runoff curve like soil conservation service curve number (SCS-CN) for the computation of surface runoff based on precipitation with an assumption of stable relation between rainfall and abstraction (Mishra et al., 2006; Singh et al., 2015; Verma et al., 2017). This study also assumes that a pattern of precipitation intensity for a region does not change significantly over time, consequently, the relationship between precipitation and storage variability can be considered stable.

Figure 2 shows the strength of this relationship by correlation coefficients in the top panel and linear regression coefficients in the bottom panel. In addition to storage variability, precipitation is also lost in other hydrological processes like evapotranspiration, runoff, etc. *Based on the linear relationship between dTWSA and cdPA the required precipitation has been estimated. Regression coefficients greater than 1 means the required precipitation is more than the amount of missing water. This is because precipitation lost in other hydrological processes like evapotranspiration, runoff ( Eq.1) is not observed by storage variability). Coefficient equals to 1 means the amount of required precipitation is the same as that storage loss, which means there is no other dominant process in the region. Coefficient less than 1 are the regions of weak precipitation-storage coupling, which can be due to other physical processes like melting of snow/frozen surfaces, groundwater extraction, irrigation, etc (non-red regions in Figure 2a) and hence the relationship between storage and precipitation is not strong in this region.* ThereforeFigure 2b shows that, for most of the trapical and subtropical regions, required- precipitation is more than the amount of missing water (i.e., regression coefficients greater than 1), except for the regions with frozen surfaces or weak precipitation-storage coupling (non-red regions in Figure 2a). For example, in higher latitudes, mass loss observed by GRACE during spring snowmelt is not directly linked to precipitation. Additionally, highly arid regions also have weak precipitation and storage signals and eventually their coupling is weak. Therefore, the proposed method is not suitable for regions with weak precipitation-storage coupling. These regions of the weak association are identified based on regression coefficients below 1 (Figure 2b), as less than one or negative relationship between storage variability and precipitation may describe a case in which storage variability is not linked to a direct precipitation effect. Also, locations having less than five months of drought in 15 years are excluded considered as regions of the weak association because we don't have enough drought samples to derive their association. These regions of weak association, (regression coefficients less than 1) are considered as unsuitable for the GRACE based recovery analysis and have been masked out in this study.
Figure 2

Based on the derived linear relationship between cdPA and dTWSA ~~(Figure 2, bottom plot)~~, a required-precipitation is estimated for each regional drought period. The method for the estimation of required-precipitation is shown in Figure 3 at an example location (133.75°E 16.75°S) in Australia. The top panel shows an agreement between cdPA (black plot) and dTWSA (red plot). In the bottom panel, an absolute required-precipitation (Figure 3b, blue plot) is calculated by adding precipitation climatology

to the estimated surplus required-precipitation (magenta plot), to fill the storage deficit (green plot). Analogous to an accounting methodology, this approach applies the assumption that generally more precipitation than usual (climatology) is required to replenish the losses incurred during drought. The example location has a strong annual signal (5 - 150 mm, with predominantly winter rain), which led to a relatively high ratio of required-precipitation to the amount of missing water.

Figure 3

### 3.3 Historical Precipitation analysis

Historical precipitation data from GPCP (1979 to 2017) are statistically analyzed using signal decomposition in order to create a simplistic precipitation forecast. Note that the motivation for providing a precipitation forecast here is not to present a state-of-the-art precipitation prediction, but to

265 demonstrate the potential utility of the terrestrial water storage deficit in determining required-precipitation and estimating a likely time to recovery. This methodology could be augmented with any type of more complex precipitation forecasting approach~~es~~.

#### 3.3.1 Precipitation signal decomposition

Historical precipitation data is decomposed into a linear trend, ~~as well as seasonal~~, inter-annual signal,

annual/climatological cycle, and sub-seasonal components in order to explore temporal variability. First, a linear trend and an annual signal~~l/climatology~~ (mean of each month, e.g., all January, February, etc.) ~~and a linear trend~~ are extracted from the original signal. ~~They are directly used for signal reconstruction with the assumption that a similar long-period trend will continue~~. Then, the residual signal is filtered by a 12-month low-pass window to split it into a smooth inter-annual signal and a high-frequency sub-seasonal signal. The linear trend and inter-annual signal together are considered to

contribute to long-term variability. The individual variance of the annual, long-term, and sub-seasonal signals is normalized by their sum, in order to get their fractional contribution to local variability (Figure 4). This provides an overview of the relative importance and spatial distribution of these components in global temporal variability. *Figure 4 shows the fractional variance of the decomposed*

*signal. For most regions, annual signals dominate in precipitation (Figure 4a). However, regions where the wet season is not explicit in their climatology, high-frequency signal plays a major role, for example in central Europe, eastern Siberia, western N. America, southern Australia, etc. (Figure 4c). Contrarily, the long-term signal obtained by combining linear trend and the inter-annual signal has the least variability globally (Figure 4b). The smooth inter-annual signals are driven by climate indices like*

*El Niño southern oscillation (ENSO), Pacific decadal oscillation (PDO), and the North Pacific mode (NPM), etc. (Özger et al., 2009). The linear trends can be an indicator of change in precipitation intensity or drought at the beginning or end of the analysis period (1979-2017). We acknowledge that drought at the beginning can produce some artifact in the misinterpretation of linear trend. We assume that similar linear and inter-annual trend will continue.*

~~As figure 4 shows, most regions are dominated by a seasonal cycle in precipitation.~~

Figure 4

#### 3.3.2 Signal reconstruction and forecasting skill

Based on the above findings, we formulate a statistical model for hindcasting precipitation. The ~~extracted~~ annual signal and the linear trend extracted by signal decomposition (section 3.3.1) are

295 directly used for the precipitation reconstruction, with the assumption of the continuation of the similar variability. Further, interannual variability in the precipitation data is added by autoregression for 10-14

months depending on the duration length of significant autocorrelation. Finally, the sub-seasonal signal is added, which is obtained from the residual of the inter-annual signal. This high-frequency signal has only 0-3 months of temporal autocorrelation, accordingly, we have limited skill in synthesizing sub-seasonal signals, can only be reconstructed for 0-3 months due to the lack of significant temporal autocorrelation.

Figure 5 shows the precipitation hindcast for January 2016-December 2017 at an example location (56.25°W 27.75°S) in the La-Plata basin. Figure 5a shows that the reconstructestimated precipitation (red plot) compared to its climatology (blue plot) and has a good agreement in 2017 with the GPCP observations (black plot) due to the significant contribution of interannual signal forecast for the first yearfor the same duration. However, the reconstructed signal for the second year is predominantly climatology driven with some linear treand. Figure 5b shows the synthesis of non-climatological signal signal forecastreconstructed interannual precipitation by autoregression. The figure shows that interannual autoregression (blue plot) signals have a good association with the observed interannual signal (black plot) until the first 11 months. The sub-seasonal auto autoregression is significant only for two months in the example location. The final hindcast is an integration of a linear trend, climatology, sub-seasonal, and interannual auto autoregression.

The precipitation reconstruction skill is used for a simplistic normal forecast. Further, two additional precipitation scenarios are simulated by adding respectively one and two standard deviations of precipitation to the normal forecast, which is used in probability recovery analysis.

Figure 5

### 3.3.3 Hindcast evaluation

The statistically reconstructed global precipitation time series for two years (January 2016 - December 2017) is evaluated by GPCP observations using Nash-Sutcliffe efficiency (NSE). NSE illustrates the model efficiency over the mean, i.e., if Nash- Sutcliffe coefficients are zero or less than zero, then the model is equal or worst than the observational mean respectively. Figure 6 (red region) shows that the reconstructed full signal is in good agreement with GPCP observations. In these regions, fractional variability in the climatology and long-term signal are most robust (Figure 4a & 4b). Regions dominated by the high-frequency fractional variance (Figure 4c) are not well represented in our model (the white and blue area of Figure 6).

Figure 6

## 3.4 Probabilistic recovery

Precipitation ihas the major control on drought dynamics. Knowing the amount of precipitation required to overcome a drought (at any instance and any location globally), presents the opportunity for the estimation of a likely drought recovery period. We can apply a probabilistic approach by using the historical precipitation forecast model to simulate different precipitation scenarios based on the historical distribution of precipitation for each region. Here, we propose three precipitation scenarios: 1) normal precipitation (as described in section 3.3.2), 2) one standard deviation wetter than normal precipitation is assumed as a (wet monthyear) and 3) three standard deviations wetter than normal precipitation is assumed as an (exceptionally wet monthyear). The latter two scenarios are based on a standard deviation from the local precipitation climatology, to simulate average rainy and extremely rainy months, respectively. Again, we assume that i

In order to overcome a deficit due to drought, the ecosystem needs to receive a surplus of water that surpasses the climatological average. It follows that if drier than normal conditions were to persist indefinitely, then a drought could theoretically go on forever. The climatological average is integrated with the estimated surplus required precipitation (Figure 3b, magenta plot) to obtain the absolute required precipitation (Figure 3b, blue plot). Whenever precipitation is more than the absolute required precipitation (Figure 3b, blue plot).; the system advances in recovery to its pre-drought state. Based on this hypothesis, we simulated the three scenarios for how long any instance of drought will continue, given the expected three precipitation cases. Note that the scenarios suggest the needed recovery time

for normal, wet, and exceptionally wet monthsyears, hence providing a minimum baseline for the duration of drought recovery.

## 4    Results

### 4.1    Observed recovery time based on GRACE and GPCP observation

In this study, drought is defined by the negative deviation of TWSA from its record-length climatology. The observed recovery duration is measured directly from the storage deficit, as described previously (Figure 1, Thomas et al., 2014). For our approach, we need to know when the observed precipitation is more than the absolute required--precipitation (sSection 3.2). Figure 67 shows the recovery estimation of all the droughts occurred during 2002-2017 at four random example locations: Northwest tropical
Australia (123.25°E 17.75°S), Northeast Argentina in La-Plata basin (56.25°W and 27.75°S), North India in Ganges Basin (78.75°E and 27.75°N), North Brazil in Amazon basin (57.25°W and 2.25°S). Whenever the observed precipitation (Figure 67, red plot i.e. GPCP) is larger than the required-precipitation (blue plot) for its respective month, the drought should end. Ideally, GRACE should also observe it simultaneously.
Figure 67
~~In Figure 7, observed precipitation (red dashed line) and absolute required precipitation (blue line) are shown only during drought periods (green shaded area).~~ The figure shows that the precipitation during a drought typically stays below its monthly required-precipitation until the end of the drought. In most cases, precipitation crossed the required-precipitation limit in precisely the same month when GRACE
observed the end of storage deficit. Even for the case of recurring droughts with two or more months gap, both methods observed the end of drought on approximately the same month. To examine our method in detail we randomly selected a drought month and validated our approach and estimated the recovery time based on different precipitation scenarios in the following section.

### 4.2    Example of storage deficit and required--precipitation

In this section, we discuss drought in an example month of January 2016. During the study period (2002-2017), the year 2015-2016 was the strongest El-Nino on record, and many regions experienced ~~some~~ drought. Nevertheless, it is ~~a random selection of the month~~ for the demonstration of recovery analysis and can be applied to any other time window. Figure 78 shows the regions under drought in January 2016 (Figure 78a) and the estimated required-precipitation to overcome the drought (Figure
78b). Here, the severity of a drought defined by the amount of water shortage in a month All colors other than white in the figure are drought-affected regions in January 2016, within the region of strong precipitation-storage relations (discussed in section 3.2). The color bar demonstrates the severity of the drought, i.e., the amount of missing water (top panel) and the respective amount of required-
precipitation (bottom panel). Figure 78a shows the eastern Amazon, southern Australia, south-east Africa, and north India were under severe drought in 2016 winter. For most of the region in the southern hemisphere amount of required precipitation is double the storage deficit because January is a summer month and water demand is higher.
Figure 78

### 4.2.1    Validation

To validate our approach, we compared recovery periods in Figure 89. The figure shows the recovery period from the January 2016 drought state, observed by GRACE (Figure 89a) and estimated recovery based on absolute required--precipitation and GPCP observations (Figure 89b). Figure 89c highlights the consistency in the estimated recovery period where one indicates a 1– 2 months difference, 2
indicates 3–4 months difference, 3 indicates 5–8 months difference, and 4 indicates 9+ months difference. The black ~~blue~~ area in ~~the~~ figure 8c is the region with extremely different recovery estimates, ~~which can be accounted to an error in datasets.~~ For the January 2016 drought, approximately

a) Recovery period observed by GRACE

80% of the masked global land area demonstrated a similar recovery period (+/- 1-2 months) to what was predicted (category 1 in Figure ~~8~~9c).
Figure ~~8~~9

### 4.2.2  ~~PDifferent~~ precipitation scenario~~s~~

This section demonstrates the probability of recovery duration in different precipitation scenarios. In the first section, we talk about the expected recovery percentage within a month in three different precipitation scenarios. And in the second section, we projected the duration needed to overcome the
January 2016 drought within the study period (until March 2017).

#### 4.2.2.1  The expected one-month recovery state
Spatiotemporal patterns of drought at the global scale are largely uncharacterized. Often, one-month of surplus precipitation is not enough to fill the entire deficit. However, if it rains significantly above
average immediately after/during the drought, the recovery time decreases dramatically. Therefore we compared three different surplus precipitation scenario (discussed in section 3.4) to estimate recovery percentage ~~We stimulated one-month (February 2016) recovery percentage~~ for the January 2016 drought~~, given the three different precipitation scenarios (discussed in section 3.4)~~. The surplus precipitation within a month (February 2016) is divided by the required reconstru~~estima~~ted
precipitation to calculated percentage recovery. In most of the drought-affected regions, the recovery percentage of our forecasted normal precipitation (~~s~~Section 3.3.2) for February 2016 is more than the recovery percentage of observed GPCP precipitation (Figure ~~9~~8d). This indicates, February 2016 was drier than our estimated normal.  Most of the region recovered in extremely wet scenario~~s~~ (Figure ~~9~~8c) within a month, except, regions dominated by summer monsoon (Figure ~~9~~8c, blue/cyan colored area)
with less than 30 % recovery, as February is not a rainy season for this region. This shows a case that regions with high amplitude seasonal cycles in precipitation mostly recover during their rainy season, which varies globally.
Figure ~~9~~8

#### 4.2.2.2  Best estimated time for recovery
Recovery time varies from immediate (i.e., one month) to several years across different climate zones and depending on the severity of the drought. Figure 10 shows the predicted recovery duration of the January 2016 drought state, which ranges from a month (yellow~~blue~~ color) to not recoverable within the study period of 15 months (~~dark red~~black color). Figure 10d shows the recovery duration observed by GRACE, which is considered as truth. Figure~~s~~ 10a & 10b show that most of the region under severe
drought in 2016 did not recover with even one standard deviation wetter than normal precipitation and the drought in th~~is~~ese region~~s~~ continued beyond a year. In the extremely wetter (three standard deviations) than normal situation (Figure 10c) most of the regions recovered within 4-5 months, except for regions of most severe drought, such as the South East Amazon, and Southern Africa. Even in the extremely wet scenario, the monsoon region~~s~~ (Figure 10c~~, cyan color~~) recovered only during their rainy
season (in 6-7 months from January 2016). This demonstrates that information on the state of precipitation compared to its usual can provide an idea of the expected drought recovery duration provided we know the amount of precipitation required.
Figure 10

## 5    Discussion

Here we define~~ drought intensity and duration~~drought using the observed storage deficit from GRACE TWSA, which is a 3-months or greater negative deviation from the historical, record-length climatology for each region, following Thomas et al. (2014). Generally, we considered this to be a better metric of integrated drought effects than a negative departure from climatology in precipitation or soil moisture because the former includes all components of the water cycle and represents the integrated state of the
local~~ the~~ water budget closure, dS/dt. We observe that occasionally precipitation anomalies are depressed a couple of months before GRACE sees the beginning of drought onset because the net water

mass balance can stay stable for some time by a compensating decrease in ET and runoff. Similarly, precipitation shows a positive deviation from climatology (i.e., excess precipitation) well before GRACE observes the end of the drought because of the time-lag to fill the rootzone soil moisture (Eltahir and Yeh, 1999). (Dettinger, 2013; Maxwell et al., 2013) also argued that drought onset is quicker than drought termination. Sometimes very heavy rain can quickly bring a region entirely out of a drought, but in many cases, continuous surplus precipitation is needed to bring the entire soil water column (i.e., from the surface to groundwater) to fully recover. Only GRACE can measure total variations in all of the hydrological compartments in a region.

The critical feature of the GRACE-based drought recovery framework is the estimation of required precipitation to fill a storage deficit. Figure 2 shows that TWSA is closely associated with cumulative precipitation anomaly for most regions, except in deserts and high-latitudes. In large arid regions, monthly storage variability is significantly low due to low rainfall. In high-latitudes, seasonal water storage variability is mainly driven by temperature because of snow accumulation and melt. Typically in cold regions, winter snow accumulation and spring snowmelt drive increases and declines in TWSA, decoupling the storage variability from precipitation variability, which leads to a phase shift in their seasonality and weak correlation between them (Reager and Famiglietti, 2013). For these reasons, a storage-based drought recovery metric is not as capable in desert and high-latitude areas and have been are masked out in the results section.

Variability in the historical precipitation data is analyzed by signal decomposition to develop a simple precipitation forecast model. Precipitation signals are hindcast by combining the climatology with the linear trend and an interannual signal estimated from autoregression. Figure 4 shows that in most regions seasonal variability is the strongest signal, except in big deserts, Eurasia, and northwest America. Theseese regions s have high monthly sub-seasonal variability in precipitation, which is hard to reconstruct. Additionally, due to the contribution of snowfall in higher latitudes, and very low rainfall in deserts, bias correction in precipitation data are relatively less reliable. Consequently, we have less confidence in precipitation simulations in those regions (Figure 6).

In addition to the normal precipitation forecast, two more precipitation scenarios are simulated based on one and three standard deviations from the climatology, assuming that a system recovers from drought only when the precipitation is more than the usual (climatological) precipitation of the corresponding month. Figure 910 demonstrates percentage recovery given these three different precipitation scenarios. The figure shows that most regions show significant recovery within a month in three standard deviations wetter than normal scenario, except for regions which are not in their respective rainy season. As precipitation can be scarce in non-rainy-season months, even three standard deviations wetter than the historical average precipitation would not be a substantial amount of rain to replenish the water deficit in these periods. We further investigate the recovery duration based on different precipitation scenarios (Figure 101) and find that under normal precipitation, most regions will not recover significantly within the study duration, but for three standard deviations wetter-than-normal rain, they recover within 3-4 months. However, for the regions with the strong seasonal intensity of precipitation (monsoonal region), the figure showed recovery only during its rainy season (after 6-7 months) even in the extreme wet scenario.

We validated our required precipitation estimates by comparing the recovery period observed by GRACE and estimated by our method on the GPCP observations (Figure 78) at different locations, which showed good concurrence. Also in Figure 101, the drought recovery duration for an example month of January 2016 demonstrated a good agreement between the observed recovery by GRACE and estimated recovery by GPCP for most of the masked regions (80% within +/- 1 month).

Knowing the present state of precipitation, i.e., how much surplus we have over usual climatology of a region can give an idea of expected recovery duration, provided we know the amount of precipitation needed to fill the deficit. With the improved precipitation forecasting skills, more accurate drought recovery estimates can be obtained. Nevertheless, the study demonstrates a case of application of GRACE for the estimation of required precipitation for drought recovery.

## 6    Conclusions

Increasing water-demand and future uncertainties in climate necessitate the assessment of the potential impact of drought and its expected recovery duration. The consequences of drought can be minimized through adaptation and risk management efforts, informed by the amount of missing water in a system and required-precipitation needed to bring it back to normal (as shown in figure 7). Recurring droughts due to insufficient recovery can be minimized to a large extent by managing water resources wisely particularly during the deficit period until all of the hydrological components revert to the pre-drought state. The study demonstrates the utility of GRACE terrestrial water storage anomalies (TWSA) in obtaining statistics of hydrologic drought, i.e., its recovery period and required- precipitation to recover with sensitivity test to different precipitation scenarios. The benefits of the GRACE-based drought index for drought analysis are: 1) the independency from meteorological variables unlike other drought indices (PDSI, SPEI, SPI) and 2) the spatial coverage of the GRACE data (much of the globe). However, recovery analysis is limited to the area where linear-relationships between TWSA and cumulative precipitation anomaly exhibit strong linkages. However, careful cautions are warranted to interpret the GRACE signal at 0.5 degree grid due to different post processing techniques applied by different GRACE solutions to overcome the inherent limitation in the spatial resolution of GRACE. Significant difference in the intensity of drought is observed by different GRACE solutions. Nevertheless, all GRACE solutions have same drought durations.

The findings of this study are 1) the GRACE based drought index is valid to estimate the required-precipitation for drought recovery and 2) the period of drought recovery depends on the intensity of precipitation i.e. in the dry season of the year drought continues even with above-normal precipitation. The recovery period estimated by our approach matches well with the recovery observed by GRACE for most of the masked regions (80%) for the demonstrated drought month. This approach can be extended with the availability of new GRACE follow-on (GRACE-FO) datasets, launched in May 2018. The proposed method and analyses in this study are applicable to the development of an operational drought monitoring system that can provide the actionable information for drought recovery given that the skillful precipitation prediction is available.

**Funding:** This research was funded by the GRACE science team meeting.

**Acknowledgments:** The research was carried out at the Jet Propulsion Laboratory, California Institute of Technology, under a contract with the National Aeronautics and Space Administration (80NM0018D004).

**Conflicts of Interest:** The authors declare no conflict of interest.

**Author Contributions:** For this research article Dr. Singh has contributed in the data curation, formal analysis, writing, and visualization. Dr. Reager has conceptualized, supervised, and acquired funding to conduct this study. He has also helped in writing and editing the text. Dr. Behrangi has also contributed by supervision and funding acquisition.

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

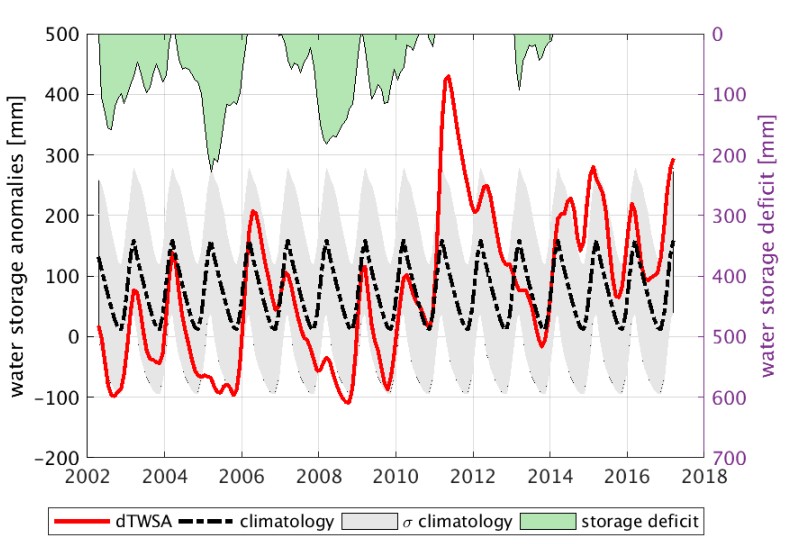

Figure 1: Water storage deficit from GRACE: The smoothed and detrended TWSA (dTWSA in red plot) is reduced by its climatology (black plot), to estimate deviation from the climatology. The negative residuals from the climatology are plotted on the upper axis as a green shaded area and scaled on the right side. The grey shade indicates ±1 standard deviation of the climatology.

a.  Correlation coefficients

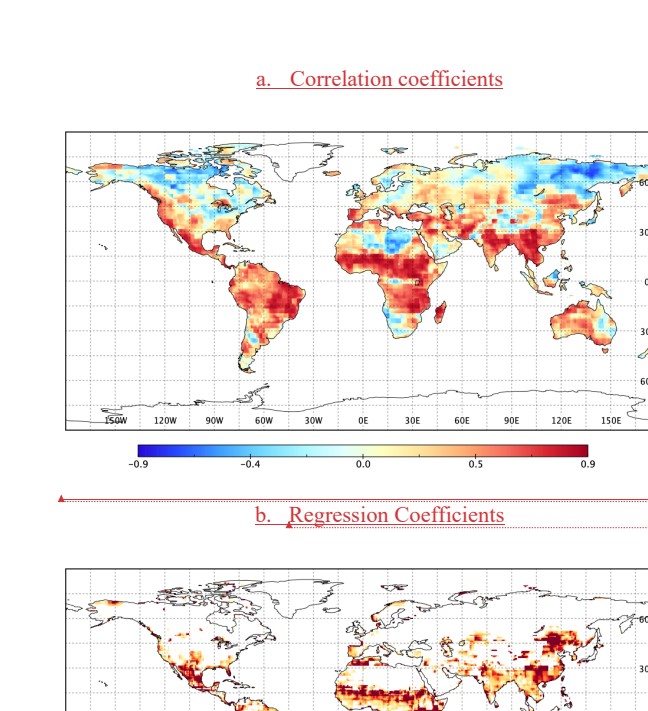

b.  Regression Coefficients

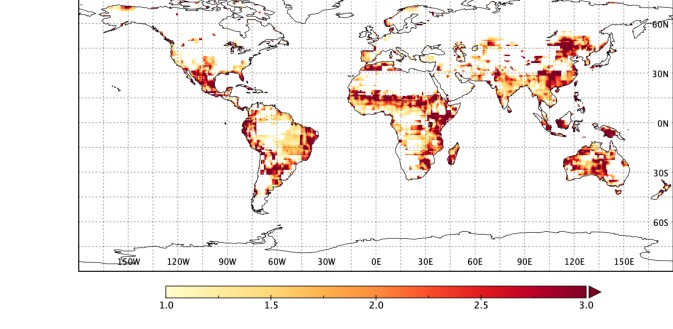

Figure 2: a) Correlation coefficients and, b) regression coefficients between cumulative detrended precipitation anomalies (cdPA) and detrended terrestrial water storage anomaly (dTWSA).

700

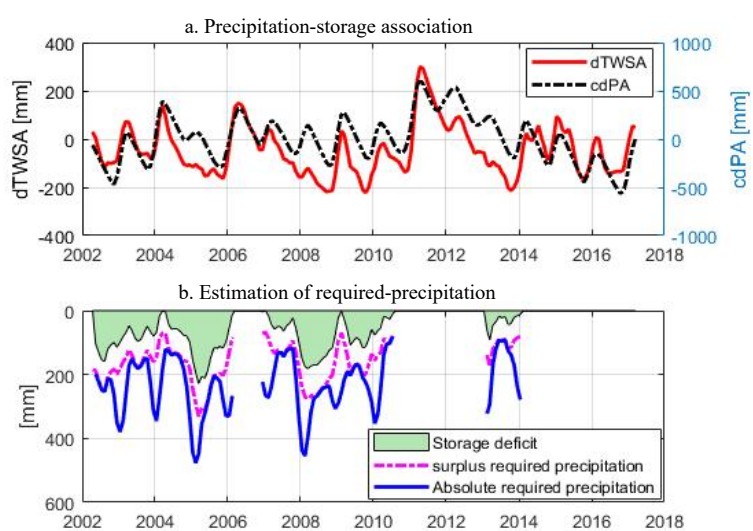

Figure 3: Estimation of the required-precipitation at an example location. a) Cumulative detrended precipitation anomaly (cdPA) compared with the detrended storage anomaly (dTWSA). b) Surplus required-precipitation is estimated (magenta plot) from the linear relationship between dTWSA and cdPA, to fill the storage deficit (green plot). Then precipitation climatology is added to obtain absolute required-precipitation (blue plot).

a.  Annual Signal

b.  Linear trend + inter-annual signal

c.  Sub-seasonal signal

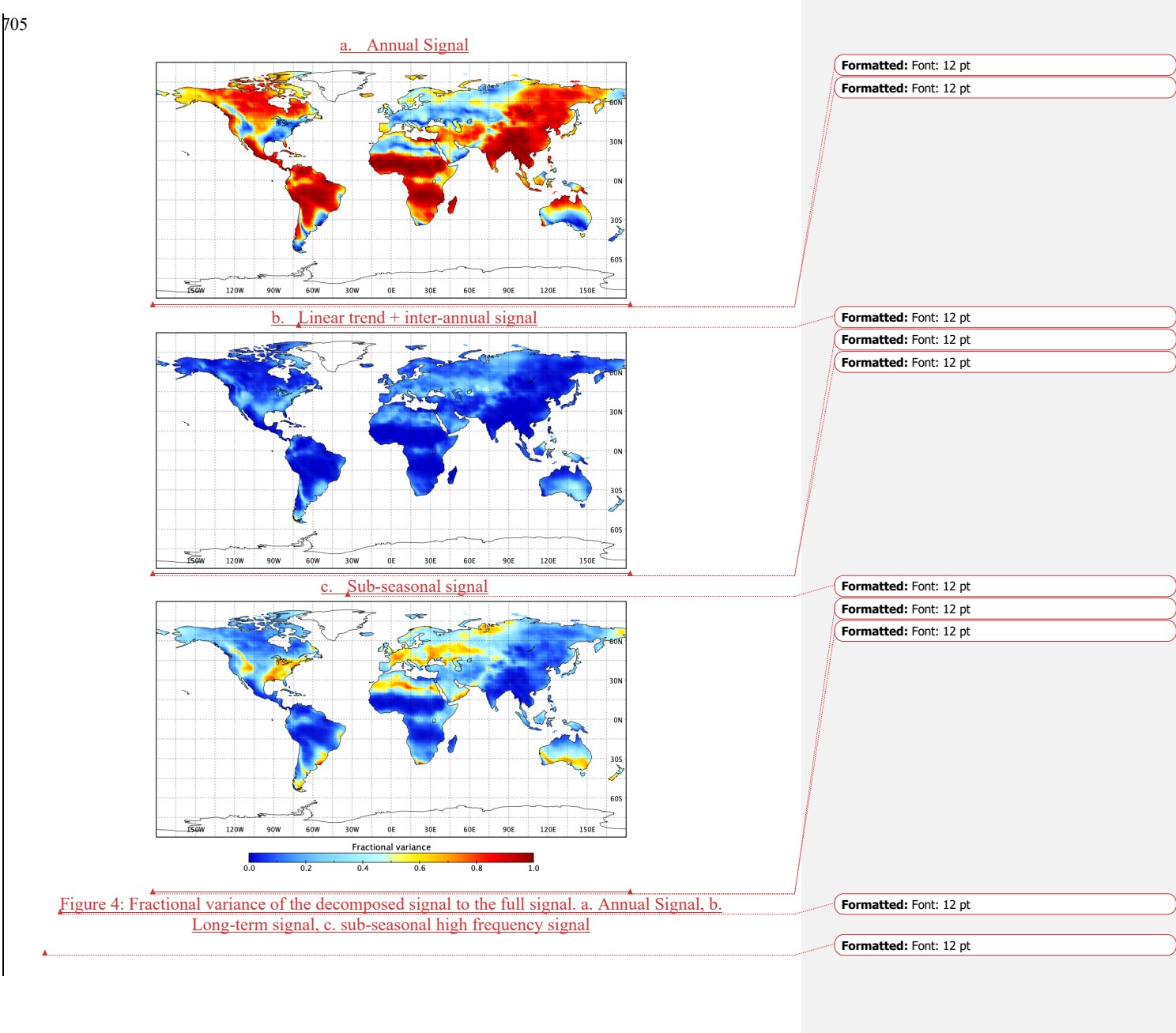

Figure 4: Fractional variance of the decomposed signal to the full signal. a. Annual Signal, b. Long-term signal, c. sub-seasonal high frequency signal

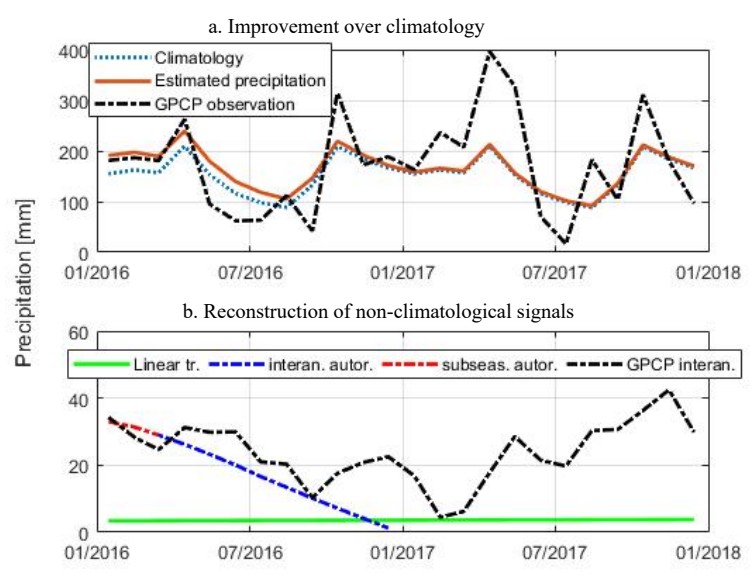

Figure 5: Reconstruction of precipitation signal for 2016-2017. a) The reconstructed signal compared with GPCP observations and its climatology. b) The reconstruction of a long-term secular signal from the linear trend, and inter-annual and sub-seasonal autoregression, compared to GPCP interannual signal.

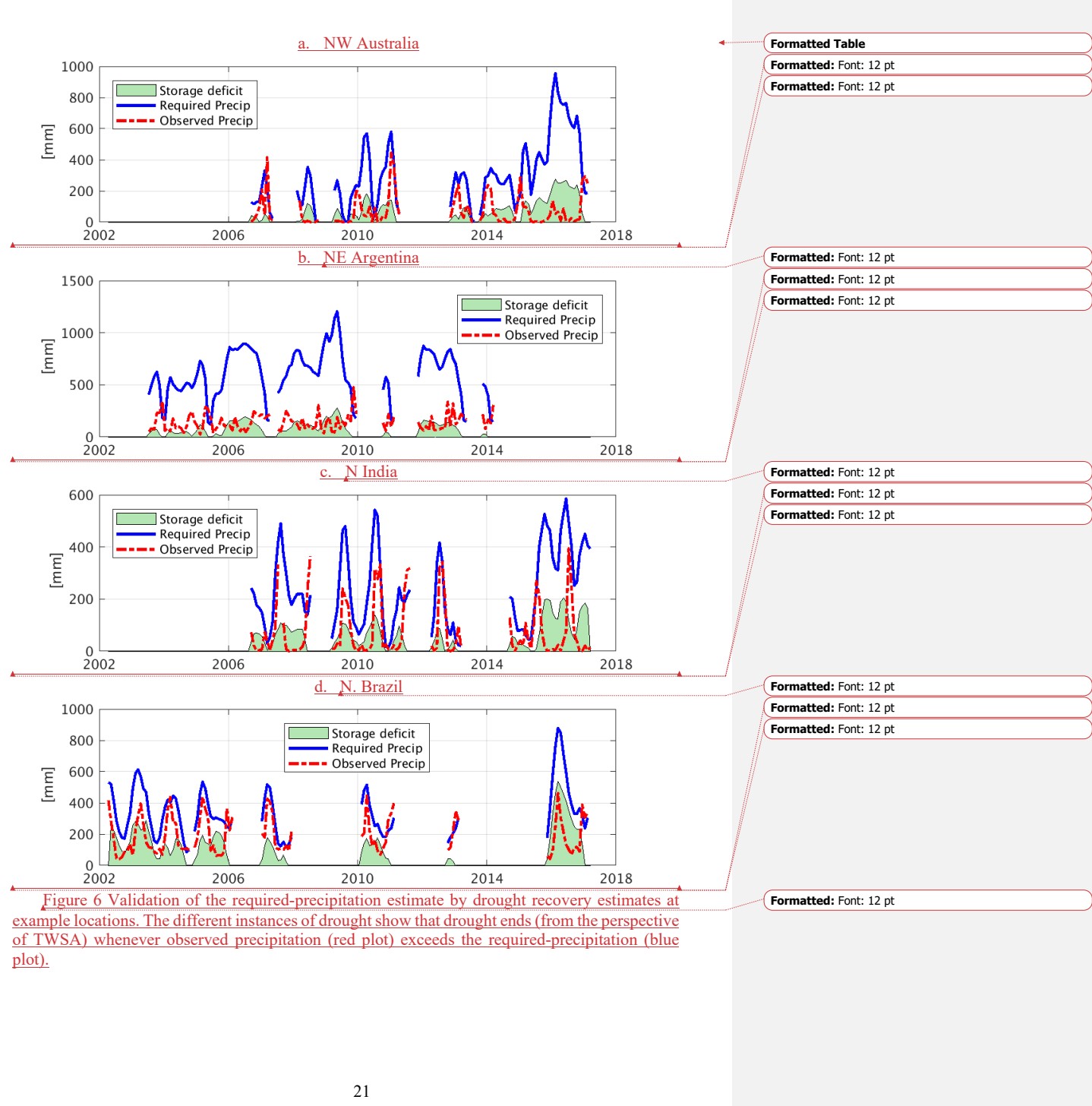

Figure 6 Validation of the required-precipitation estimate by drought recovery estimates at example locations. The different instances of drought show that drought ends (from the perspective of TWSA) whenever observed precipitation (red plot) exceeds the required-precipitation (blue plot).

a. Storage deficit

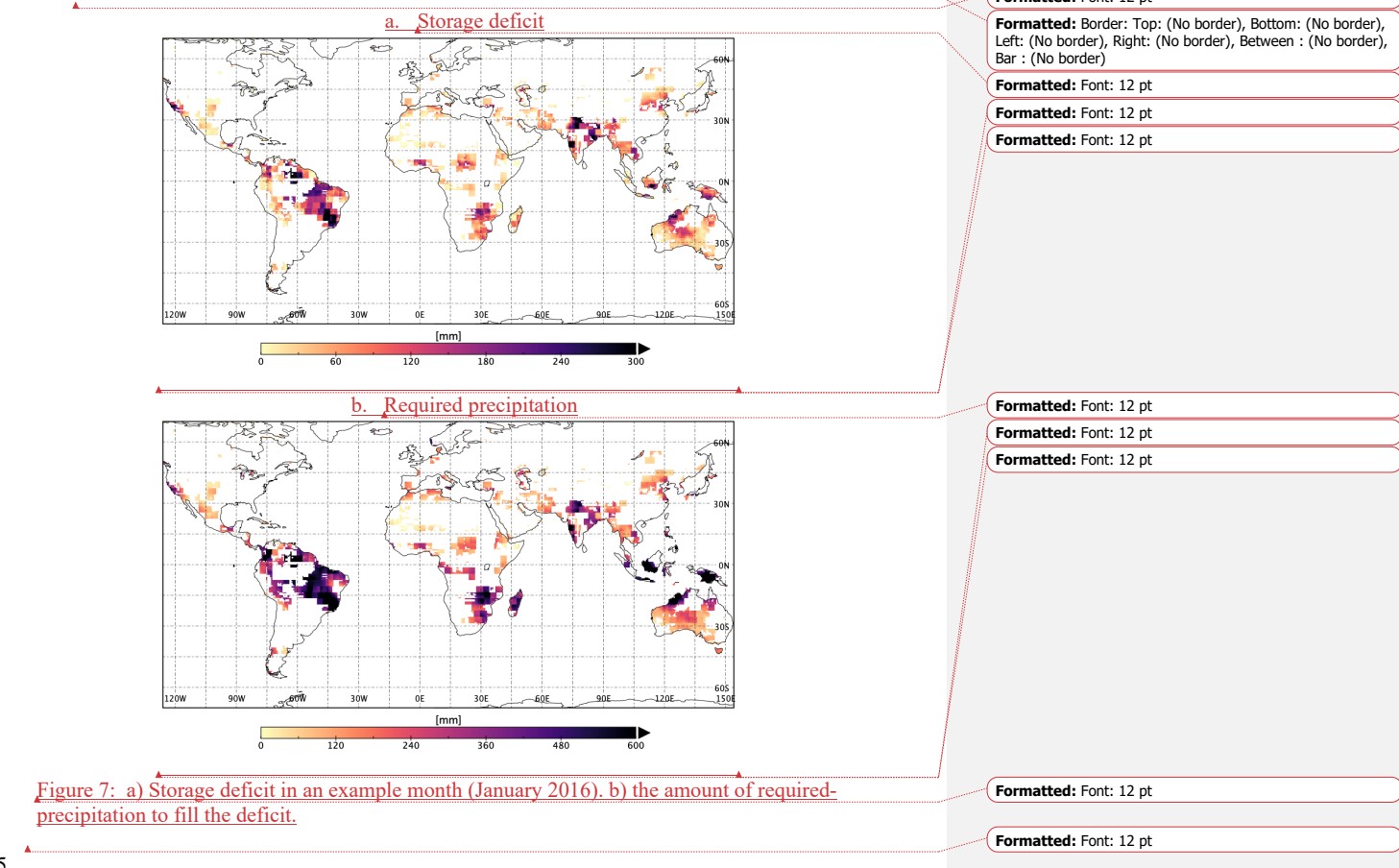

b. Required precipitation

Figure 7: a) Storage deficit in an example month (January 2016). b) the amount of required precipitation to fill the deficit.

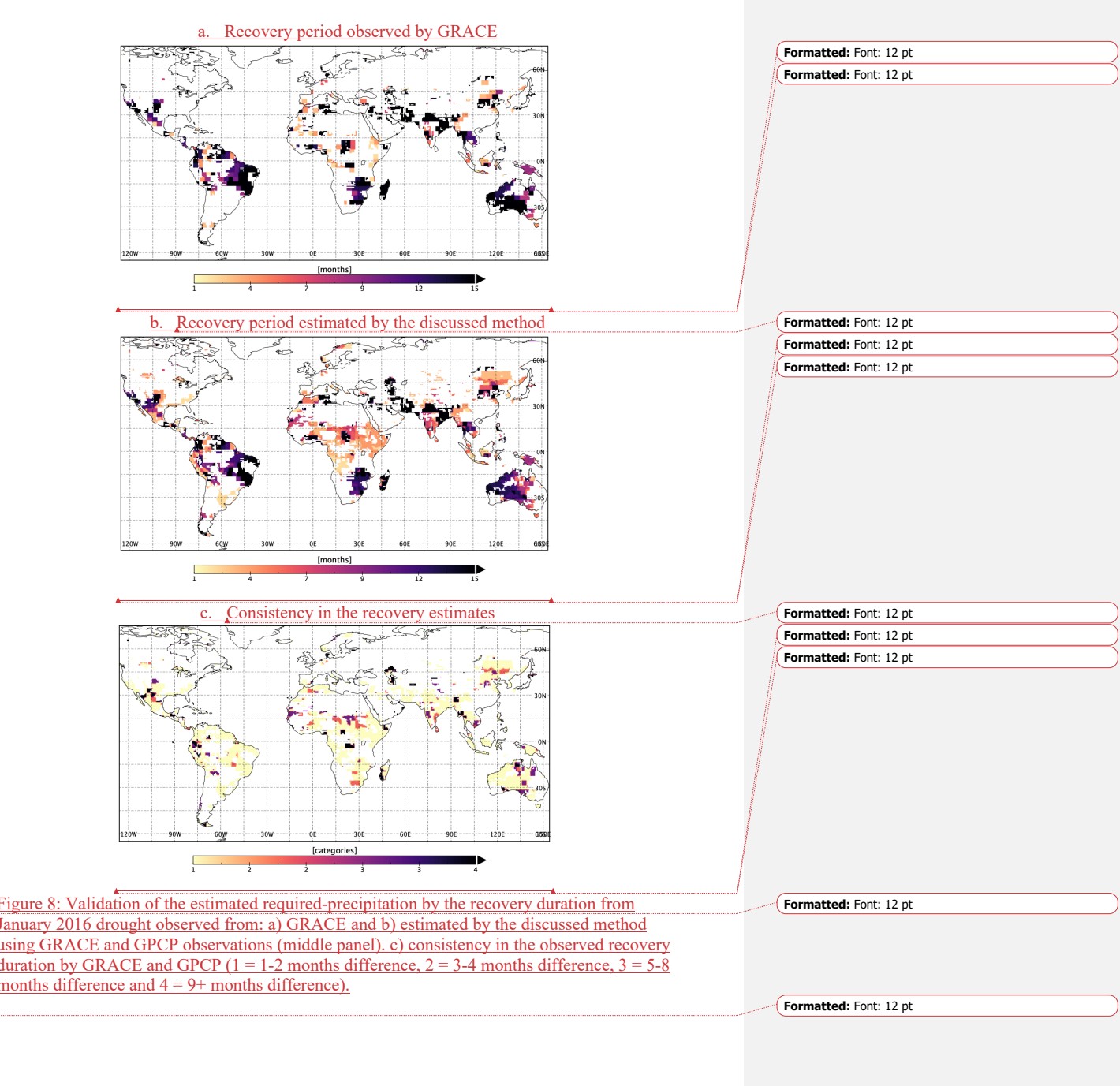

Figure 8: Validation of the estimated required-precipitation by the recovery duration from January 2016 drought observed from: a) GRACE and b) estimated by the discussed method using GRACE and GPCP observations (middle panel). c) consistency in the observed recovery duration by GRACE and GPCP (1 = 1-2 months difference, 2 = 3-4 months difference, 3 = 5-8 months difference and 4 = 9+ months difference).

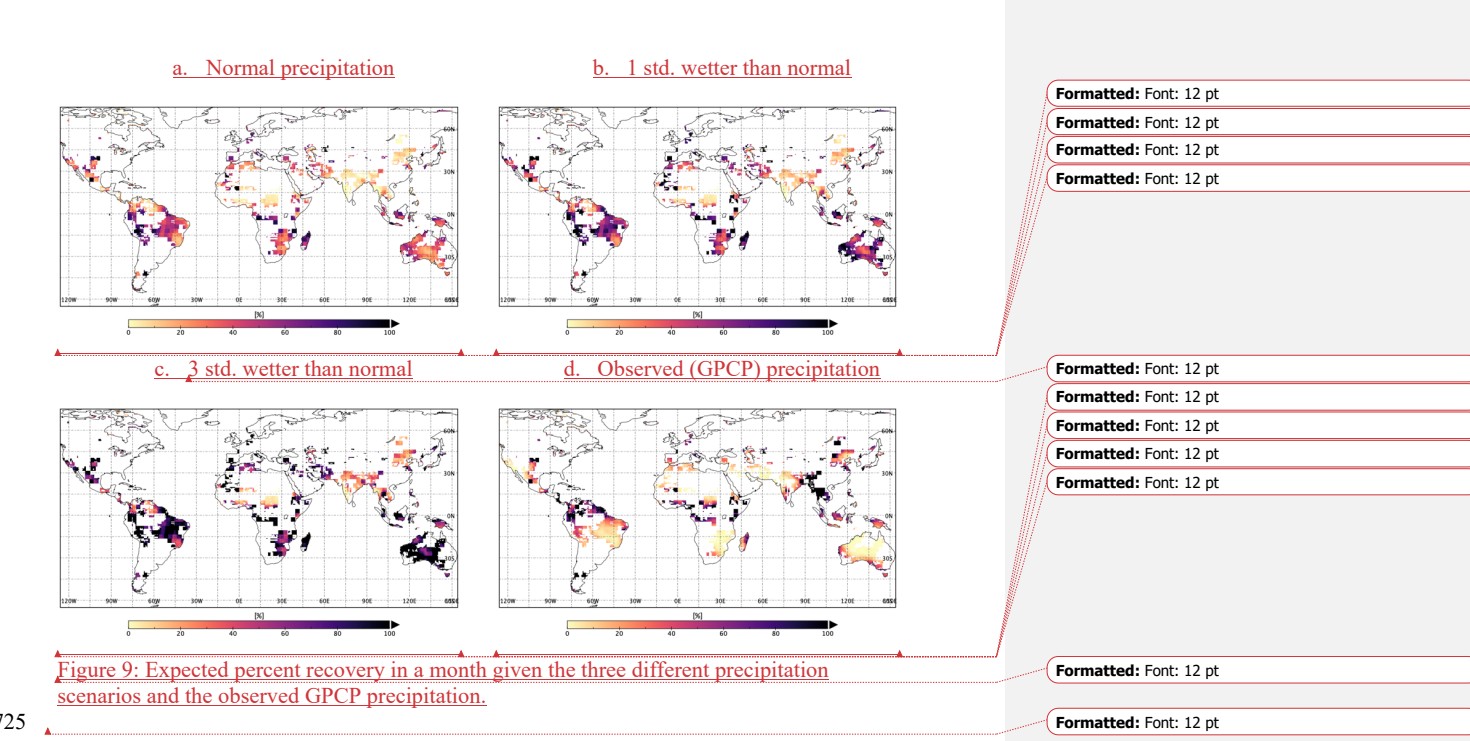

a.  Normal precipitation          b.  1 std. wetter than normal

c.  3 std. wetter than normal     d.  Observed (GPCP) precipitation

Figure 9: Expected percent recovery in a month given the three different precipitation scenarios and the observed GPCP precipitation.

a.   Normal precipitation          b.   1 std. wetter than normal

c.   3 std. wetter than normal          d.   Observed recovery duration by GRACE

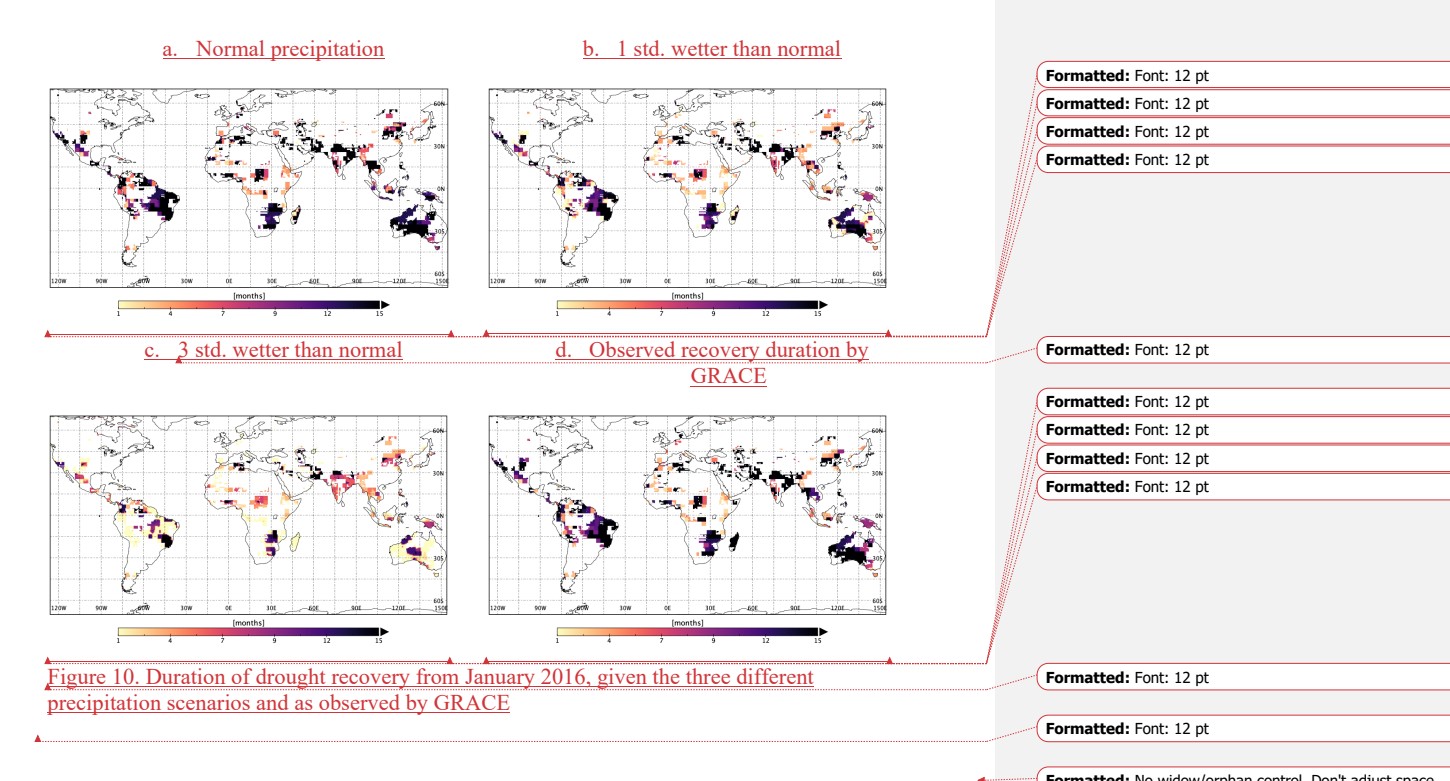

Figure 10. Duration of drought recovery from January 2016, given the three different precipitation scenarios and as observed by GRACE