# Peer review of "Estimation of hydrological drought recovery based on precipitation and GRACE water storage deficit"

_Hydrology and Earth System Sciences, 2019_

## Referee Comment (RC1) · Anonymous Referee #1 · 10 Feb 2020

Summary: the authors examine two different ways to estimate drought recovery: a storage deficit approach, in which GRACE TWSA is used to define the end of a drought, and a "required precipitation" approach that tracks (or forecasts) cumulative rainfall deficit. They conclude that there is good agreement between the two methods in most regions that satisfy tests of moderate or strong rainfall-storage coupling. Bringing these two methods together is both interesting and potentially valuable in the context of forecasts–presumably, for regions in which this analysis approach works well, a skillful precipitation forecast could be used to predict the cessation of TWSA drought up to several months in advance. Of course, this hinges on having such a skillful precipitation forecast, but the framework presented here provides a guide to how the prediction would be implemented.

[Figure]

I believe that the discussion paper can be accepted as a final HESS paper after moderate revision. My specific comments are listed below. I am particularly interested in the authors' response to comment #7, as I fear that I am missing some key element of their methodology. If I'm not missing something then I would recommend that the authors reframe or remove the forecast materials that led me to make that comment.

Specific comments:

1. line 18: what is "simplistic precipitation forecast skill"? I think some rephrasing is required.

2. Introduction: as stated in my summary, my understanding is that this study is motivated by (or, at least, could be motivated by) the problem of monitoring and forecasting the end of a drought on the basis of precipitation requirements. But it took me a while to come to that understanding, in part because the introduction does not, in my opinion, offer a clear statement of the intellectual contribution of this paper. There is good material reviewing GRACE and reviewing drought cessation estimates, but the final paragraph of the introduction simply states what the authors are going to do and not why they are doing it in the context of a gap in the literature or a target application. It would be helpful to have a few sentences that make the importance of this paper more clear.

3. GRACE data: how sensitive are these results to the choice of GRACE product? If only mascon are to be used then please justify the choice of mascon over spherical harmonics solutions for this application. Also, more than one mascon solution is now available, and it would be useful to see that the results presented here are robust to the choice of mascon product.

4. GPCP: similar question here. How sensitive is the analysis to choice of precipitation dataset? There are a number of choices available for the period of study.

5. line 110 et seq.: It is true that a long-term linear trend is often due to non-climatic

processes. But some GRACE trends ARE due to climate–for example, a major drought at the beginning or end of the record. The authors should comment on this possibility at some point in the manuscript, and discuss its implications for results in some regions.

6. line 158 et seq.: "Figure 2" in this passage is actually Figure 3.

7. Section 3.3.2 and other materials on forecasts: I have to admit that I don't understand the emphasis on these hindcasts in the paper. As the authors acknowledge, it's a simple method that doesn't provide very meaningful forecast. So what is it used for? It seems that the analysis presented in the results section only requires statistics of historical rainfall (mean and standard deviations) that can be compared to observation. The forecasts simply seem to play the role of a not-quite-perfect estimate of climatology. I do understand the authors' point about why forecasts might be useful in the context of predicting the end of drought via forecast of required precipitation. But there is no demonstration of this value in the current paper, as far as I can tell; there's only the claim that it might be valuable.

8. line 254: Doesn't blue n this figure indicate good agreement??

9. line 269 et seq.: It appears that Figure 10 is incorrectly referred to as Figure 8 throughout this passage.

10. Section 4.2.2: I assume that Figure 10 here really refers to Figure 11

11. I recommend an edit for style and grammar. The paper is clear, but there is some awkward phrasing.

---

## Referee Comment (RC2) · Anonymous Referee #2 · 16 Mar 2020

Summary:

The presented work shows an integrated precipitation approach to determine the recovery period and required precipitation to refill water storages and thus to overcome a hydrological drought. Thus, historical integrated precipitation is linked to total water storage anomalies (TWSA) by GRACE to combine and validate their precipitation-based methodology to an existing storage deficit methodology. Furthermore, three scenarios of precipitation forecast are provided to identify the best estimated time of recovery. They found that the recovery period of integrated precipitation is in good agreement with the recovery period from TWSA, especially in regions where integrated precipitation and total water storage changes showed a strong linear relationship. I think that this work discusses an important topic to have a better understanding of drought

evolution and to use this information possibly in water management. The methodology and findings are of good scientific quality and significance, but yet I have general and specific concerns, especially regarding to presentation quality, that are listed below. Thus, I recommend major revision, but believe that the manuscript could be published after addressing/clarifying my comments.

General comments

1. Until the first results were shown, it was not clear if the precipitation or the GRACE approach is the main contribution of the paper. This is important for abstract, introduction, conclusion and maybe should also be more consistent with the title and structure of the data and methods chapter. For example, [Page1 Line14] says the main goal is the combination of GRACE and precipitation, while [Page1 Line21] let assume that the author's main point is the precipitation approach and GRACE is only used as validation.

2. More clarification is needed about the drought definitions. Do you place your approach more in the context of hydrological drought or drought in general? The manuscript should be consistent according to the drought definitions. Be also clear about other drought categories of parameters, e.g.: [Page 1 Line32] meteorological drought is not only described by precipitation, also evapotranspiration. [Page1 Line34] soil moisture, precipitation, and runoff and not all hydrological parameters. For example, precipitation is a meteorological parameter.

3. Why are mascons used instead of spherical harmonics, the mascon solutions are underlying by constraints. Does the cap size of 3 x 3 degree of mascon solution then not represent a similar spatial resolution as the spherical harmonic GRACE resolution?

4. [Page3 Line103] Which method is used to regrid the data? Is there a precipitation data set with an 0.5 degree resolution? I ask myself if the downscaling from 2.5 to 0.5 degree has a significant impact.

5. [Page3 Line110] Why are the TWSA smoothed with an averaging filter? Does their

noise have a significant impact on the results?

6. [Page4 Line129-136] The linkage between integrated precipitation and GRACE is an important aspect for the validation so it should be explained more detailed. The paragraph is (probably) based on the water balance equation, which should at least be mentioned but better also shown. The assumptions that were decided to describe the relationship about evapotranspiration/runoff should be added here and it also should get clear how the precipitation is integrated in time. So for example, is it integrated continuously for each month to the previous months or is there an integration period of 3 months that is running over all months etc.?

7. [Page4 Lines144-147 and Lines158-162] It was not clear how the required precipitation is linked to the regression coefficients. It would great if the linkage for the example of a coefficient lower/higher/equal 1 in the first paragraph is clearly explained. Secondly, how do we then get the surplus required-precipitation? Is it derived by removing cdPA from dTWSA?

8. Figure 4, as well as some other figures, is analyzed too shortly (e.g. [Page5 Line181]) or, for example, only part a) of a), and b) is described. The figures provide much more information, especially about spatial differences. So, the figures should be described more in detail, which I prefer because they contain interesting findings, or removed/added to supplementary.

9. [Page5 Line188] It is not clear how the sub-seasonal signal is computed and where the number of 0 to 3 months of reconstruction is resulting from. The final hindcast is 2 years, so how did the authors manage the 0-3 months restriction of the sub-seasonal signal?

10. [Page7 Line247] The definition of severe drought was not exactly set. What is the definition or to which definition is it referred?

11. [Page7 Line253] Based on which principles are the differences of recovery months

divided into the different classes? How were the classes determined? It leads also to confusion in Figure 9. Without reading the caption it seems as if the difference is very small everywhere (from 1 to 4 months), but the number does not represent the "difference in months", rather the "class number of difference in months".

12. [Page9 Line333] Could you please discuss that the recovery period derived from precipitation is also underlying certain assumptions (e.g. about evapotranspiration)?

Specific comments

I would recommend to work through the manuscript again to remove grammatical/syntactic errors. Some examples: - [Page1 Line30] Missing commas, 'the', and 'and/or' (should also be checked: and/or is needed before last item of a list), suggestion: '... developing parts of the world, for example, the 2011 East Africa drought or the 2018 dry corridors of central America (REF).' - [Page2 Line56] have/has and "the" too much, suggestion: '... is independent of other drought indices and has global spatial coverage.' - [Page2 Line69] singular/plural, citing brackets, suggestion: '... reviewed different kinds of drought and their prediction methods based on statistical, dynamical, and hybrid methods. Panet et al. (2013) were ...' - [Page3 Line91] add date of last access for websites - [Page4 Line146] be consistent with required precipitation/required-precipitation - [Page 5 Line 181] be consistent with figure/Figure and section/Section - [Page5 Line190] estimated precipitation → reconstructed precipitation - [Page5 Line202] be consistent with climatology/annual signal

References that should be added: - [Page2 Line59] Reference for global gridded assessments - [Page2 Line62] Reference for increasing frequency of drought - [Page3 Line98] Reference for cubic convolution interpolation

[Page2 Line77] Please explain why only terrestrial water storage can be used instead of, for example, in-situ groundwater data.

[Page2 Line81] It could be added that you focus on sub-decadal drought because there

are only about 15 years of GRACE data.

[Page2 Line83] GPCP was not introduced yet.

[Page3 Line114] "Here, we define 'recovery' as a return to the climatological storage state for a given month." This is not totally clear to me, does it mean that the deviation from current dTWSA to the climatology itself in a specific month, which is referred to as severity in Thomas et al. (2014), is already the recovery?

[Page3 Line123] state of drought → severity of drought?

[Page4 Line125] Could you mark the three recovery periods in Figure 1, please? It seems as if the recovery periods are longer than 1.5, 1 and 0.5 years.

[Page5 Line167] ... are statistically analyzed using the methods of . . .

[Page5 Line184] The annual signal and linear trend extracted by signal decomposition . . .

[Page5 Line187] How was the number 10-14 months for autoregression chosen?

[Page5 Line200] worst → worse

[Page5 Line201], [Page7 Line271], and [Page7 Line283] etc.: 'In these regions...', 'this region', and 'monsoon regions' be precise which regions

[Page5 Line202] robust → dominant

[Page6 Line211] Where (reference) is it defined that one sigma represents a wet year and three sigma an exceptionally wet year?

[Page6 Line220] providing a minimum and maximum baseline?

[Page6 Line232] "In Figure7, observed precipitation (red dashed line) and absolute required precipitation (blue line) ..." This was already said.

Figure 7: This was quite hard to analyze. I would recommend to enlarge the subfigures

or put them in a different order (e.g. 4 x 1).

[Page6 Line241] some drought → drought

[Page6 Line241] Remove 'it is a random selection of the month for'

[Page7 Line254] blue → red?

[Page7 Line256] Is with 80% the total global land area or the masked global land area meant?

4.2.2 Different precipitation scenario → Precipitation scenarios

[Page7 Line 265] 'We stimulated one-month (February 2016)recovery period ...' Not clear what is meant

[Page8 Line288] Better more precise: Here we define drought severity and duration using ...

5 Discussion: Refer to section if different aspects/findings are discussed. [Page8 Line298] soil water column → water colum

[Page8 Line 299] Position of sentence in paragraph awkward in the previous context.

[Page9 Line327] Also shown in Figure 11 . . .

[Page9 Line342] 1) the independence from other drought indices → more precise, which independencies?

All Figures: Please check figure references in the text, some of the references have been mixed up. Make sure that all figure captions and title really describe what is shown (compared to what) e.g. Figure 4 fraction of a), b), and c) to what? Total of all. . . or Figure 9 validation of what by what? And consider changing colorbars, since some figure might better be represented in a different way, e.g. Figure 9 discrete colorbar.
* * *
[Figure]

590, 2019.

---

## Referee Comment (RC3) · Anonymous Referee #3 · 7 Apr 2020

**1 Summary**

The authors devise a novel method for estimating intradecadal drought recovery periods using GRACE and precipitation data globally. The total water storage estimates from GRACE are used to determine the deficit and the precipitation data is used for estimating the drought recovery periods using an empirical forecasting model. The issue is an important one in the context of ongoing climate change. Furthermore, the subject matter is also relevant for the journal and its audience. Having said that there are methodological issues in the data analysis which I will point out in the subsequent section, and the manuscript requires improvement in its narrative.

**2 General Comments**

- The title does not fully reflect the content of the manuscript. Firstly, the work only looks at short-term (intradecadal) droughts and secondly it uses precipitation in addition to GRACE to estimate the drought recovery times. These two aspects of the manuscript should be reflected in the title. Currently, going by the title, the drought recovery time is solely estimated from GRACE, which is incorrect.
- 2. The central goal of the manuscript seems to be to determine drought recovery times and that is facilitated by precipitation forecasts, and the majority of the manuscript is dedicated to figuring out an empirical way to predict precipitation. However, in the conclusions there is hardly any mention of precipitation and the empirical forecast model, and their role in drought recovery times. Rather it is concluded that the one of the findings is that GRACE can be used to derive drought indices, which appears to have been established by Thomas et al (2014).
- 3. Throughout the manuscript it is not clear as to what type of drought the authors are trying to quantify. In the title it is indicated that the authors are concerned about hydrological droughts, but nothing much is said in the manuscript. In the introduction they specify there are multiple definitions of droughts, but beyond that there is no indication on what sort of droughts the authors are interested in and which sorts will be sensitive to the method developed in the manuscript. It would be beneficial if the authors clarify this for the readers.
- 4. For the data the authors use GRACE JPL mascons for total water storage and GPCP for precipitation. Given the wide variety of data available both for total water storage (CSR mascons, GSFC mascons, CSR, GFZ, JPL, ITSG spherical harmonics, COST-G combined solutions) as well as precipitation (GPCC, CRU, Delaware), it would be interesting to know how different the drought recovery times would be if we were to choose a different pair of datasets. At least
in the case of GRACE it should be tested, because it is the starting point for the method proposed in the manuscript. Given the lack of consensus on which GRACE flavour is to be used, or how to reconcile the data, it is worthwhile to perform this test.

5. The GRACE and the GPCP datasets are represented on  $3^{\circ}$  spherical cap and  $2.5^{\circ} \times 2.5^{\circ}$  equi-angular grid. After indicating that the area of the unit representations are comparable, they represent the two datasets on a  $0.5^{\circ} \times 0.5^{\circ}$  grid to perform the analyses. There a couple of issues here. Firstly, the difference between the areas of the unit representations are at best  $\approx [10,000]km^2$  (at the equator) and at worst  $\approx [80,000]km^2$  (close to the poles). Secondly, by regridding them to a smaller grid size, they are only making map a bit smooth, but there is no change in the information content. The best way to bring them to a commensurate resolution to perform the data analysis would have been to filter them with a common filter either a Gaussian or any other contrast preserving filter, and then regrid them to any other grid size they wanted. It is essential that the authors discuss the impact of these data processing choices on the final results.

Based on these comments I recommend a major revision.

**3 Technical comments**

- 30 Please provide references for the events you have described
- 32 Please provide standard references for the drought definitions, for e.g., Wilhite and Glanz (1985). Water International
- 33 It is not clear what you want to convey by indicating the different indices.
- 38 Similar is the case for remote sensing data based drought indicators. Please clarify to the reader what their benefits and shortcomings are in order to get a perspective.
- 51 "This method can improve ..." until end of line 55. Please corroborate the statement, if it is not a conclusion of Thomas et al (2014).
- 59 "... are still a few" Please cite some of those studies
- 63 successive -> next
- 74 "However, above average ..." until end of line 77. Please clarify whether it is your opinion or a conclusion of Pan et al (2013)
- 84 In general, the introduction lacks a cogent narrative. It is hard to identify what issue you are trying to address
- 88 "... global and regional water cycle." Please provide a reference for the same.
- 104 When you say comparable, please indicate the numbers.
- 135 Please clarify to the reader why you need to integrate the precipitation timeseries.
- 142 The variability of precipitation intensity can be checked. It is unclear why this needs to be assumed.
- 189 The paragraph reads like the caption of Figure 5. Please interpret the figure for the reader as to what you want to convey through that figure.
- 199 Is the NSE performed on the full signals or after removing the climatology signal? It is well known that the climatology will dominate the metric if it is retained. Please clarify.
- 204 In Figure 6, please indicate the regions of weak association. Also, instead of a continuous scale, it would be better to use a discrete scale colorbar, i.e., one colour for a range of values. It is more convenient for the human eye to interpret such images.
- 265 stimulated -> simulated?
- 299 "hydrological compartments" Do you mean storage compartments?
- 342 "independency from other drought indices" Do you mean to say that SPI depends on other drought indices? Please clarify the "independence" argument.
- 343 "spatial coverage" Indices based on NDVI also cover much of the globe. How is this an advantage specific to the GRACE method?

Apart from the specific comments, I would like to indicate that it was rather frustrating to read such a methodology-heavy manuscript devoid of any equations. Even if the equations involved are simple and straight-forward I believe they will provide clarity for the reader. Please consider incorporating equations.

Your results largely fall into the sequential and diverging types of data for which colorbrewer2.org provides very good advice on choosing colorbars. Typically, sequential data require only one colour with varying intensity to indicate the sequences and diverging data requires two colours of varying intensities. Furthermore, the standard colorbars are not color-blind friendly. I strongly recommend that you follow the rules indicated in the website to improve the graphics in the manuscript. **HESSD**

---

## Author Comment (AC1) · 13 May 2020

Summary: the authors examine two different ways to estimate drought recovery: a storage deficit approach, in which GRACE TWSA is used to define the end of a drought, and a "required precipitation" approach that tracks (or forecasts) cumulative rainfall deficit. They conclude that there is good agreement between the two methods in most regions that satisfy tests of moderate or strong rainfall-storage coupling. Bringing these two methods together is both interesting and potentially valuable in the context of forecasts—presumably, for regions in which this analysis approach works well, a skillful precipitation forecast could be used to predict the cessation of TWSA drought up to several months in advance. Of course, this hinges on having such a skillful precipitation forecast, but the framework presented here provides a guide to how the prediction

would be implemented. I believe that the discussion paper can be accepted as a final HESS paper after moderate revision. My specific comments are listed below. I am particularly interested in the authors' response to comment #7, as I fear that I am missing some key element of their methodology. If I'm not missing something then I would recommend that the authors reframe or remove the forecast materials that led me to make that comment.

Response: We thank the reviewer for the positive comments.

Specific comments: 1. line 18: what is "simplistic precipitation forecast skill"? I think some rephrasing is required.

Author's response: We rephrased it to "simplistic precipitation forecast skill based on climatology and linear trend."

2. Introduction: as stated in my summary, my understanding is that this study is motivated by (or, at least, could be motivated by) the problem of monitoring and forecasting the end of a drought on the basis of precipitation requirements. But it took me a while to come to that understanding, in part because the introduction does not, in my opinion, offer a clear statement of the intellectual contribution of this paper. There is good material reviewing GRACE and reviewing drought cessation estimates, but the final paragraph of the introduction simply states what the authors are going to do and not why they are doing it in the context of a gap in the literature or a target application. It would be helpful to have a few sentences that make the importance of this paper more clear.

Author's Response: Thanks, we added a line as advised. "The intellectual contribution of this paper is in the estimation drought recovery and conceptually bringing a framework for drought recovery forecast based on precipitation deficit. "

3. GRACE data: how sensitive are these results to the choice of GRACE product? If only mascon are to be used then please justify the choice of mascon over spherical

**HESSD**
harmonics solutions for this application. Also, more than one mascon solution is now available, and it would be useful to see that the results presented here are robust to the choice of mascon product.

Author's response: The GRACE analysis in this paper is based on climatological anomalies of the three monthly smoothed and detrended TWS signals, therefore fine differences between different GRACE solutions after all these postprocessing gets minimized. Mascon based GRACE products have a relatively similar spatial resolution (3x3deg) as that of GPCP (2.5x 2.5deg). Section 2.2 talks about it, "The spatial resolution of the original GRACE solution (3-degree mascon) and GPCP (2.5-degree) are comparable. However, as mascon size varies with latitude, therefore to improve the interpretation both datasets are brought to the 0.5-degree grid. "

4. GPCP: similar question here. How sensitive is the analysis to choice of precipitation dataset? There are a number of choices available for the period of study.

Author's response: Yes we agree there are many precipitation products like CRU, GPCC, etc. However, GPCP is a widely used global precipitation data. GPCP combines the strength offered by in situ as well as satellite data. In many regions of the world in situ data are sparse, so using a product that only utilizes in situ data may not be the best choice. GPCP applies gauge under catch correction to in situ precipitation measurement, which has been found important to improve snowfall measurement (Behrangi et al. 2018). Besides, in section 3.3 historical analysis of the data is done using 1979-2017 precipitation data. For this period GPCP is the best available data. Behrangi, A., A. Gardner, J. T. Reager, J. B. Fisher, D. Yang, G. J. Huffman, and R. F. Adler (2018), Using GRACE to Estimate Snowfall Accumulation and Assess Gauge Undercatch Corrections in High Latitudes, Journal of Climate, 31(21), 8689-8704, doi: 10.1175/jcli-d-18-0163.1.

5. line 110 et seq.: It is true that a long-term linear trend is often due to non-climatic processes. But some GRACE trends ARE due to climate-for example, a major drought

**HESSD**
at the beginning or end of the record. The authors should comment on this possibility at some point in the manuscript, and discuss its implications for results in some regions.

Author's response: Thanks for bringing in, we added a line: "We acknowledge the caveat of a possibility of sudo trend due to unusual signal at the beginning or end of the record in some regions."

6. line 158 et seq.: "Figure 2" in this passage is actually Figure 3. Author's response: The maps in Figure 2 demonstrate the strength of the TWA-precipitation relationship globally. So, Figure 2 is correct.

7. Section 3.3.2 and other materials on forecasts: I have to admit that I don't understand the emphasis on these hindcasts in the paper. As the authors acknowledge, it's a simple method that doesn't provide very meaningful forecast. So what is it used for? It seems that the analysis presented in the results section only requires statistics of historical rainfall (mean and standard deviations) that can be compared to observation. The forecasts simply seem to play the role of a not-quite-perfect estimate of climatology. I do understand the authors' point about why forecasts might be useful in the context of predicting the end of drought via forecast of required precipitation. But there is no demonstration of this value in the current paper, as far as I can tell; there's only the claim that it might be valuable.

Author's response: The signal reconstruction and forecast discussed in section 3.3.2 is essential as we used it to create a normal signal first and then used standard deviation to simulate two additional precipitation scenarios of wet and extremely wet conditions. The normal signal is composed of predominantly climatology and long-term trend as the demonstrated model has the least competence in the estimation of inter-annual signals (0-3months). These precipitation scenarios are further needed to demonstrate the possible recovery duration from drought.

8. line 254: Doesn't blue n this figure indicate good agreement?? Author's response: That's right, thanks. Blue is changed to red.
9. line 269 et seq.: It appears that Figure 10 is incorrectly referred to as Figure 8 throughout this passage. Author's response: That's right, thanks, we corrected it.

10. Section 4.2.2: I assume that Figure 10 here really refers to Figure 11 Author's response: That's right, thanks for pointing out. Corrected.

11. I recommend an edit for style and grammar. The paper is clear, but there is some awkward phrasing. Author's response: Edited the manuscript. Many thanks for the supportive comments.

---

## Author Comment (AC2) · 14 May 2020

Summary: The presented work shows an integrated precipitation approach to determine the re- covery period and required precipitation to refill water storages and thus to overcome a hydrological drought. Thus, historical integrated precipitation is linked to total wa- ter storage anomalies (TWSA) by GRACE to combine and validate their precipitation- based methodology to an existing storage deficit methodology. Furthermore, three scenarios of precipitation forecast are provided to identify the best estimated time of re- covery. They found that the recovery period of integrated precipitation is in good agree- ment with the recovery period from TWSA, especially in regions where integrated pre- cipitation and total water storage changes showed a strong linear relationship. I think that this work discusses an important topic to have a better understanding of drought evolution and to use this information possibly in water management. The methodology and findings are of good scientific quality and significance, but yet I have general and specific concerns, especially regarding to presentation quality, that are listed below. Thus, I recommend major revision, but believe that the manuscript could be published after addressing/clarifying my comments.

Response: We agree and thank the reviewer for guiding the paper in such a detail to improve clarity and focus.

General comments 1. Until the first results were shown, it was not clear if the precipitation or the GRACE approach is the main contribution of the paper. This is important for abstract, introduction, conclusion and maybe should also be more consistent with the title and structure of the data and methods chapter. For example, [Page1 Line14] says the main goal is the combination of GRACE and precipitation, while [Page1 Line21] let assume that the author's main point is the precipitation approach and GRACE is only used as validation.

Author's response: We thank the reviewer for bringing this up. The paper uses both GRACE and GPCP equally, therefore, the title is modified as 'Estimation of hydrological drought recovery based on precipitation and GRACE water storage deficit'. GRACE is also used for validation but the main focus of this work is drought recovery estimate based on required precipitation, which is estimated from GRACE. We added a line in the introduction for more clarity. "The intellectual contribution of this paper is in the estimation drought recovery and conceptually bringing a framework for drought recovery forecast based on precipitation deficit. "

2. More clarification is needed about the drought definitions. Do you place your approach more in the context of hydrological drought or drought in general? The manuscript should be consistent according to the drought definitions. Be also clear about other drought categories of parameters, e.g.: [Page 1 Line32] meteorological drought is not only described by precipitation, also evapotranspiration. [Page1 Line34]

[Figure]

soil moisture, precipitation, and runoff and not all hydrological parameters. For example, precipitation is a meteorological parameter.

Author's response: A sentence is modified in the introduction to clarify that the study is more in the context of hydrological drought. ' This study focusses on hydrological drought, which requires, combining both surface (snow and surface water), and subsurface (soil moisture and groundwater) hydrological information. ' Thanks for pointing it, we modified the drought categories of parameters as 'including agricultural (soil moisture deficit), meteorological (eg. precipitation deficit or increase in evapotranspiration), and hydrological (storage deficit for eg. in streamflow/groundwater) droughts.'

3. Why are mascons used instead of spherical harmonics, the mascon solutions are underlying by constraints. Does the cap size of 3 x 3 degree of mascon solution then not represent a similar spatial resolution as the spherical harmonic GRACE resolution?

Author's response: The GRACE analysis in this paper is based on climatological anomalies of the three monthly smoothed and detrended TWS signal, therefore fine differences between different GRACE solutions after all these postprocessing gets minimized. Mascon based GRACE product has a relatively similar spatial resolution (3x3 deg) as that of GPCP (2.5x2.5deg). Section 2.2 talks about it, Section 2.2 talks about it, "The spatial resolution of the original GRACE solution (3-degree mascon) and GPCP (2.5-degree) are comparable. However, as mascon size varies with latitude, therefore to improve the interpretation both datasets are brought to the 0.5-degree grid. " However, we also acknowledge the spatial difference between them at different latitudes.

4. [Page3 Line103] Which method is used to regrid the data? Is there a precipitation data set with an 0.5-degree resolution? I ask myself if the downscaling from 2.5 to 0.5 degree has a significant impact.

Author's response: We used bilinear interpolation to regrid the GPCP data, in order to harmonize it with the GRACE grid. We agree with the reviewer's concern that it won't add any information by re-gridding 2.5 degree to 0.5 degrees and there are many precipitation products like CRU, GPCC, etc. However, GPCP is the best available global precipitation data, considering its spatial coverage, a combination of i-situ and remote sensing observations, and a longer time frame. GPCP combines the strength offered by in situ as well as satellite data. In many regions of the world in situ data are sparse, so using a product that only utilizes in situ data may not be the best choice. GPCP applies gauge under catch correction to in situ precipitation measurement, which has been found important to improve snowfall measurement (Behrangi et al. 2018). Besides, in section 3.3 historical analysis of the data is done using 1979-2017 precipitation data. For this period GPCP is the best available data.

Behrangi, A., A. Gardner, J. T. Reager, J. B. Fisher, D. Yang, G. J. Huffman, and R. F. Adler (2018), Using GRACE to Estimate Snowfall Accumulation and Assess Gauge Undercatch Corrections in High Latitudes, Journal of Climate, 31(21), 8689-8704, doi: 10.1175/jcli-d-18-0163.1.

[Page3 Line110] Why are the TWSA smoothed with an averaging filter? Does their noise have a significant impact on the results? Author's response: As drought develops in a smooth progression and we are looking for the amount of missing mass in a system caused by drought. Therefore, a 3months moving average is considered a better representation of the progression of drought. Monthly observations also have a similar relationship between TWS and precipitation but signals are neat and more intuitive after averaging filter.

5. [Page4 Line129-136] The linkage between integrated precipitation and GRACE is an important aspect for the validation so it should be explained more detailed. The paragraph is (probably) based on the water balance equation, which should at least be mentioned but better also shown. The assumptions that were decided to describe the relationship about evapotranspiration/runoff should be added here and it also should get clear how the precipitation is integrated in time. So for example, is it integrated continuously for each month to the previous months, or is there an integration period of 3 months that is running over all months, etc.?

Author's response: We understand the reviewer's point and added the following lines: " dS/dt = P – ET – R Eq. 1

The water balance equation based on hydrological fluxes ( Eq. 1) shows that the change in terrestrial water storage (dS) in a region for a given month (dt) depends on the monthly precipitation (P, mm/month); evapotranspiration (ET, mm/month) and the streamflow (R, which includes both surface water and subsurface water) (Swenson and Wahr, 2006). We assumed the relationship between P and (ET + R) remains constant for a region. Accordingly, the variability in precipitation shows the possible variation of storage in a month. Therefore, the amount of required-precipitation to overcome a deficit can be estimated using the association between precipitation and TWSA." Swenson, S. and Wahr, J.: Estimating Large-Scale Precipitation Minus Evapotranspiration from GRACE Satellite Gravity Measurements, J. Hydrometeor., 7(2), 252–270, doi:10.1175/JHM478.1, 2006.

6. [Page4 Lines144-147 and Lines158-162] It was not clear how the required precipitation is linked to the regression coefficients. It would great if the linkage for the example of a coefficient lower/higher/equal 1 in the first paragraph is clearly explained. Secondly, how do we then get the surplus required-precipitation? Is it derived by removing cdPA from dTWSA?

Author's response: It is a great idea; we added a small description: "Based on the linear relationship between dTWSA and cdPA the required precipitation has been estimated. Regression coefficients greater than 1 means the required precipitation is more than the amount of missing water. It is because precipitation lost in other hydrological processes like evapotranspiration, runoff ( Eq.1) is not observed by storage variability. Coefficient equals to 1 means the amount of required precipitation is the same as that storage loss, which means there is no other dominant process in that region. Coefficient less than 1 are the regions of weak precipitation-storage coupling, which can be due to other physical processes like melting of snow/frozen surfaces, groundwater extraction, irrigation, etc (non-red regions in Figure 2a)"

Figure 4, as well as some other figures, is analyzed too shortly (e.g. [Page5 Line181]) or, for example, only part a) of a), and b) is described. The figures provide much more information, especially about spatial differences. So, the figures should be described more in detail, which I prefer because they contain interesting findings, or removed/added to supplementary.

Author's response: We agree with the reviewer's point and added small description of the figure. "Figure 4 shows the fractional variance of the decomposed signal. For most regions annual signal dominates in precipitation (Figure 4a). However, regions where the wet season is not explicit in their climatology, high-frequency signal plays a major role, for example in central Europe, eastern Siberia, western N. America, southern Australia, etc. (Figure 4c). Contrarily, the long-term signal obtained by combining linear trend and the inter-annual signal has the least variability globally (Figure 4b). These smooth signals are driven by climate indices like El Niño southern oscillation (ENSO), Pacific decadal oscillation (PDO), and the North Pacific mode (NPM), etc. (Özger et al., 2009). The annual and long-term signals are directly applied for the signal reconstruction with the assumption that a similar trend will continue. Özger, M., Mishra, A. K. and Singh, V. P.: Low-frequency drought variability associated with climate indices, Journal of Hydrology, 364(1), 152–162, doi:10.1016/j.jhydrol.2008.10.018, 2009"

7. [Page5 Line188] It is not clear how the sub-seasonal signal is computed and where the number of 0 to 3 months of reconstruction is resulting from. The final hindcast is 2 years, so how did the authors manage the 0-3 months restriction of the sub-seasonal signal?

Author's response: Based on the possible confusion the reviewers pointed out, we added a sentence. "Sub-seasonal signal is obtained from the residual of inter-annual signal. This high-frequency signal has 0-3 months of temporal autocorrelation; accordingly, we have limited skill in synthesizing sub-seasonal signal."

8. [Page7 Line247] The definition of severe drought was not exactly set. What is the

definition or to which definition is it referred? Author's response: We added a line to make it clear. Thanks "Here, severity of a drought defined by the amount of water shortage per month."

[Page7 Line253] Based on which principles are the differences of recovery months divided into the different classes? How were the classes determined? It leads also to confusion in Figure 9. Without reading the caption it seems as if the difference is very small everywhere (from 1 to 4 months), but the number does not represent the "difference in months", rather the "class number of differences in months". Author's response: Y-label of the Figure-9c is modified (thanks for pointing it). The first two classes are defined by 2 months difference, as the majority of regions have less difference than the third class has 4 months difference and the last class has no upper limit.

9. [Page9 Line333] Could you please discuss that the recovery period derived from precipitation is also underlying certain assumptions (e.g. about evapotranspiration)? Author's response: We added the following line. "As discussed in Section 3.2, the underlying assumption of this work is that the relationship between precipitation, runoff, and evaporation for each location will remains unchanged. The required precipitation is derived from the GRACE observations, it inherits the relationship between P and ET based on equation 1. Therefore, the estimated required precipitation includes the impact of evaporation and runoff loss. "

Specific comments I would recommend working through the manuscript again to remove grammatical/syntactic errors. Some examples: - [Page1 Line30] Missing commas, 'the', and 'and/or' (should also be checked: and/or is needed before last item of a list), suggestion: '. . . developing parts of the world, for example, the 2011 East Africa drought or the 2018 dry corridors of central America (REF).'

Author's response: Thanks for pointing it, we have modified and added references as following: "example the 2011 East African drought (Lyon and DeWitt, 2012) or the 2014-16 dry corridors of central America (Guevara-Murua et al., 2018) Lyon, B. and

DeWitt, D. G.: A recent and abrupt decline in the East African long rains, Geophysical Research Letters, 39(2), doi:10.1029/2011GL050337, 2012. Guevara-Murua, A., Williams, C. A., Hendy, E. J. and Imbach, P.: 300 years of hydrological records and societal responses to droughts and floods on the Pacific coast of Central America, Clim. Past, 14(2), 175–191, doi:10.5194/cp-14-175-2018, 2018."

- [Page2 Line56] have/has and "the" too much, suggestion: '. . . is independent of other drought indices and has global spatial coverage.'

Author's response: Thanks for the correction, we have modified it as "The GRACE-based drought index is independent of the meteorological estimates and their combined uncertainties"

- [Page2 Line69] singular/plural, citing brackets, suggestion: '. . . reviewed different kinds of drought and their prediction methods based on statistical, dynamical, and hybrid methods. Panet et al. (2013) were ...' – Author's response: Corrected the citing bracket and singular/plural

[Page3 Line91] add the date of last access for websites – Author's response: Added the access date

[Page4 Line146] be consistent with required precipitation/required-precipitation – [Page 5 Line 181] be consistent with figure/Figure and section/Section - [Page5 Line190] estimated precipitation → reconstructed precipitation - [Page5 Line202] be consistent with climatology/annual signal Author's response: Changed to a consistent expression. Thanks!

References that should be added: - [Page2 Line59] Reference for global gridded assessments – Author's response: We added the following reference added (Gerdener et al., 2020; Li et al., 2019) Li, B., Rodell, M., Kumar, S., Beaudoing, H. K., Getirana, A., Zaitchik, B. F., Goncalves, L. G. de, Cossetin, C., Bhanja, S., Mukherjee, A., Tian, S., Tangdamrongsub, N., Long, D., Nanteza, J., Lee, J., Policelli, F., Goni, I.
B., Daira, D., Bila, M., Lannoy, G. de, Mocko, D., Steele‐Dunne, S. C., Save, H. and Bettadpur, S.: Global GRACE Data Assimilation for Groundwater and Drought Monitoring: Advances and Challenges, Water Resources Research, 55(9), 7564–7586, doi:10.1029/2018WR024618, 2019. Gerdener, H., Engels, O. and Kusche, J.: A framework for deriving drought indicators from the Gravity Recovery and Climate Experiment (GRACE), Hydrology and Earth System Sciences, 24(1), 227–248, doi:https://doi.org/10.5194/hess-24-227-2020, 2020.

[Page2 Line62] Reference for increasing frequency of drought – Author's response: Following reference added (Cook et al., 2014) Cook, B. I., Smerdon, J. E., Seager, R. and Coats, S.: Global warming and 21st century drying, Clim Dyn, 43(9), 2607–2627, doi:10.1007/s00382-014-2075-y, 2014.

[Page3 Line98] Reference for cubic convolution interpolation Author's response: Reference added (Keys, 1981) Keys, R.: Cubic convolution interpolation for digital image processing, IEEE Trans. Acoust., Speech, Signal Process., 29(6), 1153–1160, doi:10.1109/TASSP.1981.1163711, 1981.

[Page2 Line77] Please explain why only terrestrial water storage can be used instead of, for example, in-situ groundwater data. Author's response: We added a sentence. Thanks "With the sparse availability of in-situ groundwater observations and limited soil moisture observations upto top 5cm of the soil, complete profile of the water stored in a column can only be obtained from the GRACE-based terrestrial water storage."

[Page2 Line81] It could be added that you focus on sub-decadal drought because there are only about 15 years of GRACE data. Author's response: We modified the sentence. Thanks again! Here, we focus on sub-decadal drought only because of the availability of GRACE data for 15 years. The study can be extended for a longer time frame with the GRACE- follow on observations.

[Page2 Line83] GPCP was not introduced yet. Author's response: Added: Global Precipitation Climatology Project (GPCP)

[Figure]

[Page3 Line114] "Here, we define 'recovery' as a return to the climatological storage state for a given month." This is not totally clear to me, does it mean that the deviation from current dTWSA to the climatology itself in a specific month, which is referred to as severity in Thomas et al. (2014), is already the recovery? Author's response: Yes, decrease in severity is recovery.

[Page3 Line123] state of drought → severity of drought? Author's response: severity of drought changed to intensity of drought

[Page4 Line125] Could you mark the three recovery periods in Figure 1, please? It seems as if the recovery periods are longer than 1.5, 1 and 0.5 years. Author's response: oh yes! you are right. each grid is two years so it is almost 4, 2 and 1 years. We corrected it. Thanks for pointing. [Page5 Line167] ... are statistically analyzed using the methods of . . . Author's response: Added: using signal decomposition

[Page5 Line187] How was the number 10-14 months for autoregression chosen? Author's response: Based on the duration of significant auto-correlation with inter-annual signal.

[Page5 Line184] The annual signal and linear trend extracted by signal decomposition [Page5 Line200] worst → worse. [Page5 Line201], [Page7 Line271], and [Page7 Line283] etc.: 'In these regions...', 'this region', and 'monsoon regions' be precise which regions. [Page5 Line202] robust → dominant Author's response: corrected. Thanks!!

[Page6 Line211] Where (reference) is it defined that one sigma represents a wet year and three sigma an exceptionally wet year? Author's response: We assumed it, to generate three precipitation scenarios. The sentence is modified accordingly. "We assumed that one standard deviation wetter than normal precipitation as wet month and three standard deviations wetter than normal precipitation as exceptionally wet month."

[Page6 Line220] providing a minimum and maximum baseline? Author's response:

Even in exceptionally wet scenario in dry season, system fails to recover. Therefore, it does not provide maximum baseline.

[Page6 Line232] "In Figure7, observed precipitation (red dashed line) and absolute required precipitation (blue line) ..." This was already said. Author's response: Deleted. Thanks for pointing it. Figure 7: This was quite hard to analyze. I would recommend to enlarge the subfigures or put them in a different order (e.g. 4 x 1). Author's response: Modified most of the figures.

[Page6 Line241] some drought → drought [Page6 Line241] Remove 'it is a random selection of the month for' Author's response: Removed, thanks! [Page7 Line254] blue → red? Author's response: Corrected blue to red.

[Page7 Line256] Is with 80% the total global land area or the masked global land area meant? Author's response: Masked global area. Added the word 'masked'. Thanks! 4.2.2 Different precipitation scenario → Precipitation scenarios

[Page7 Line 265] 'We stimulated one-month (February 2016)recovery period ...' Not clear what is meant Author's response: This section shows the recovery percentage within a month based on the three-precipitation scenario.

[Page8 Line288] Better more precise: Here we define drought severity and duration using ... Author's response: Added 'drought intensity and duration' 5 Discussion: Refer to section if different aspects/findings are discussed. [Page8 Line298] soil water column → water column Author's response: Deleted 'soil', thanks! [Page8 Line 299] The Position of the sentence in paragraph awkward in the previous context. Author's response: Deleted the sentence. Thanks! [Page9 Line327] Also shown in Figure 11 . . . Author's response: Added (as shown in figure 8)

[Page9 Line342] 1) the independence from other drought indices → more precise, which independencies? Author's response: Added names of indices (PDSI, SPEI, SPI) and independency from the uncertainties of different meteorological variables and their

complex interactions. Thanks

All Figures: Please check figure references in the text, some of the references have been mixed up. Make sure that all figure captions and title really describe what is shown (compared to what) e.g. Figure 4 fraction of a), b), and c) to what? Total of all. . . or Figure 9 validation of what by what? And consider changing colorbars, since some figure might better be represented in a different way, e.g. Figure 9 discrete colorbar. Author's response: Modified most of the figures, please see the attachment. Many thanks for the very detailed review and constructive comments.

[Figure]

[Figure]

Figure 1: Water storage deficit from GRACE: The smoothed and detrended TWSA (dTWSA in red plot) is reduced by its climatology (black plot), to estimate deviation from the climatology. The negative residuals from the climatology are plotted on the upper axis as a green shaded area and scaled on the right side. The grey shade indicates ±1 standard deviation of the climatology.

**Fig. 1.**

---

## Author Comment (AC3) · 14 May 2020

The authors devise a novel method for estimating intradecadal drought recovery periods using GRACE and precipitation data globally. The total water storage estimates from GRACE are used to determine the deficit and the precipitation data is used for estimating the drought recovery periods using an empirical forecasting model. The issue is an important one in the context of ongoing climate change. Furthermore, the subject matter is also relevant for the journal and its audience. Having said that there are methodological issues in the data analysis which I will point out in the subsequent section, and the manuscript requires improvement in its narrative.

Author's response: We appreciate the constructive comment. We went over the paper

and tried to improve it by adding more clarification and improving the figures.

1.The title does not fully reflect the content of the manuscript. Firstly, the work only looks at short-term (intradecadal) droughts and secondly it uses precipitation in addition to GRACE to estimate the drought recovery times. These two aspects of the manuscript should be reflected in the title. Currently, going by the title, the drought recovery time is solely estimated from GRACE, which is incorrect.

Author's Response: Thanks for bringing this up, we have modified the title as follows: "Estimation of hydrological drought recovery based on precipitation and GRACE water storage deficit "

2.The central goal of the manuscript seems to be to determine drought recovery times and that is facilitated by precipitation forecasts, and the majority of the manuscript is dedicated to figuring out an empirical way to predict precipitation. However, in the conclusions, there is hardly any mention of precipitation and the empirical forecast model, and their role in drought recovery times. Rather it is concluded that the one of the findings is that GRACE can be used to derive drought indices, which appears to have been established by Thomas et al (2014).

Author's response: We understand the reviewer's concern. We mentioned in the manuscript that the precipitation forecast is not the focus of this work, so we preferred not to discuss it. The main idea of precipitation prediction is to generate 3 scenarios and it is mentioned. Section 3.3 states that 'Note that the motivation for providing a precipitation forecast here is not to present a state-of-the-art precipitation prediction but to demonstrate the potential utility of the terrestrial water storage deficit in determining required-precipitation and estimating a likely time to recovery. This methodology could be augmented with any type of more complex precipitation forecasting approaches.' I agree Thomas et al (2014) has already established that GRACE can be used to derive drought indices. However, the conclusion states that the 'GRACE based drought index is valid to estimate the required-precipitation for drought recovery.'

3. Throughout the manuscript, it is not clear as to what type of drought the authors are trying to quantify. In the title, it is indicated that the authors are concerned about hydrological droughts, but nothing much is said in the manuscript. In the introduction, they specify there are multiple definitions of droughts, but beyond that there is no indication on what sort of droughts the authors are interested in and which sorts will be sensitive to the method developed in the manuscript. It would be beneficial if the authors clarify this for the readers.

Author's Response: Thanks for bringing this up, we modified a sentence in the introduction. ' This study focusses on hydrological drought, which requires, combining both surface (snow and surface water), and subsurface (soil moisture and groundwater) hydrological information. '

4. For the data the authors use GRACE JPL mascons for total water storage and GPCP for precipitation. Given the wide variety of data available both for total water storage (CSR mascons, GSFC mascons, CSR, GFZ, JPL, ITSG spher- ical harmonics, COST-G combined solutions) as well as precipitation (GPCC, CRU, Delaware), it would be interesting to know how different the drought recov- ery times would be if we were to choose a different pair of datasets. At least in the case of GRACE it should be tested, because it is the starting point for the method proposed in the manuscript. Given the lack of consensus on which GRACE flavour is to be used, or how to reconcile the data, it is worthwhile to perform this test.

Author's response: We agree with the reviewer's concern. However, the GRACE analysis in this paper is based on climatological anomalies of the three monthly smoothed and detrended TWS signal. Therefore, the fine differences between different GRACE solutions get minimized after all these post-processing. Mascon based GRACE products have an approximately similar spatial resolution (3x 3) as that of GPCP (2.5x 2.5), so we chose one of the mascon solutions. There are many precipitation products also as CRU, GPCC, etc. However, GPCP is the best available global precipitation data, considering its spatial coverage, a combination of in-situ and remote

sensing observations, and a longer time frame. GPCP combines the strength offered by in situ as well as satellite data. In many regions of the world in situ data are sparse, so using a product that only utilizes in situ data may not be the best choice. GPCP applies gauge under catch correction to in situ precipitation measurement, which has been found important to improve snowfall measurement (Behrangi et al. 2018). Besides, in section 3.3 historical analysis of the data is done using 1979-2017 precipitation data. For this period GPCP is the best available data.

Behrangi, A., A. Gardner, J. T. Reager, J. B. Fisher, D. Yang, G. J. Huffman, and R. F. Adler (2018), Using GRACE to Estimate Snowfall Accumulation and Assess Gauge Undercatch Corrections in High Latitudes, Journal of Climate, 31(21), 8689-8704, doi: 10.1175/jcli-d-18-0163.1.

5. The GRACE and the GPCP datasets are represented on 3◦ spherical cap and 2.5◦ × 2.5◦ equi-angular grid. After indicating that the area of the unit representations are comparable, they represent the two datasets on a 0.5◦ × 0.5◦ grid to perform the analyses. There a couple of issues here. Firstly, the difference between the areas of the unit representations are at best ≈ [10, 000]km2 (at the equator) and at worst ≈ [80, 000]km2 (close to the poles). Secondly, by regridding them to a smaller grid size, they are only making map a bit smooth, but there is no change in the information content. The best way to bring them to a commensu- rate resolution to perform the data analysis would have been to filter them with a common filter either a Gaussian or any other contrast preserving filter, and then regrid them to any other grid size they wanted. It is essential that the authors discuss the impact of these data processing choices on the final results.

Author's Response: Thanks for bringing it so precisely. The mascon solution in the study is re-gridded by multiplying it with a scaling factor, to improve the interpretation of signals at sub-mascon resolution. This is essential as the shape and size of mascon changes with latitude. We agree that there are significant differences between the mascon (3x3 grid) and GPCP (2.5) area at different locations. The Following sentence

is added in section 3.2, thanks for the comment with numbers. 'Though GRACE mascon and GPCP 2.5 degree are considered as comparable, nevertheless areas of the unit representations are different at different locations like at equator $\approx$ 10, 000 km2 and close to poles 80, 000 km2. However, as drought is a smooth process the impact of neighboring pixels should not affect the analysis significantly.'

Based on these comments I recommend a major revision. 3 Technical comments 30 Please provide references for the events you have described Author's response: Reference added "example the 2011 East African drought (Lyon and DeWitt, 2012) or the 2014-16 dry corridors of central America (Guevara-Murua et al., 2018) Lyon, B. and DeWitt, D. G.: A recent and abrupt decline in the East African long rains, Geophysical Research Letters, 39(2), doi:10.1029/2011GL050337, 2012. Guevara-Murua, A., Williams, C. A., Hendy, E. J. and Imbach, P.: 300 years of hydrological records and societal responses to droughts and floods on the Pacific coast of Central America, Clim. Past, 14(2), 175–191, doi:10.5194/cp-14-175-2018, 2018."

32 Please provide standard references for the drought definitions, for e.g., Wilhite and Glanz (1985). Water International Author's response: Added the reference. Thanks (Wilhite and Glantz, 1985) 33 It is not clear what you want to convey by indicating the different indices. Author's response: In this study GRACE TWS is used as a drought index, therefore it is essential to describe a little about other common drought indices. 38 Similar is the case for remote sensing data based drought indicators. Please clarify to the reader what their benefits and shortcomings are in order to get a perspective. Author's Response: Thank, we added a sentence in the introduction With the sparse availability of in-situ groundwater observations and limited soil moisture observations (up to top 5cm of the soil), a complete profile of the water stored in a column can only be obtained from the GRACE-based terrestrial water storage.

51 "This method can improve ..." until end of line 55. Please corroborate the statement, if it is not a conclusion of Thomas et al (2014).

Author's response: The lines are moved to a paragraph below to separate it from Thomas et al. paper discussion and a sentence is added to it. "... This quantification of total required storage for drought recovery can only be estimated using GRACE observation." 59 "... are still a few" Please cite some of those studies Author's response: Reference added (Gerdener et al., 2020; Li et al., 2019)

Gerdener, H., Engels, O. and Kusche, J.: A framework for deriving drought indicators from the Gravity Recovery and Climate Experiment (GRACE), Hydrology and Earth System Sciences, 24(1), 227–248, doi:https://doi.org/10.5194/hess-24-227-2020, 2020.

Li, B., Rodell, M., Kumar, S., Beaudoing, H. K., Getirana, A., Zaitchik, B. F., Goncalves, L. G. de, Cossetin, C., Bhanja, S., Mukherjee, A., Tian, S., Tangdamrongsub, N., Long, D., Nanteza, J., Lee, J., Policelli, F., Goni, I. B., Daira, D., Bila, M., Lannoy, G. de, Mocko, D., Steele‐Dunne, S. C., Save, H. and Bettadpur, S.: Global GRACE Data Assimilation for Groundwater and Drought Monitoring: Advances and Challenges, Water Resources Research, 55(9), 7564–7586, doi:10.1029/2018WR024618, 2019.

63 successive –> next Author's response: Changed, thanks! 74 "However, above average ..." until end of line 77. Please clarify whether it is your opinion or a conclusion of Pan et al (2013) Author's response: The following line is added to separate it from Pan et al paper. Thanks! "Pan et-al., approach is exclusively precipitation based, however, ..."

84 In general, the introduction lacks a cogent narrative. It is hard to identify what issue you are trying to address

Author's response: We added an explicit sentence for that. "The intellectual contribution of this paper is in the estimation drought recovery and conceptually bringing a framework for drought recovery forecast based on precipitation deficit. "

88 "... global and regional water cycle." Please provide a reference for the same. Author's response: Added global (Eicker et al., 2016; Fasullo et al., 2016) and regional water cycle (Singh et al., 2018; Springer et al., 2017) Eicker, A., Forootan, E., Springer, A., Longuevergne, L. and Kusche, J.: Does GRACE see the terrestrial water cycle "intensifying"?, Journal of Geophysical Research: Atmospheres, 121(2), 733–745, doi:10.1002/2015JD023808, 2016. Fasullo, J. T., Lawrence, D. M. and Swenson, S. C.: Are GRACE-era Terrestrial Water Trends Driven by Anthropogenic Climate Change?, Advances in Meteorology, 2016, e4830603, doi:https://doi.org/10.1155/2016/4830603, 2016. Singh, A., Behrangi, A., Fisher, J. B. and Reager, J. T.: On the Desiccation of the South Aral Sea Observed from Spaceborne Missions, Remote Sensing, 10(5), 793, doi:10.3390/rs10050793, 2018. Springer, A., Eicker, A., Bettge, A., Kusche, J. and Hense, A.: Evaluation of the Water Cycle in the European COSMO-REA6 Reanalysis Using GRACE, Water, 9(4), 289, doi:10.3390/w9040289, 2017.

104 When you say comparable, please indicate the numbers. Author's response: Here we gave numbers of 3x3 degree for Mascon and 2.5 degree for GPCP. Additionally, as per your suggestion area details are added in section 3.2

135 Please clarify to the reader why you need to integrate the precipitation time- series. Author's response: Modified the sentence as following The smoothed and detrended precipitation anomaly is then integrated in time to get storage anomaly, which is termed as cumulative detrended smoothed precipitation anomaly (cdPA).

142 The variability of precipitation intensity can be checked. It is unclear why this needs to be assumed. Author's response: This assumption is for the estimation of required precipitation to consider the relationship between precipitation and storage variability stable. For example, a region having mostly slow rain has one kind of storage-precipitation relationship and if it gets unusual heavy rain then the relationship changes. Therefore, we assume here, that there is less variability in the precipitation intensity of a region.

189 The paragraph reads like the caption of Figure 5. Please interpret the figure for the

reader as to what you want to convey through that figure. Author's Response: Thanks for bringing this, we added a couple of sentences. "This precipitation reconstruction skill is used for a simplistic normal forecast. Further, two additional precipitation scenarios are simulated by adding respectively one and two standard deviations of precipitation to the normal forecast, which is used in the probability recovery analysis."

199 Is the NSE performed on the full signals or after removing the climatology signal? It is well known that the climatology will dominate the metric if it is retained. Please clarify. Author's Response: Many thanks for correcting it. Yes, initially NSE was calculated on the full signal, we have modified it now and performed the NSE on the non-climatological signal.

204 In Figure 6, please indicate the regions of weak association. Also, instead of a continuous scale, it would be better to use a discrete scale colorbar, i.e., one colour for a range of values. It is more convenient for the human eye to interpret such images. Author's response: Regions dominated by sub-seasonal signal has a weak association. Modified all of the figures with special consideration on the color bar.

265 stimulated –> simulated? Author's response: Corrected, thanks! 299 "hydrological compartments" – Do you mean storage compartments? Author's response: Modified to hydrological storage compartments

342 "independency from other drought indices" – Do you mean to say that SPI depends on other drought indices? Please clarify the "independence" argument. Author's response: Not, really  Thanks for bringing this. We modified the sentence. "GRACE-based drought index is independent of the meteorological uncertainties and their complex interactions." 343 "spatial coverage" – Indices based on NDVI also cover much of the globe. How is this an advantage specific to the GRACE method? Author's response: Removed 'spatial coverage' from the sentence.

Apart from the specific comments, I would like to indicate that it was rather frustrating to read such a methodology-heavy manuscript devoid of any equations. Even if the equa-

tions involved are simple and straight-forward I believe they will provide clarity for the reader. Please consider incorporating equations. Author's response: We understand the reviewer's viewpoint and added the water balance equation as following: dS/dt = P – ET – R Eq. 1

The water balance equation based on hydrological fluxes ( Eq. 1) shows that the change in terrestrial water storage (dS) in a region for a given month (dt) depends on the monthly precipitation (P, mm/month); evapotranspiration (ET, mm/month) and the streamflow (R, which includes both surface water and subsurface water) (Swenson and Wahr, 2006). We assumed the relationship between P and (ET + R) remains constant for a region. Accordingly, the variability in precipitation shows the possible variation of storage in a month. Therefore, the amount of required-precipitation to overcome a deficit can be estimated using the association between precipitation and TWSA." Swenson, S. and Wahr, J.: Estimating Large-Scale Precipitation Minus Evapotranspiration from GRACE Satellite Gravity Measurements, J. Hydrometeor., 7(2), 252–270, doi:10.1175/JHM478.1, 2006.

Your results largely fall into the sequential and diverging types of data for which colorbrewer2.org provides very good advice on choosing colorbars. Typically, sequential data require only one colour with varying intensity to indicate the sequences and diverging data requires two colours of varying intensities. Furthermore, the standard colorbars are not color-blind friendly. I strongly recommend that you follow the rules indicated in the website to improve the graphics in the manuscript. Author's response: All of the maps are modified with new color bars, please check the attachment.
* * *
Figure 1: Water storage deficit from GRACE: The smoothed and detrended TWSA (dTWSA in red plot) is reduced by its climatology (black plot), to estimate deviation from the climatology. The negative residuals from the climatology are plotted on the upper axis as a green shaded area and scaled on the right side. The grey shade indicates ±1 standard deviation of the climatology.

**Fig. 1.**

---

## Author Comment (AC4) · 14 May 2020

Please see the attached figures! Thank you very much!!

———————————————

a. Correlation coefficients

[Figure]

b. Regression Coefficients

[Figure]

Figure 2: a) Correlation coefficients and, b) regression coefficients between cumulative detrended precipitation anomalies (cdPA) and detrended terrestrial water storage anomaly (dTWSA).

**Fig. 1.**

[Figure]

a. Annual Signal

b. Linear trend + inter-annual signal

c. Sub-seasonal signal

Figure 4: Fractional variance of the decomposed signal to the full signal. a. Annual Signal, b. Long-term signal, c. sub-seasonal high frequency signal

**Fig. 2.**

Figure 6: Nash-Sutcliffe coefficients for 2016-17 precipitation hindcasting.

**Fig. 3.**

[Figure]

Figure 7 Validation of the required-precipitation estimate by drought recovery estimates at example locations. The different instances of drought show that drought ends (from the perspective of TWSA) whenever observed precipitation (red plot) exceeds the required-precipitation (blue plot).

**Fig. 4.**

a. Storage deficit

b. Required precipitation

Figure 8: a) Storage deficit in an example month (January 2016). b) the amount of required-precipitation to fill the deficit.

**Fig. 5.**

[Figure]

Figure 9: Validation of the estimated required-precipitation by the recovery duration from January 2016 drought observed from: a) GRACE and b) estimated by the discussed method using GRACE and GPCP observations (middle panel). c) consistency in the observed recovery duration by GRACE and GPCP (1 = 1-2 months difference, 2 = 3-4 months difference, 3 = 5-8 months difference and 4 = 9+ months difference).

**Fig. 6.**

a. Normal precipitation          b. 1 std. wetter than normal

c. 3 std. wetter than normal     d. Observed (GPCP) precipitation

Figure 10: Expected percent recovery in a month given the three different precipitation scenarios and
the observed GPCP precipitation.

**Fig. 7.**

---

## Author Comment (AC7) · 7 Jun 2020

We highly appreciate the reviewer for extensive and generous comments on the manuscript. Please find below the response. The modifications in the manuscript are shown within quotation marks.

Summary: The presented work shows an integrated precipitation approach to determine the re- covery period and required precipitation to refill water storages and thus to overcome a hydrological drought. Thus, historical integrated precipitation is linked to total wa- ter storage anomalies (TWSA) by GRACE to combine and validate their precipitation- based methodology to an existing storage deficit methodology. Furthermore, three scenarios of precipitation forecast are provided to identify the best estimated time of recovery. They found that the recovery period of integrated precipitation is in good agree- ment with the recovery period from TWSA, especially in regions where integrated pre- cipitation and total water storage changes showed a strong linear relationship. I think that this work discusses an important topic to have a better understanding of drought evolution and to use this information possibly in water management. The methodology and findings are of good scientific quality and significance, but yet I have general and specific concerns, especially regarding to presentation quality, that are listed below. Thus, I recommend major revision, but believe that the manuscript could be published after addressing/clarifying my comments.

Response: We agree and thank the reviewer for guiding the paper in such detail to improve its clarity and focus.

General comments 1. Until the first results were shown, it was not clear if the precipitation or the GRACE approach is the main contribution of the paper. This is important for abstract, introduction, conclusion and maybe should also be more consistent with the title and structure of the data and methods chapter. For example, [Page1 Line14] says the main goal is the combination of GRACE and precipitation, while [Page1 Line21] let assume that the author's main point is the precipitation approach and GRACE is only used as validation.

Author's response: We thank the reviewer for bringing this up. The paper uses both GRACE and GPCP equally, therefore, the title is modified as Estimation of hydrological drought recovery based on precipitation and GRACE water storage deficit. GRACE is also used for validation but the main focus of this work is drought recovery estimate based on required precipitation, which is estimated from GRACE. Added a line "The intellectual contribution of this paper is in the estimation drought recovery and conceptually bringing a framework for drought recovery forecast based on precipitation deficit. " (line 109)

2. More clarification is needed about the drought definitions. Do you place your

approach more in the context of hydrological drought or drought in general? The manuscript should be consistent according to the drought definitions. Be also clear about other drought categories of parameters, e.g.: [Page 1 Line32] meteorological drought is not only described by precipitation, also evapotranspiration. [Page1 Line34] soil moisture, precipitation, and runoff and not all hydrological parameters. For example, precipitation is a meteorological parameter.

Author's response: A sentence is modified in the introduction to clarify that the study is more in the context of hydrological drought. ' This study focusses on hydrological drought, which requires, combining both surface (snow and surface water), and subsurface (soil moisture and groundwater) hydrological information.' (line 42) drought categories of parameters are modified: "including agricultural (soil moisture deficit), meteorological (eg. precipitation deficit or increase in evapotranspiration), and hydrological (storage deficit for eg. in streamflow/groundwater) droughts." (line 40)

3. Why are mascons used instead of spherical harmonics, the mascon solutions are underlying by constraints. Does the cap size of 3 x 3 degree of mascon solution then not represent a similar spatial resolution as the spherical harmonic GRACE resolution?

Author's response: The GRACE analysis in this paper is based on climatological anomalies of the three monthly smoothed and detrended TWS signal, therefore fine differences between different GRACE solutions gets minimized. Underlying information of mascon and SH solution is same but in the equatorial region Mascon shows higher spatial resolution. Additionally, mascon based GRACE product have a relatively similar spatial resolution (3ïČřx 3ïČř) as that of GPCP (2.5ïČřx 2.5ïČř), so we selected Mascon solution.

4. [Page3 Line103] Which method is used to regrid the data? Is there a precipitation data set with an 0.5 degree resolution? I ask myself if the downscaling from 2.5 to 0.5 degree has a significant impact.

Author's response: GPCP is a widely used global precipitation data. In section 3.3

historical analysis 1979-2017 of the precipitation data is done. For this period GPCP 2.5degree is the best available data which is interpolated to 0.5 degree by using the bilinear method to harmonize it with the GRACE grid. There are many higher resolution precipitation products like TRMM, CRU, GPCC, etc. However, GPCP combines the strength offered by in situ as well as satellite data to obtain global picture while others are limited to 60 degrees North and South latitudes. Additionally, GPCP applies gauge under catch correction to in situ precipitation measurement, which has been found important to improve snowfall measurement (Behrangi et al. 2018). Besides, in section 3.3 historical analysis of the data is done using 1979-2017 precipitation data. For this period GPCP is the best available data. Behrangi, A., A. Gardner, J. T. Reager, J. B. Fisher, D. Yang, G. J. Huffman, and R. F. Adler (2018), Using GRACE to Estimate Snowfall Accumulation and Assess Gauge Under catch Corrections in High Latitudes, Journal of Climate, 31(21), 8689-8704, doi: 10.1175/jcli-d-18-0163.1. Added in the manuscript (line 140) "Global Precipitation Climatology Project (GPCP) is a widely used global precipitation data. Most of the other observational products don't produce precipitation estimates beyond 60deg S/N for longer historical period (1979 – present). Besides, GPCP applies gauge under catch correction to in situ precipitation measurement, which has been found important to improve snowfall measurement." (Behrangi et al., 2018)

5. [Page3 Line110] Why are the TWSA smoothed with an averaging filter? Does their noise have a significant impact on the results?

Author's response: As drought develops in a smooth progression and we are looking for the amount of missing mass in a system caused by drought. Therefore, a 3months moving average is considered a better representation of the progression of drought. Monthly observations also have a similar relationship between TWS and precipitation but signals are neat and better interpretable with averaging filter.

6. [Page4 Line129-136] The linkage between integrated precipitation and GRACE is an important aspect for the validation so it should be explained more detailed. The

paragraph is (probably) based on the water balance equation, which should at least be mentioned but better also shown. The assumptions that were decided to describe the relationship about evapotranspiration/runoff should be added here and it also should get clear how the precipitation is integrated in time. So for example, is it integrated continuously for each month to the previous months, or is there an integration period of 3 months that is running over all months, etc.?

Author's response: We understand the reviewer's point and added the following lines: " $dS/dt = P - ET - R$ Eq. 1

The water balance equation based on hydrological fluxes ( Eq. 1) shows that the change in terrestrial water storage (dS) in a region for a given month (dt) depends on the monthly precipitation (P, mm/month); evapotranspiration (ET, mm/month) and the streamflow (R, which includes both surface water and subsurface water) (Swenson and Wahr, 2006). Assuming the relationship between precipitation and ET + R remains constant for a region, the variability in precipitation gives an idea of possible variation in the storage Swenson, S. and Wahr, J.: Estimating Large-Scale Precipitation Minus Evapotranspiration from GRACE Satellite Gravity Measurements, J. Hydrometeor., 7(2), 252–270, doi:10.1175/JHM478.1, 2006. " (line181)

7. [Page4 Lines144-147 and Lines158-162] It was not clear how the required precipitation is linked to the regression coefficients. It would great if the linkage for the example of a coefficient lower/higher/equal 1 in the first paragraph is clearly explained. Secondly, how do we then get the surplus required-precipitation? Is it derived by removing cdPA from dTWSA?

Author's response: It is a great idea; we added a small description: "Based on the linear relationship between dTWSA and cdPA the required precipitation has been estimated. Regression coefficients greater than 1 means the required precipitation is more than the amount of missing water. This is because precipitation lost in other hydrological processes like evapotranspiration, runoff ( Eq.1) is not observed by storage variability).

Coefficient equals to 1 means the amount of required precipitation is the same as that storage loss, which means there is no other dominant process in the region. Coefficient less than 1 are the regions of weak precipitation-storage coupling, which can be due to other physical processes like melting of snow/frozen surfaces, groundwater extraction, irrigation, etc (non-red regions in Figure 2a)." (line 210)

8. Figure 4, as well as some other figures, is analyzed too shortly (e.g. [Page5 Line181]) or, for example, only part a) of a), and b) is described. The figures provide much more information, especially about spatial differences. So, the figures should be described more in detail, which I prefer because they contain interesting findings, or removed/added to supplementary.

Author's response: We agree with the reviewer's point and added a small description of the figure. "Figure 4 shows the fractional variance of the decomposed signal. For most regions, annual signals dominate in precipitation (Figure 4a). However, regions where the wet season is not explicit in their climatology, high-frequency signal plays a major role, for example in central Europe, eastern Siberia, western N. America, southern Australia, etc. (Figure 4c). Contrarily, the long-term signal obtained by combining linear trend and the inter-annual signal has the least variability globally (Figure 4b). These smooth signals are driven by climate indices like El NinÌČo southern oscillation (ENSO), Pacific decadal oscillation (PDO), and the North Pacific mode (NPM), etc. (OÌĹzger et al., 2009). The annual and long-term signals are directly applied for the signal reconstruction with the assumption that a similar trend will continue. " OÌĹzger, M., Mishra, A. K. and Singh, V. P.: Low-frequency drought variability associated with climate indices, Journal of Hydrology, 364(1), 152–162, doi:10.1016/j.jhydrol.2008.10.018, 2009" (line 260)

8. [Page5 Line188] It is not clear how the sub-seasonal signal is computed and where the number of 0 to 3 months of reconstruction is resulting from. The final hindcast is 2 years, so how did the authors manage the 0-3 months restriction of the sub-seasonal signal?

Author's response: A sentence added "The sub-seasonal signal is obtained from the residual of the inter-annual signal. This high-frequency signal has 0-3 months of temporal autocorrelation; accordingly, we have limited skill in synthesizing sub-seasonal signal." (line 275)

9. [Page7 Line247] The definition of severe drought was not exactly set. What is the definition or to which definition is it referred?

Author's response: Added the following sentence "Here, the severity of a drought defined by the amount of water shortage in a month." (line 348) [Page7 Line253] Based on which principles are the differences of recovery months divided into the different classes? How were the classes determined? It leads also to confusion in Figure 9. Without reading the caption it seems as if the difference is very small everywhere (from 1 to 4 months), but the number does not represent the "difference in months", rather the "class number of differences in months".

Author's response: Label of the figure is modified (thanks for pointing). The first two classes are defined by 2 months difference, as the majority of regions have less difference than the third class has 4 months difference and the last class has no upper limit.

10. [Page9 Line333] Could you please discuss that the recovery period derived from precipitation is also underlying certain assumptions (e.g. about evapotranspiration)?

Author's response: The underlying assumption of this work is that the relationship between precipitation, runoff, and evaporation for each location will remain unchanged. As the required precipitation is derived from the GRACE observations, it inherits the relationship between P and ET based on equation 1. Therefore, the estimated required precipitation includes the impact of evaporation and runoff loss.

Specific comments I would recommend to work through the manuscript again to remove grammatical/syntactic errors. Some examples: - [Page1 Line30] Missing com-

mas, 'the', and 'and/or' (should also be checked: and/or is needed before the last item of a list), suggestion: '. . . developing parts of the world, for example, the 2011 East Africa drought or the 2018 dry corridors of central America (REF).' Author's response: Modified and references added "example the 2011 East African drought (Lyon and DeWitt, 2012) or the 2014-16 dry corridors of central America (Guevara-Murua et al., 2018) ." - [Page2 Line56] have/has and "the" too much, suggestion: '. . . is independent of other drought indices and has global spatial coverage.'

Author's response: Modified the sentence. "The GRACE-based drought index is independent of the meteorological estimates and their combined uncertainties." - [Page2 Line69] singular/plural, citing brackets, suggestion: '. . . reviewed different kinds of drought and their prediction methods based on statistical, dynamical, and hybrid methods. Panet et al. (2013) were ...' – Author's response: Corrected the citing bracket and singular/plural [Page3 Line91] add date of last access for websites – Author's response: Added the access date [Page4 Line146] be consistent with required precipitation/required-precipitation – [Page 5 Line 181] be consistent with figure/Figure and section/Section - [Page5 Line190] estimated precipitation → reconstructed precipitation - [Page5 Line202] be consistent with climatology/annual signal Author's response: Changed to a consistent expression References that should be added: - [Page2 Line59] Reference for global gridded assessments – Author's response: Reference added (Gerdener et al., 2020; Li et al., 2019) [Page2 Line62] Reference for increasing frequency of drought – Author's response: Reference added (Cook et al., 2014) [Page3 Line98] Reference for cubic convolution interpolation Author's response: Reference added (Keys, 1981) [Page2 Line77] Please explain why only terrestrial water storage can be used instead of, for example, in-situ groundwater data.

Author's response: A line added. "With the sparse availability of in-situ groundwater observations and limited soil moisture observations upto top 5cm of the soil, complete profile of the water stored in a column can only be obtained from the GRACE-based terrestrial water storage." (line 105)

[Page2 Line81] It could be added that you focus on sub-decadal drought because there are only about 15 years of GRACE data. Author's response: Modified the line as below: "Here, we focus on sub-decadal drought only because of the availability of GRACE data for 15 years. The study can be extended for a longer time frame with the GRACE- follow on observations. "(line 115) [Page2 Line83] GPCP was not introduced yet. Author's response: corrected as Global Precipitation Climatology Project (GPCP) [Page3 Line114] "Here, we define 'recovery' as a return to the climatological storage state for a given month." This is not totally clear to me, does it mean that the deviation from current dTWSA to the climatology itself in a specific month, which is referred to as severity in Thomas et al. (2014), is already the recovery? Author's response: Yes, the decrease in severity is recovery. [Page3 Line123] state of drought → severity of drought? Author's response: severity of drought changed to intensity of drought

[Page4 Line125] Could you mark the three recovery periods in Figure 1, please? It seems as if the recovery periods are longer than 1.5, 1, and 0.5 years. Author's response: Thanks for pointed out. Yes, each grid is two years so it is almost 4, 2 and 1 years. [Page5 Line167] ... are statistically analyzed using the methods of . . . Author's response: Added "using signal decomposition " [Page5 Line187] How was the number 10-14 months for autoregression chosen? Author's response: Based on the duration of significant auto-correlation with the inter-annual signal. [Page5 Line184] The annual signal and linear trend extracted by signal decomposition [Page5 Line200] worst → worse. [Page5 Line201], [Page7 Line271], and [Page7 Line283] etc.: 'In these regions...', 'this region', and 'monsoon regions' be precise which regions. [Page5 Line202] robust → dominant Author's response: corrected [Page6 Line211] Where (reference) is it defined that one sigma represents a wet year and three sigma an exceptionally wet year? Author's response: These conditions are assumed in to generate three scenarios, the sentence is modified accordingly. "one standard deviation wetter than normal precipitation is assumed as a wet month and three standard deviations wetter than normal precipitation is assumed as an exceptionally wet month." [Page6 Line220] providing a minimum and maximum baseline? Author's response: Even in

exceptionally wet scenarios in the dry season, the system fails to recover. Therefore, it does not provide a maximum baseline. [Page6 Line232] "In Figure7, observed precipitation (red dashed line) and absolute required precipitation (blue line) ..." This was already said. Author's response: Deleted Figure 7: This was quite hard to analyze. I would recommend to enlarge the subfigures or put them in a different order (e.g. 4 x 1). Author's response: Modified [Page6 Line241] some drought → drought Author's response: Deleted 'some' [Page6 Line241] Remove 'it is a random selection of the month for' Author's response: Removed [Page7 Line254] blue → red? Author's response: Corrected the color [Page7 Line256] Is with 80% the total global land area or the masked global land area meant? Author's response: Masked global area. 4.2.2 Different precipitation scenario → Precipitation scenarios Author's response: Modified [Page7 Line 265] 'We stimulated one-month (February 2016) recovery period ...' Not Author's response: Modified as "This section shows the recovery percentage within a month based on the three precipitation scenarios." [Page8 Line288] Better more precise: Here we define drought severity and duration using ... Author's response: Added 'drought intensity and duration' 5 Discussion: Refer to section if different aspects/findings are discussed. [Page8 Line298] soil water column → water column Author's response: Deleted 'soil' [Page8 Line 299] Position of sentence in paragraph awkward in the previous context. Author's response: Deleted the sentence [Page9 Line327] Also shown in Figure 11 . . . Author's response: Added (as shown in figure 8) [Page9 Line342] 1) the independence from other drought indices → more precise, which independencies? Author's response: Added names of indices (PDSI, SPEI, SPI). Thanks All Figures: Please check figure references in the text, some of the references have been mixed up. Make sure that all figure captions and title really describe what is shown (compared to what) e.g. Figure 4 fraction of a), b), and c) to what? Total of all. . . or Figure 9 validation of what by what? And consider changing colorbars, since some figure might better be represented in a different way, e.g. Figure 9 discrete colorbar. Author's response: Modified most of the figures, please see the attachment. Many thanks for the very detailed review and constructive comments.

a. Correlation coefficients

[Figure]

b. Regression Coefficients

[Figure]

Figure 2: a) Correlation coefficients and, b) regression coefficients between cumulative detrended precipitation anomalies (cdPA) and detrended terrestrial water storage anomaly (dTWSA).

**Fig. 1.** figure2

[Figure]

a. Annual Signal

b. Linear trend + inter-annual signal

c. Sub-seasonal signal

Figure 4: Fractional variance of the decomposed signal to the full signal. a. Annual Signal, b. Long-term signal, c. sub-seasonal high frequency signal

**Fig. 2.** figure3

[Figure]

Figure 6 Validation of the required-precipitation estimate by drought recovery estimates at example locations. The different instances of drought show that drought ends (from the perspective of TWSA) whenever observed precipitation (red plot) exceeds the required-precipitation (blue plot).

**Fig. 3.** figure6

[Figure]

Figure 7: a) Storage deficit in an example month (January 2016). b) the amount of required-precipitation to fill the deficit.

**Fig. 4.** figure7

a. Recovery period observed by GRACE

b. Recovery period estimated by the discussed method

c. Consistency in the recovery estimates

Figure 8: Validation of the estimated required-precipitation by the recovery duration from January 2016 drought observed from: a) GRACE and b) estimated by the discussed method using GRACE and GPCP observations (middle panel). c) consistency in the observed recovery duration by GRACE and GPCP (1 = 1-2 months difference, 2 = 3-4 months difference, 3 = 5-8 months difference and 4 = 9+ months difference).

**Fig. 5.** figure8

a.   Normal precipitation    b.   1 std. wetter than normal

c.   3 std. wetter than normal    d.   Observed (GPCP) precipitation

Figure 10: Expected percent recovery in a month given the three different precipitation scenarios and
the observed GPCP precipitation.

**Fig. 6.** figure9

a. Normal precipitation        b. 1 std. wetter than normal

c. 3 std. wetter than normal        d. Observed (GPCP) precipitation

Figure 10. Duration of drought recovery from January 2016, given the three different precipitation
scenarios and as observed by GRACE

**Fig. 7.** figure10

---

## Author Comment (AC8) · 7 Jun 2020

We appreciate the constructive comment. We went over the paper and tried to improve it by adding more clarification and improving the figures. The modifications in the manuscript are put under quotation marks.

The authors devise a novel method for estimating intradecadal drought recovery periods using GRACE and precipitation data globally. The total water storage estimates from GRACE are used to determine the deficit and the precipitation data is used for estimating the drought recovery periods using an empirical forecasting model. The issue is an important one in the context of ongoing climate change. Furthermore, the subject matter is also relevant for the journal and its audience. Having said that there

are methodological issues in the data analysis which I will point out in the subsequent section, and the manuscript requires improvement in its narrative.

1. The title does not fully reflect the content of the manuscript. Firstly, the work only looks at short-term (intradecadal) droughts and secondly it uses precipitation in addition to GRACE to estimate the drought recovery times. These two aspects of the manuscript should be reflected in the title. Currently, going by the title, the drought recovery time is solely estimated from GRACE, which is incorrect.

Author's response: Thanks for bringing this up, we have modified the title as follows: 'Estimation of hydrological drought recovery based on precipitation and GRACE water storage deficit.'

2. The central goal of the manuscript seems to be to determine drought recovery times and that is facilitated by precipitation forecasts, and the majority of the manuscript is dedicated to figuring out an empirical way to predict precipitation. However, in the conclusions there is hardly any mention of precipitation and the empirical forecast model, and their role in drought recovery times. Rather it is concluded that the one of the findings is that GRACE can be used to derive drought indices, which appears to have been established by Thomas et al (2014).

Author's response: We understand the reviewer's concern. We mentioned in the manuscript that the precipitation forecast is not the focus of this work, so we preferred not to discuss it. The main idea of precipitation prediction is to generate 3 scenarios and it is mentioned. Section 3.3 states that 'Note that the motivation for providing a precipitation forecast here is not to present a state-of-the-art precipitation prediction, but to demonstrate the potential utility of the terrestrial water storage deficit in determining required-precipitation and estimating a likely time to recovery. This methodology could be augmented with any type of more complex precipitation forecasting approaches.' I agree, Thomas et al (2014) has already established that GRACE can be used to derive drought indices. However, the conclusion states that the 'GRACE based drought index

is valid to estimate the required-precipitation for drought recovery.'

3. Throughout the manuscript it is not clear as to what type of drought the authors are trying to quantify. In the title it is indicated that the authors are concerned about hydrological droughts, but nothing much is said in the manuscript. In the introduction they specify there are multiple definitions of droughts, but beyond that there is no indication on what sort of droughts the authors are interested in and which sorts will be sensitive to the method developed in the manuscript. It would be beneficial if the authors clarify this for the readers.

Author's response: Thanks for bringing this up, we modified a sentence in the introduction. ' This study focusses on hydrological drought, which requires, combining both surface (snow and surface water), and subsurface (soil moisture and groundwater) hydrological information. '

4. For the data the authors use GRACE JPL mascons for total water storage and GPCP for precipitation. Given the wide variety of data available both for total water storage (CSR mascons, GSFC mascons, CSR, GFZ, JPL, ITSG spher- ical harmonics, COST-G combined solutions) as well as precipitation (GPCC, CRU, Delaware), it would be interesting to know how different the drought recov- ery times would be if we were to choose a different pair of datasets. At least in the case of GRACE it should be tested, because it is the starting point for the method proposed in the manuscript. Given the lack of consensus on which GRACE flavour is to be used, or how to reconcile the data, it is worthwhile to perform this test.

Author's response: We understand the reviewer's concern. However, the GRACE analysis in this paper is based on climatological anomalies of the three monthly smoothed and detrended TWS signal, therefore fine differences between different GRACE solutions gets minimized. Mascon based GRACE product have approximately similar spatial resolution (3ïĆřx 3ïĆř) as that of GPCP (2.5ïĆřx 2.5ïĆř). Section 2.2 talks about it. Yes, we agree, there are many precipitation products like CRU, GPCC, etc. However,

GPCP is a widely used global precipitation data. GPCP combines the strength offered by in situ as well as satellite data. In many regions of the world in situ data are sparse, so using a product that only utilizes in situ data may not be the best choice. GPCP applies gauge under catch correction to in situ precipitation measurement, which has been found important to improve snowfall measurement (Behrangi et al. 2018). Besides, in section 3.3 historical analysis of the data is done using 1979-2017 precipitation data. For this period GPCP is the best available data. Behrangi, A., A. Gardner, J. T. Reager, J. B. Fisher, D. Yang, G. J. Huffman, and R. F. Adler (2018), Using GRACE to Estimate Snowfall Accumulation and Assess Gauge Under catch Corrections in High Latitudes, Journal of Climate, 31(21), 8689-8704, doi: 10.1175/jcli-d-18-0163.1. Added in the manuscript in line 140 "Global Precipitation Climatology Project (GPCP) is a widely used global precipitation data. Most of the other observational products don't produce precipitation estimates beyond 60deg S/N for longer historical period (1979 – present). Besides, GPCP applies gauge under catch correction to in situ precipitation measurement, which has been found important to improve snowfall measurement (Behrangi et al., 2018)."

5. The GRACE and the GPCP datasets are represented on 3âŮę spherical cap and 2.5âŮę × 2.5âŮę equi-angular grid. After indicating that the area of the unit representations are comparable, they represent the two datasets on a 0.5âŮę × 0.5âŮę grid to perform the analyses. There a couple of issues here. Firstly, the difference between the areas of the unit representations are at best $\approx [10, 000]km^2$ (at the equator) and at worst $\approx [80, 000]km^2$ (close to the poles). Secondly, by regridding them to a smaller grid size, they are only making map a bit smooth, but there is no change in the information content. The best way to bring them to a commensu- rate resolution to perform the data analysis would have been to filter them with a common filter either a Gaussian or any other contrast preserving filter, and then regrid them to any other grid size they wanted. It is essential that the authors discuss the impact of these data processing choices on the final results.
Author's response: Thanks for bringing it so precisely. The mascon solution in the study is re-gridded by multiplying it with a scaling factor, to improve the interpretation of signals at sub-mascon resolution. This is essential as the shape and size of mascon changes with latitude. We agree that there are significant differences between the mascon (3x3 grid) and GPCP (2.5) area at different locations. The Following sentence is added in section 4.2.1, thanks for the comment with numbers. 'Though GRACE mascon and GPCP 2.5 degree are considered as comparable, nevertheless areas of the unit representations are different at different locations like at equator $\approx$ 10, 000 km2 and close to poles 80, 000 km2. However, as drought is a smooth process the impact of neighboring pixels should not affect the analysis significantly.'

Based on these comments I recommend a major revision. 3 Technical comments 30 Please provide references for the events you have described Author's response: Reference added

example the 2011 East African drought (Lyon and DeWitt, 2012) or the 2014-16 dry corridors of central America (Guevara-Murua et al., 2018)

32 Please provide standard references for the drought definitions, for e.g., Wilhite and Glanz (1985). Water International Author's response: Added (Wilhite and Glantz, 1985)

33 It is not clear what you want to convey by indicating the different indices. Author's response: In this study GRACE TWS is used as a drought index, therefore it is essential to describe some other common drought indices and their limitations. The paragraph has been restructured to make it more clear.

38 Similar is the case for remote sensing data based drought indicators. Please clarify to the reader what their benefits and shortcomings are in order to get a perspective. Author's response: Thanks, we added a line in the introduction With the sparse availability of in-situ groundwater observations and limited soil moisture observations (up to top 5cm of the soil), a complete profile of the water stored in a column can only be obtained from the GRACE-based terrestrial water storage.

51 "This method can improve ..." until the end of line 55. Please corroborate the statement, if it is not a conclusion of Thomas et al (2014). Author's response: The lines are moved to a paragraph below to separate it from Thomas et al. paper discussion and a line added to it. . . . This quantification of total required storage for drought recovery can only be estimated using GRACE observation.

59 "... are still a few" Please cite some of those studies Author's response: Reference added (Gerdener et al., 2020; Li et al., 2019)

63 successive –> next Author's response: Changed

74 "However, above average ..." until end of line 77. Please clarify whether it is your opinion or a conclusion of Pan et al (2013) Author's response: The following line is added to separate it from Pan et al paper. Pan et-al., approach is exclusively precipitation based, however, . . .

84 In general, the introduction lacks a cogent narrative. It is hard to identify what issue you are trying to address Author's response: Added a line "The intellectual contribution of this paper is in the estimation drought recovery and conceptually bringing a framework for drought recovery forecast based on precipitation deficit. "

88 "... global and regional water cycle." Please provide a reference for the same. Author's response: Added "global (Eicker et al., 2016; Fasullo et al., 2016) and regional water cycle (Singh et al., 2018; Springer et al., 2017)." 104 When you say comparable, please indicate the numbers. Author's response: we discussed that 3 degree Mascon and 2.5 degree GPCP data products are comparable in spatial terms. Additionally, as per your suggestion area details are added in the section 4.2.1

135 Please clarify to the reader why you need to integrate the precipitation time- series. Modified the line as follows: "The smoothed and detrended precipitation anomaly is then integrated in time to get storage anomaly, which is termed as cumulative detrended smoothed precipitation anomaly (cdPA)."

[Figure]

142 The variability of precipitation intensity can be checked. It is unclear why this needs to be assumed. Author's response: This assumption is for the estimation of required precipitation to consider the relationship between precipitation and storage variability stable. For example, a region having mostly slow rain has one kind of storage-precipitation relationship and if it gets unusual heavy rain then the relationship changes. Therefore, we assume here, that there is less variability in the precipitation intensity of a region.

189 The paragraph reads like the caption of Figure 5. Please interpret the figure for the reader as to what you want to convey through that figure. Author's response: Thanks, a couple of sentences added. This precipitation reconstruction skill is used for a simplistic normal forecast. Further, two additional precipitation scenarios are simulated by adding respectively one and two standard deviations of precipitation to the normal forecast, which is used in probability recovery analysis.

199 Is the NSE performed on the full signals or after removing the climatology sig- nal? It is well known that the climatology will dominate the metric if it is retained. Please clarify. Author's response: Many thanks for the correction, NSE section is removed.

204 In Figure 6, please indicate the regions of weak association. Also, instead of a continuous scale, it would be better to use a discrete scale colorbar, i.e., one colour for a range of values. It is more convenient for the human eye to interpret such images. Author's response: Regions dominated by sub-seasonal signal has a weak association. Modified the figure

265 stimulated –> simulated? Author's response: Corrected 299 "hydrological compartments" – Do you mean storage compartments? Author's response: Modified: hydrological storage compartments

342 "independency from other drought indices" – Do you mean to say that SPI depends on other drought indices? Please clarify the "independence" argument. Author's response: Modified: The GRACE-based drought index is independent of the

meteorological estimates and their combined uncertainties

343 "spatial coverage" – Indices based on NDVI also cover much of the globe. How is this an advantage specific to the GRACE method? Author's response: Modified: The GRACE-based drought index is independent of the meteorological estimates and their combined uncertainties Apart from the specific comments, I would like to indicate that it was rather frustrating to read such a methodology-heavy manuscript devoid of any equations. Even if the equations involved are simple and straight-forward I believe they will provide clarity for the reader. Please consider incorporating equations.

Author's response: Equation and its description added " dS/dt = P – ET – R Eq. 1

The water balance equation based on hydrological fluxes ( Eq. 1) shows that the change in terrestrial water storage (dS) in a region for a given month (dt) depends on is the monthly precipitation (P, mm/month); evapotranspiration (ET, mm/month) and streamflow (R, which includes both surface water and subsurface water) (Swenson and Wahr, 2006). Assuming the relationship between precipitation and ET + R remains constant for a region, the variability in precipitation gives an idea of possible variation in the storage. Swenson, S. and Wahr, J.: Estimating Large-Scale Precipitation Minus Evapotranspiration from GRACE Satellite Gravity Measurements, J. Hydrometeor., 7(2), 252–270, doi:10.1175/JHM478.1, 2006. "

Your results largely fall into the sequential and diverging types of data for which colorbrewer2.org provides very good advice on choosing colorbars. Typically, sequential data require only one colour with varying intensity to indicate the sequences and diverging data requires two colours of varying intensities. Furthermore, the standard colorbars are not color-blind friendly. I strongly recommend that you follow the rules indicated in the website to improve the graphics in the manuscript.

Author's response: All of the maps are modified with new color bars, please check the attachment.

[Figure]

a. Correlation coefficients

[Figure]

b. Regression Coefficients

[Figure]

Figure 2: a) Correlation coefficients and, b) regression coefficients between cumulative detrended precipitation anomalies (cdPA) and detrended terrestrial water storage anomaly (dTWSA).

**Fig. 1.** figure2

[Figure]

Figure 4: Fractional variance of the decomposed signal to the full signal. a. Annual Signal, b. Long-term signal, c. sub-seasonal high frequency signal

**Fig. 2.** figure3

[Figure]

Figure 6 Validation of the required-precipitation estimate by drought recovery estimates at example locations. The different instances of drought show that drought ends (from the perspective of TWSA) whenever observed precipitation (red plot) exceeds the required-precipitation (blue plot).

**Fig. 3.** figure6

[Figure]

Figure 7: a) Storage deficit in an example month (January 2016). b) the amount of required-precipitation to fill the deficit.

**Fig. 4.** figure7

[Figure]

Figure 8: Validation of the estimated required-precipitation by the recovery duration from January 2016 drought observed from: a) GRACE and b) estimated by the discussed method using GRACE and GPCP observations (middle panel). c) consistency in the observed recovery duration by GRACE and GPCP (1 = 1-2 months difference, 2 = 3-4 months difference, 3 = 5-8 months difference and 4 = 9+ months difference).

**Fig. 5.** figure8

a. Normal precipitation     b. 1 std. wetter than normal

c. 3 std. wetter than normal     d. Observed (GPCP) precipitation

Figure 10: Expected percent recovery in a month given the three different precipitation scenarios and the observed GPCP precipitation.

**Fig. 6.** figure9

a. Normal precipitation  b. 1 std. wetter than normal

c. 3 std. wetter than normal  d. Observed (GPCP) precipitation

Figure 10. Duration of drought recovery from January 2016, given the three different precipitation scenarios and as observed by GRACE

**Fig. 7.** figure10

---

## Author Response (AR1)

**Reviewer1:**

Thank you very much for your support and generous comments. Please find below the response and respective modifications in the manuscript (shown in the quotation mark).

Summary: the authors examine two different ways to estimate drought recovery: a storage deficit approach, in which GRACE TWSA is used to define the end of a drought, and a required precipitation approach that tracks (or forecasts) cumulative rainfall deficit. They conclude that there is good agreement between the two methods in most regions that satisfy tests of moderate or strong rainfall-storage coupling. Bringing these two methods together is both interesting and potentially valuable in the context of forecasts–presumably, for regions in which this analysis approach works well, a skillful precipitation forecast could be used to predict the cessation of TWSA drought up to several months in advance. Of course, this hinges on having such a skillful precipitation forecast, but the framework presented here provides a guide to how the prediction would be implemented. I believe that the discussion paper can be accepted as a final HESS paper after moderate revision. My specific comments are listed below. I am particularly interested in the authors' response to comment #7, as I fear that I am missing some key element of their methodology. If I'm not missing something then I would recommend that the authors reframe or remove the forecast materials that led me to make that comment.

Response: We really appreciate the supportive and positive comments from the reviewer.

Specific comments:
1. line 18: what is "simplistic precipitation forecast skill"? I think some rephrasing is required.

Author's response: We rephrased it to "simplistic precipitation forecast skill based on the integration of climatology and long-term trend." (line 21)

2. Introduction: as stated in my summary, my understanding is that this study is motivated by (or, at least, could be motivated by) the problem of monitoring and forecasting the end of a drought on the basis of precipitation requirements. But it took me a while to come to that understanding, in part because the introduction does not, in my opinion, offer a clear statement of the intellectual contribution of this paper. There is good material reviewing GRACE and reviewing drought cessation estimates, but the final paragraph of the introduction simply states what the authors are going to do and not why they are doing it in the context of a gap in the literature or a target application. It would be helpful to have a few sentences that make the importance of this paper clearer.

Author's Response: Thanks, we added a line as advised. "The intellectual contribution of this paper is in the estimation drought recovery and conceptually bringing a framework for drought recovery forecast based on precipitation deficit." (line 110) . Additionally, we rearranged the paragraph to make it clearer.

3. GRACE data: how sensitive are these results to the choice of GRACE product? If only mascon are to be used then please justify the choice of mascon over spherical harmonics solutions for this application. Also, more than one mascon solution is now available, and

it would be useful to see that the results presented here are robust to the choice of mascon product.

Author's response: We understand the reviewer's concern about the possible discrepancies between different GRACE solutions, as they are produced using different approaches by different centers. We added a couple of lines in the GRACE section and added a supplementary material comparing different GRACE solution for an example location. In this study we used JPL based mascon GRACE solution as it has a relatively similar spatial resolution (3°x 3°) as that of GPCP (2.5°x 2.5°). We have also acknowledged that though GRACE mascon and GPCP 2.5 degree is considered as comparable, nevertheless areas of the unit representations are different at different locations
Following lines are added in the section 2.1

"The GRACE level-3 solution is officially available from three different centers, which are produced using different approaches (spherical harmonics or mascon), different filers, smoothing factors, etc. and eventually, there can be discrepancies between different TWS estimates from GRACE solutions (Jing et al., 2019). The differences between GRACE solutions from different centers are mostly very small at a basin level and lie within the error bounds of the GRACE solution itself (Sakumura et al., 2014). However, at 0.5-degree grid the difference between the amount of missing water estimated by the different GRACE solutions increases substantially (as seen in the supplementary material). This caveat can be mitigated in future by using ensemble of GARCE products. Nevertheless, they are consistent with the detection of drought duration. Please look into the supplementary material for the comparison between water storage deficit estimated by the different GRACE solutions.

Supplementary material

The official GRACE Science Data System continuously released monthly GRACE solution for three different processing centers: GeoforschungsZentrum Potsdam (GFZ), Center for Space Research at University of Texas, Austin (CSR), and Jet Propulsion Laboratory (JPL). These three solutions used different parameters and strategies, such as different degree and order, spherical harmonic coefficient, spatial filter and smoothing factor (Jing et al., 2019). The JPL mascon (JPL-M) and CSR mascon (CSR-M) solutions were provided at 0.5 degrees at https://grace.jpl.nasa.gov/data/get-data/monthly-mass-grids-land/ and http://www2.csr.utexas.edu/grace/RL05_mascons.html respectively. GFZ produces only spherical harmonic solution (GFZ-SH), which is downloaded from http://isdc.gfz-potsdam.de/grace-isdc/.
Figure11 shows water storage deficit and estimated required extra precipitation to overcome the drought based on three different GRACE solutions, using the method described in the article for an example location in India (centered on 77.25°E 15.25°N). Sakumura et al., 2014 demonstrated that at a basin-scale, the differences between them are very less. However, the plot shows that there are discrepancies between the amount of missing water at 0.5degree grid because every center has a different method to downscale GRACE inherent spatial resolution to high-resolution grid. Figure 11a shows the detrended climatological anomaly of the three GRACE solutions and cumulative GPCP precipitation has similar variability. The negative anomaly from climatology is

considered as drought and is plotted in figure11b. It shows that the difference between CSR and JPL solutions are relatively less than GFZ solution because the first two are the mascon-based solution and are available at 0.5-degree grid (after scaling), while GFZ solution is spherical harmonics based and is re-gridded to 0.5 degree from 1-degree spatial resolution by simple bilinear interpolation. Based on the linear relationship between cumulative detrended GPCP anomaly and detrended GRACE anomalies, required extra precipitation is estimated (figure 11c). The figure shows that the required precipitation varies based on GRACE solutions. Nevertheless, all three GRACE solutions are consistent with the detection of drought duration.

a.

[Figure]

b.

[Figure]

c.

[Figure]

*Figure 11: Water storage deficit and estimated required precipitation based on the spherical harmonic solution from GFZ and the mascon solutions from CSR and JPL .a. Cumulative detrended precipitation anomaly (cdPA) compared with the detrended storage anomaly (dTWSA). b) The negative residuals from the GRACE climatology considered as drought. c) Surplus required-precipitation is estimated based on the linear relationship between dTWSA and cdPA, to fill the storage deficit (middle panel)).*

Jing, W., Zhang, P. and Zhao, X.: A comparison of different GRACE solutions in terrestrial water storage trend estimation over Tibetan Plateau, Sci Rep, 9, doi:10.1038/s41598-018-38337-1, 2019.

Sakumura, C., Bettadpur, S. and Bruinsma, S.: Ensemble prediction and intercomparison analysis of GRACE time-variable gravity field models, Geophysical Research Letters, 41(5), 1389–1397, doi:10.1002/2013GL058632, 2014.

4.  GPCP: similar question here. How sensitive is the analysis to choice of precipitation dataset? There are a number of choices available for the period of study.

Author's response: Yes, we agree, there are many precipitation products like CRU, GPCC, etc. However, GPCP is a widely used global precipitation data. GPCP combines the strength offered by in situ as well as satellite data. In many regions of the world in situ data are sparse, so using a product that only utilizes in situ data may not be the best choice. GPCP applies gauge under catch correction to in situ precipitation measurement, which has been found important to improve snowfall measurement (Behrangi et al. 2018). Besides, in section 3.3 historical analysis of the data is done using 1979-2017 precipitation data. For this period GPCP is the best available data.

Following sentences are added in the GPCP section
"Global Precipitation Climatology Project (GPCP) is a widely used global precipitation data. Most of the other observational products don't produce precipitation estimates beyond 60deg S/N for longer historical period (1979 – present). Besides, GPCP applies gauge under catch correction to in situ precipitation measurement, which has been found important to improve snowfall measurement (Behrangi et al., 2018)
Behrangi, A., A. Gardner, J. T. Reager, J. B. Fisher, D. Yang, G. J. Huffman, and R. F. Adler (2018), Using GRACE to Estimate Snowfall Accumulation and Assess Gauge Under catch Corrections in High Latitudes, Journal of Climate, 31(21), 8689-8704, doi: 10.1175/jcli-d-18-0163.1."

5.  line 110 et seq.: It is true that a long-term linear trend is often due to non-climatic processes. But some GRACE trends ARE due to climate–for example, a major drought at the beginning or end of the record. The authors should comment on this possibility at some point in the manuscript, and discuss its implications for results in some regions.

Author's response: Thanks for bringing in, we added a line: "We acknowledge the caveat of a possibility of sudo-trend due to unusual signal at the beginning or end of the record in some regions."

6. line 158 et seq.: "Figure 2" in this passage is actually Figure 3.
Author's response: The maps in Figure 2 demonstrate the strength of the TWA precipitation relationship globally. So, Figure 2 is correct.

7. Section 3.3.2 and other materials on forecasts: I have to admit that I don't understand the emphasis on these hindcasts in the paper. As the authors acknowledge, it's a simple method that doesn't provide very meaningful forecasts. So what is it used for? It seems that the analysis presented in the results section only requires statistics of historical rainfall (mean and standard deviations) that can be compared to observation. The forecasts simply seem to play the role of a not-quite-perfect estimate of climatology. I do understand the authors' point about why forecasts might be useful in the context of predicting the end of drought via forecast of required precipitation. But there is no demonstration of this value in the current paper, as far as I can tell; there's only the claim that it might be valuable.

Author's response: The signal reconstruction and forecast discussed in section 3.3.2 is essential as we used it to create a normal signal first and then used standard deviation to simulate two additional precipitation scenarios of wet and extremely wet conditions. The normal signal is composed of predominantly climatology and long-term trend as the demonstrated model has the least competence in the estimation of inter-annual signals (0-3months). These precipitation scenarios are further needed to demonstrate the possible recovery duration from drought. Nevertheless, it is a very simplistic forecast and we agree with the reviewer that it can be further simplified by using mean and standard deviation. The idea here is to demonstrate that given the three possible scenarios of precipitation, we can estimate the recovery period because by using the GRACE-precipitation relationship we know how much is the required precipitation.

8. line 254: Doesn't blue n this figure indicate good agreement??
Author's response: That's right, thanks. But colors are modified to the sequential lightness-hue ramp as per the other reviewer's comments.

9. line 269 et seq.: It appears that Figure 10 is incorrectly referred to as Figure 8 throughout this passage.
Author's response: That's right, thanks, we corrected it.

10 Section 4.2.2: I assume that Figure 10 here really refers to Figure 11
Author's response: That's right, thanks for pointing out. Figures are modified and their numbers are corrected.

 I recommend an edit for style and grammar. The paper is clear, but there is some awkward phrasing.
Author's response: Edited the manuscript. Many thanks for the supportive comments

**Reviewer 2**

We are highly thankful for the detailed and supportive comments provided by the reviewer. Please find the response below. Modifications in the manuscript are shown within quotation marks.

Summary:

The presented work shows an integrated precipitation approach to determine the recovery period and required precipitation to refill water storages and thus to overcome a hydrological drought. Thus, historical integrated precipitation is linked to total water storage anomalies (TWSA) by GRACE to combine and validate their precipitation- based methodology to an existing storage deficit methodology. Furthermore, three scenarios of precipitation forecast are provided to identify the best estimated time of recovery. They found that the recovery period of integrated precipitation is in good agreement with the recovery period from TWSA, especially in regions where integrated precipitation and total water storage changes showed a strong linear relationship. I think that this work discusses an important topic to have a better understanding of drought evolution and to use this information possibly in water management. The methodology and findings are of good scientific quality and significance, but yet I have general and specific concerns, especially regarding to presentation quality, that are listed below. Thus, I recommend major revision, but believe that the manuscript could be published after addressing/clarifying my comments.

Response: We agree and appreciate the reviewer for guiding the paper in such a detail to improve its clarity and focus.

General comments

1. Until the first results were shown, it was not clear if the precipitation or the GRACE approach is the main contribution of the paper. This is important for abstract, introduction, conclusion and maybe should also be more consistent with the title and structure of the data and methods chapter. For example, [Page1 Line14] says the main goal is the combination of GRACE and precipitation, while [Page1 Line21] let assume that the author's main point is the precipitation approach and GRACE is only used as validation.

Author's response: We thank the reviewer for bringing this up. The paper uses both GRACE and GPCP equally, therefore, the title is modified as 'Estimation of hydrological drought recovery based on precipitation and GRACE water storage deficit'. GRACE is also used for validation but

the main focus of this work is drought recovery estimation, based on the required precipitation, which is estimated from GRACE ad precipitation dataset. Added a line "The intellectual contribution of this paper is in the estimation drought recovery and conceptually bringing a framework for drought recovery forecast based on precipitation deficit. " (line 109)

2. More clarification is needed about the drought definitions. Do you place your approach more in the context of hydrological drought or drought in general? The manuscript should be consistent according to the drought definitions. Be also clear about other drought categories of parameters, e.g.: [Page 1 Line32] meteorological drought is not only described by precipitation, also evapotranspiration. [Page1 Line34] soil moisture, precipitation, and runoff and not all hydrological parameters. For exam- ple, precipitation is a meteorological parameter.

Author's response: A sentence is modified in the introduction to clarify that the study is more in the context of hydrological drought. ' This study focusses on  hydrological drought, which requires, combining both surface (snow and surface water), and subsurface (soil moisture and groundwater) hydrological information.' (line 43)

drought categories of parameters are modified:

"including agricultural (soil moisture deficit), meteorological (eg. precipitation deficit or increase in evapotranspiration), and hydrological (storage deficit for eg. in streamflow/groundwater) droughts." (line 40)

3. Why are mascons used instead of spherical harmonics, the mascon solutions are underlying by constraints. Does the cap size of 3 x 3 degree of mascon solution then not represent a similar spatial resolution as the spherical harmonic GRACE resolution?

Author's response:
We understand the reviewer's concern about the possible discrepancies between different GRACE solutions, as they are produced using different approaches by different centers. Mascon based GRACE solution has less leakage of signals than spherical harmonic solution and do not necessitate empirical filters to remove north-south stripes. Therefore, spatial resolution of 3x3 mascon is a bit different than spherical harmonics. We added a couple of lines in the GRACE section and added a supplementary material comparing different GRACE solution for an example location.  In this study we used JPL based mascon GRACE solution as it has a relatively similar spatial resolution (3°x 3°) as that of GPCP (2.5°x 2.5°). We have also acknowledged that though GRACE mascon and GPCP 2.5 degree is considered as comparable, nevertheless areas of the unit representations are different at different locations. Following lines are added in the section 2.1

"The GRACE level-3 solution is available from different centers are produced using different approaches (spherical harmonics or mascon), different filers, smoothing factors, etc. and eventually, there can be discrepancies between different TWS estimates from GRACE solutions (Jing et al., 2019). The differences between GRACE solutions from different centers are mostly very small at a basin level and lie within the error bounds of the GRACE solution itself

(Sakumura et al., 2014). However, at 0.5-degree grid the difference between the amount of missing water estimated by the different GRACE solutions increases substantially (as seen in the supplementary material). This caveat can be mitigated in future by using ensemble of GARCE products. Nevertheless, they are consistent with the detection of drought duration. Please look into the supplementary material for the comparison between water storage deficit estimated by the different GRACE solutions.

In the conclusion section following lines are added
'However, careful cautions are warranted to interpret the GRACE signal at 0.5 degree grid due to different post processing techniques applied by different GRACE solutions to overcome the inherent limitation in the spatial resolution of GRACE.
Significant ant difference in intensity of drought is observed by different GRACE solutions. Nevertheless, all GRACE solutions have same drought duration.'

Supplementary material

The official GRACE Science Data System continuously released monthly GRACE solution for three different processing centers: GeoforschungsZentrum Potsdam (GFZ), Center for Space Research at University of Texas, Austin (CSR), and Jet Propulsion Laboratory (JPL). These three solutions used different parameters and strategies, such as different degree and order, spherical harmonic coefficient, spatial filter and smoothing factor (Jing et al., 2019). The JPL mascon (JPL-M) and CSR mascon (CSR-M) solutions were provided at 0.5 degrees at https://grace.jpl.nasa.gov/data/get-data/monthly-mass-grids-land/ and http://www2.csr.utexas.edu/grace/RL05_mascons.html respectively. GFZ produces only spherical harmonic solution (GFZ-SH), which is downloaded from http://isdc.gfz-potsdam.de/grace-isdc/.

Figure11 shows water storage deficit and estimated required extra precipitation to overcome the drought based on three different GRACE solutions, using the method described in the article for an example location in India (centered on 77.25°E 15.25°N). Sakumura et al., 2014 demonstrated that at a basin-scale, the differences between them are very less. However, the plot shows that there are discrepancies between the amount of missing water at 0.5degree grid because every center has a different method to downscale GRACE inherent spatial resolution to high-resolution grid. Figure 11a shows the detrended climatological anomaly of the three GRACE solutions and cumulative GPCP precipitation has similar variability. The negative anomaly from climatology is considered as drought and is plotted in figure11b. It shows that the difference between CSR and JPL solutions are relatively less than GFZ solution because the first two are the mascon-based solution and are available at 0.5-degree grid (after scaling), while GFZ solution is spherical harmonics based and is re-gridded to 0.5 degree from 1-degree spatial resolution by simple bilinear interpolation. Based on the linear relationship between cumulative detrended GPCP anomaly and detrended GRACE anomalies, required extra precipitation is estimated (figure 11c). The figure shows that the required precipitation varies based on GRACE solutions. Nevertheless, all three GRACE solutions are consistent with the detection of drought duration.

[Figure]

*Figure 21: Water storage deficit and estimated required precipitation based on the spherical harmonic solution from GFZ and the mascon solutions from CSR and JPL .a. Cumulative detrended precipitation anomaly (cdPA) compared with the detrended storage anomaly (dTWSA). b) The negative residuals from the GRACE climatology considered as drought. c) Surplus required-precipitation is estimated based on the linear relationship between dTWSA and cdPA, to fill the storage deficit (middle panel)).*

Jing, W., Zhang, P. and Zhao, X.: A comparison of different GRACE solutions in terrestrial water storage trend estimation over Tibetan Plateau, Sci Rep, 9, doi:10.1038/s41598-018-38337-1, 2019.

Sakumura, C., Bettadpur, S. and Bruinsma, S.: Ensemble prediction and intercomparison analysis of GRACE time-variable gravity field models, Geophysical Research Letters, 41(5), 1389–1397, doi:10.1002/2013GL058632, 2014.
.

4.  [Page3 Line103] Which method is used to regrid the data? Is there a precipitation data set with an 0.5 degree resolution? I ask myself if the downscaling from 2.5 to 0.5 degree has a significant impact.

Author's response:  GPCP is a widely used global precipitation data. In section 3.3 historical analysis 1979-2017 of the precipitation data is done. For this period GPCP 2.5degree is the best available data which is interpolated to 0.5 degree by using the bilinear method to harmonize it with the GRACE grid. There are many higher resolution precipitation products like TRMM, CRU, GPCC, etc. However GPCP combines the strength offered by in situ as well as satellite data to obtain global picture while others are limited to 60 degrees North and South latitudes. Additionally, GPCP applies gauge under catch correction to in situ precipitation measurement, which has been found important to improve snowfall measurement (Behrangi et al. 2018). Besides, in section 3.3 historical analysis of the data is done using 1979-2017 precipitation data. For this period GPCP is the best available data. We also acknowledge the caveat of different re-gridding method in GRACE and GPCP.

Following sentences added in the GPCP section:

"Global Precipitation Climatology Project (GPCP) is a widely used global precipitation data. Most of the other observational products don't produce precipitation estimates beyond 60deg S/N for longer historical period (1979 – present). Besides, GPCP applies gauge under catch correction to in situ precipitation measurement, which has been found important to improve snowfall measurement. (Behrangi et al., 2018)

Nevertheless, areas of the unit representations are different in tens of thousands $km^2$ at different locations which get worst towards pole. We also acknowledge the possible caveat due to different methods of re-gridding of both the datasets, which can be improved in future work. However, as drought is a smooth process the impact of neighboring pixels should not affect the analysis significantly.

Behrangi, A., A. Gardner, J. T. Reager, J. B. Fisher, D. Yang, G. J. Huffman, and R. F. Adler (2018), Using GRACE to Estimate Snowfall Accumulation and Assess Gauge Under catch Corrections in High Latitudes, Journal of Climate, 31(21), 8689-8704, doi: 10.1175/jcli-d-18-0163.1.

5.  [Page3 Line110] Why are the TWSA smoothed with an averaging filter? Does their noise have a significant impact on the results?

Author's response: As drought develops in a smooth progression and we are looking for the amount of missing mass in a system caused by drought. Therefore, 3months moving average is

considered as a better representation of the progression of drought. Monthly observations also have similar relationship between TWS and precipitation but signals are neat and better interpretable after averaging filter.

6. [Page4 Line129-136] The linkage between integrated precipitation and GRACE is an important aspect for the validation so it should be explained more detailed. The paragraph is (probably) based on the water balance equation, which should at least be mentioned but better also shown. The assumptions that were decided to describe the relationship about evapotranspiration/runoff should be added here and it also should get clear how the precipitation is integrated in time. So for example, is it integrated continuously for each month to the previous months or is there an integration period of 3 months that is running over all months etc.?

Author's response: We understand the reviewer's point and added the following lines in section 3.2:

$$dS/dt = P - ET - R \qquad \text{Eq. 1}$$

"

The water balance equation based on hydrological fluxes ( Eq. 1) shows that the change in terrestrial water storage (dS) in a region for a given month (dt) depends on the monthly precipitation (P, mm/month); evapotranspiration (ET, mm/month) and the streamflow (R, which includes both surface water and subsurface water) (Swenson and Wahr, 2006). Assuming the relationship between precipitation and ET + R remains constant for a region, the variability in precipitation gives an idea of possible variation in the storage

Swenson, S. and Wahr, J.: Estimating Large-Scale Precipitation Minus Evapotranspi- ration from GRACE Satellite Gravity Measurements, J. Hydrometeor., 7(2), 252–270, doi:10.1175/JHM478.1, 2006. " (line181)

7. [Page4 Lines144-147 and Lines158-162] It was not clear how the required precipita- tion is linked to the regression coefficients. It would great if the linkage for the example of a coefficient lower/higher/equal 1 in the first paragraph is clearly explained. Sec- ondly, how do we then get the surplus required-precipitation? Is it derived by removing cdPA from dTWSA?

Author's response: It is a great idea; we added a small description in section 3.2:

"Based on the linear relationship between dTWSA and cdPA the required precipitation has been estimated. Regression coefficients greater than 1 means the required precipitation is more than the amount of missing water. This is because precipitation lost in other hydrological processes like evapotranspiration, runoff ( Eq.1) is not observed by storage variability).  Coefficient equals to 1 means the amount of required precipitation is the same as that storage loss, which means there is no other dominant process in the region. Coefficient less than 1 are the regions of weak

precipitation-storage coupling, which can be due to other physical processes like melting of snow/frozen surfaces, groundwater extraction, irrigation, etc (non-red regions in Figure 2a)." (line 210)

Figure 4, as well as some other figures, is analyzed too shortly (e.g. [Page5 Line181]) or, for example, only part a) of a), and b) is described. The figures provide much more information, especially about spatial differences. So, the figures should be described more in detail, which I prefer because they contain interesting findings, or removed/added to supplementary.

Author's response: We agree with the reviewer's point and added small description of the figure.

"Figure 4 shows the fractional variance of the decomposed signal. For most regions, annual signal dominate in precipitation (Figure 4a). However, regions where the wet season is not explicit in their climatology, high-frequency signal plays a major role, for example in central Europe, eastern Siberia, western N. America, southern Australia, etc. (Figure 4c). Contrarily, the long-term signal obtained by combining linear trend and the inter-annual signal has the least variability globally (Figure 4b). These smooth signals are driven by climate indices like El Niño southern oscillation (ENSO), Pacific decadal oscillation (PDO), and the North Pacific mode (NPM), etc. (Özger et al., 2009). The annual and long-term signals are directly applied for the signal reconstruction with the assumption that a similar trend will continue. "

Özger, M., Mishra, A. K. and Singh, V. P.: Low-frequency drought variability associated with climate indices, Journal of Hydrology, 364(1), 152–162, doi:10.1016/j.jhydrol.2008.10.018, 2009" (line 260)

8. [Page5 Line188] It is not clear how the sub-seasonal signal is computed and where the number of 0 to 3 months of reconstruction is resulting from. The final hindcast is 2 years, so how did the authors manage the 0-3 months restriction of the sub-seasonal signal?

Author's response: A sentence is added in section 3.3.2

"The sub-seasonal signal is obtained from the residual of the inter-annual signal. This high frequency signal has 0-3 months of temporal autocorrelation; accordingly, we have limited skill in synthesizing sub-seasonal signal." (line 295)

9. [Page7 Line247] The definition of severe drought was not exactly set. What is the definition or to which definition is it referred?

Author's response: A line added

"Here, the severity of a drought defined by the amount of water shortage in a month." (line 348)

[Page7 Line253] Based on which principles are the differences of recovery months divided into the different classes? How were the classes determined? It leads also to confusion in Figure 9. Without reading the caption it seems as if the difference is very small everywhere (from 1 to 4

months), but the number does not represent the "difference in months", rather the "class number of differences in months".

Author's response: Label of the figure is modified (thanks for pointing). The first two classes are defined by 2 months difference, as majority of regions have less difference than the third class has 4 months difference and last class is has no upper limit.

10. [Page9 Line333] Could you please discuss that the recovery period derived from precipitation is also underlying certain assumptions (e.g. about evapotranspiration)?

Author's response: The underlying assumption of this work is that the relationship between precipitation, runoff and evaporation for each location will remain unchanged. As the required precipitation is derived from the GRACE observations, it inherits the relationship between P and ET based on equation 1. Therefore, the estimated required precipitation includes the impact of evaporation and runoff loss.

Specific comments

I would recommend to work through the manuscript again to remove grammatical/syntactic errors. Some examples: - [Page1 Line30] Missing commas, 'the', and 'and/or' (should also be checked: and/or is needed before last item of a list), suggestion: '. . . developing parts of the world, for example, the 2011 East Africa drought or the 2018 dry corridors of central America (REF).'

Author's response: Modified and references added

"example the 2011 East African drought (Lyon and DeWitt, 2012) or the 2014-16 dry corridors of central America (Guevara-Murua et al., 2018) ."

- [Page2 Line56] have/has and "the" too much, suggestion: '. . . is independent of other drought indices and has global spatial coverage.'

Author's response: Modified the sentence. "The GRACE-based drought index is independent of the meteorological estimates and their combined uncertainties."

- [Page2 Line69] singular/plural, citing brackets, suggestion: '. . . reviewed different kinds of drought and their prediction methods based on statistical, dynamical, and hybrid methods. Panet et al. (2013) were ...' –

Author's response: Corrected the citing bracket and singular/plural

[Page3 Line91] add date of last access for websites –

Author's response: Added the access date

[Page4 Line146] be consistent with required precipitation/required-precipitation – [Page 5 Line 181] be consistent with figure/Figure and section/Section - [Page5 Line190] estimated precipitation → reconstructed precipitation - [Page5 Line202] be consistent with climatology/annual signal

Author's response: Changed to a consistent expression

References that should be added: - [Page2 Line59] Reference for global gridded assessments –

Author's response: Reference added (Gerdener et al., 2020; Li et al., 2019)

[Page2 Line62] Reference for increasing frequency of drought –

Author's response:  Reference added (Cook et al., 2014)

[Page3 Line98] Reference for cubic convolution interpolation

Author's response: Reference added (Keys, 1981)

[Page2 Line77] Please explain why only terrestrial water storage can be used instead of, for example, in-situ groundwater data.

Author's response: A line added.

"With the sparse availability of in-situ groundwater observations and limited soil moisture observations upto top 5cm of the soil, complete profile of the water stored in a column can only be obtained from the GRACE-based terrestrial water storage." (line 105)

[Page2 Line81] It could be added that you focus on sub-decadal drought because there are only about 15 years of GRACE data.

Author's response: Modified the line as below:

"Here, we focus on sub-decadal drought only because of the availability of GRACE data for 15 years. The study can be extended for a longer time frame with the GRACE- follow on observations. "(line 117)

[Page2 Line83] GPCP was not introduced yet.

Author's response: corrected as Global Precipitation Climatology Project (GPCP)

[Page3 Line114] "Here, we define 'recovery' as a return to the climatological storage state for a given month." This is not totally clear to me, does it mean that the deviation from current

dTWSA to the climatology itself in a specific month, which is referred to as severity in Thomas et al. (2014), is already the recovery?

Author's response: Yes, decrease in severity is recovery in a particular month.

[Page3 Line123] state of drought → severity of drought?

Author's response: severity of drought changed to intensity of drought

[Page4 Line125] Could you mark the three recovery periods in Figure 1, please? It seems as if the recovery periods are longer than 1.5, 1 and 0.5 years.

Author's response: Thanks for pointed out. Yes, each grid is two years so it is almost 4, 2 and 1 years.

[Page5 Line167] ... are statistically analyzed using the methods of . . .

Author's response: Added "using signal decomposition "

[Page5 Line187] How was the number 10-14 months for autoregression chosen?

Author's response: Based on the duration of significant auto-correlation with inter-annual signal.

[Page5 Line184] The annual signal and linear trend extracted by signal decomposition [Page5 Line200] worst → worse. [Page5 Line201], [Page7 Line271], and [Page7 Line283] etc.: 'In these regions...', 'this region', and 'monsoon regions' be precise which regions. [Page5 Line202] robust → dominant

Author's response: corrected

[Page6 Line211] Where (reference) is it defined that one sigma represents a wet year

and three sigma an exceptionally wet year?

Author's response: These conditions are assumed in to generate three scenarios, the sentence is modified accordingly.

"one standard deviation wetter than normal precipitation is assumed as wet month and three standard deviations wetter than normal precipitation is assumed as exceptionally wet month."

[Page6 Line220] providing a minimum and maximum baseline?

Author's response: Even in an exceptionally wet scenarios in the dry season, the system fails to recover. Therefore, it does not provide a maximum baseline.

[Page6 Line232] "In Figure7, observed precipitation (red dashed line) and absolute required precipitation (blue line) ..." This was already said.

Author's response: Deleted

Figure 7: This was quite hard to analyze. I would recommend to enlarge the subfigures or put them in a different order (e.g. 4 x 1).

Author's response: Modified

[Page6 Line241] some drought → drought

Author's response: Deleted 'some'

[Page6 Line241] Remove 'it is a random selection of the month for'

Author's response: Removed

[Page7 Line254] blue → red?

Author's response: Corrected the color

[Page7 Line256] Is with 80% the total global land area or the masked global land area meant?

Author's response: Masked global area.

4.2.2 Different precipitation scenario → Precipitation scenarios
[Page7 Line 265] 'We stimulated one-month (February 2016) recovery period ...' Not

Author's response: clear what is meant

This section shows the recovery percentage within a month based on the three precipitation scenarios.

[Page8 Line288] Better more precise: Here we define drought severity and duration using ...

Author's response: Added 'drought intensity and duration'

5 Discussion: Refer to section if different aspects/findings are discussed. [Page8 Line298] soil water column → water column

Author's response: Deleted 'soil'

[Page8 Line 299] Position of sentence in paragraph awkward in the previous context.

Author's response: Deleted the sentence

[Page9 Line327] Also shown in Figure 11 . . .

Author's response: Added (as shown in figure 8)

[Page9 Line342] 1) the independence from other drought indices → more precise, which independencies?

Author's response: Added names of indices (PDSI, SPEI, SPI). Thanks

All Figures: Please check figure references in the text, some of the references have been mixed up. Make sure that all figure captions and title really describe what is shown (compared to what) e.g. Figure 4 fraction of a), b), and c) to what? Total of all. . . or Figure 9 validation of what by what? And consider changing colorbars, since some figure might better be represented in a different way, e.g. Figure 9 discrete colorbar.

Author's response: Modified most of the figures, please see the attachment. Many thanks for the very detailed review and constructive comments.

**Reviewer 3:**

We appreciate the constructive comment. We went over the paper and tried to improve it by adding more clarification and improving the figures. Modifications in the manuscript are shown in quotation marks

The authors devise a novel method for estimating intradecadal drought recovery peri- ods using GRACE and precipitation data globally. The total water storage estimates from GRACE are used to determine the deficit and the precipitation data is used for estimating the drought recovery periods using an empirical forecasting model. The is- sue is an important one in the context of ongoing climate change. Furthermore, the subject matter is also relevant for the journal and its audience. Having said that there are methodological issues in the data analysis which I will point out in the subsequent section, and the manuscript requires improvement in its narrative.

1.The title does not fully reflect the content of the manuscript. Firstly, the work only looks at short-term (intradecadal) droughts and secondly it uses precipitation in addition to GRACE to estimate the drought recovery times. These two aspects of the manuscript should be reflected in the title. Currently, going by the title, the drought recovery time is solely estimated from GRACE, which is incorrect.

Author's response: Thanks for bringing this up, we have modified the title as follows:

"Estimation of hydrological drought recovery based on precipitation and GRACE water storage deficit"

2.The central goal of the manuscript seems to be to determine drought recovery times and that is facilitated by precipitation forecasts, and the majority of the manuscript is dedicated to figuring out an empirical way to predict precipitation. However, in the conclusions there is hardly any mention of precipitation and the empirical forecast model, and their role in drought recovery times. Rather it is concluded that the one of the findings is that GRACE can be used to derive drought indices, which appears to have been established by Thomas et al (2014).

Author's response: We understand the reviewer's concern. We mentioned in the manuscript that the precipitation forecast is not the focus of this work, so we preferred not to discuss it. The main idea of precipitation prediction is to generate 3 scenarios and it is mentioned. Section 3.3 states that 'Note that the motivation for providing a precipitation forecast here is not to present a state-of-the-art precipitation prediction, but to demonstrate the potential utility of the terrestrial water storage deficit in determining required-precipitation and estimating a likely time to recovery. This methodology could be augmented with any type of more complex precipitation forecasting approaches.'

I agree, Thomas et al (2014) has already established that GRACE can be used to derive drought indices. However, the conclusion emphasis its importance in the estimation of required precipitation, as following: 'GRACE based drought index is valid to estimate the required-precipitation for drought recovery.'

3. Throughout the manuscript it is not clear as to what type of drought the authors are trying to quantify. In the title it is indicated that the authors are concerned about hydrological droughts, but nothing much is said in the manuscript. In the introduction they specify there are multiple definitions of droughts, but beyond that there is no indication on what sort of droughts the authors are interested in and which sorts will be sensitive to the method developed in the manuscript. It would be beneficial if the authors clarify this for the readers.

Author's response: Thanks for bringing this up, we modified a sentence in the introduction. 'This study focusses on hydrological drought, which requires, combining both surface (snow and surface water), and subsurface (soil moisture and groundwater) hydrological information. '

4. For the data the authors use GRACE JPL mascons for total water storage and GPCP for precipitation. Given the wide variety of data available both for total water storage (CSR mascons, GSFC mascons, CSR, GFZ, JPL, ITSG spher- ical harmonics, COST-G combined solutions) as well as precipitation (GPCC, CRU, Delaware), it would be interesting to know how different the drought recov- ery times would be if we were to choose a different pair of datasets. At least in the case of GRACE it should be tested, because it is the starting point for the method proposed in the manuscript. Given the lack of consensus on which GRACE flavour is to be used, or how to reconcile the data, it is worthwhile to perform this test.

Author's response: We understand the reviewer's concern about the possible discrepancies between different GRACE solutions, as they are produced using different approaches by different centers. We added a couple of lines in the GRACE section and added a supplementary material comparing different GRACE solution for an example location.  In this study we used JPL based mascon GRACE solution as it has a relatively similar spatial resolution (3°x 3°) as that of GPCP (2.5°x 2.5°). We have also acknowledged that though GRACE mascon and GPCP 2.5 degree is considered as comparable, nevertheless areas of the unit representations are different at different locations. Following lines are added in the section 2.1

"The GRACE level-3 solution is available from different centers are produced using different approaches (spherical harmonics or mascon), different filers, smoothing factors, etc. and eventually, there can be discrepancies between different TWS estimates from GRACE solutions (Jing et al., 2019). The differences between GRACE solutions from different centers are mostly very small at a basin level and lie within the error bounds of the GRACE solution itself (Sakumura et al., 2014). However, at 0.5-degree grid the difference between the amount of missing water estimated by the different GRACE solutions increases substantially (as seen in the supplementary material). This caveat can be mitigated in future by using ensemble of GARCE products. Nevertheless, they are consistent with the detection of drought duration. Please look into the supplementary material for the comparison between water storage deficit estimated by the different GRACE solutions.

In the conclusion section following lines are added

'However, careful cautions are warranted to interpret the GRACE signal at 0.5 degree grid due to different post processing techniques applied by different GRACE solutions to overcome the inherent limitation in the spatial resolution of GRACE.
Significant ant difference in intensity of drought is observed by different GRACE solutions. Nevertheless, all GRACE solutions have same drought duration.'

Supplementary material

The official GRACE Science Data System continuously released monthly GRACE solution for three different processing centers: GeoforschungsZentrum Potsdam (GFZ), Center for Space Research at University of Texas, Austin (CSR), and Jet Propulsion Laboratory (JPL). These three solutions used different parameters and strategies, such as different degree and order, spherical harmonic coefficient, spatial filter and smoothing factor (Jing et al., 2019). The JPL mascon (JPL-M) and CSR mascon (CSR-M) solutions were provided at 0.5 degrees at https://grace.jpl.nasa.gov/data/get-data/monthly-mass-grids-land/ and http://www2.csr.utexas.edu/grace/RL05_mascons.html respectively. GFZ produces only spherical harmonic solution (GFZ-SH), which is downloaded from http://isdc.gfz-potsdam.de/grace-isdc/.

Figure11 shows water storage deficit and estimated required extra precipitation to overcome the drought based on three different GRACE solutions, using the method described in the article for an example location in India (centered on 77.25°E 15.25°N). Sakumura et al., 2014 demonstrated that at a basin-scale, the differences between them are very less. However, the plot shows that there are discrepancies between the amount of missing water at 0.5degree grid because every center has a different method to downscale GRACE inherent spatial resolution to high-resolution grid. Figure 11a shows the detrended climatological anomaly of the three GRACE solutions and cumulative GPCP precipitation has similar variability. The negative anomaly from climatology is considered as drought and is plotted in figure11b. It shows that the difference between CSR and JPL solutions are relatively less than GFZ solution because the first two are the mascon-based solution and are available at 0.5-degree grid (after scaling), while GFZ solution is spherical harmonics based and is re-gridded to 0.5 degree from 1-degree spatial resolution by simple bilinear interpolation. Based on the linear relationship between cumulative detrended GPCP anomaly and detrended GRACE anomalies, required extra precipitation is estimated (figure 11c). The figure shows that the required precipitation varies based on GRACE solutions. Nevertheless, all three GRACE solutions are consistent with the detection of drought duration.

a.

[Figure]

*Figure 31: Water storage deficit and estimated required precipitation based on the spherical harmonic solution from GFZ and the mascon solutions from CSR and JPL .a. Cumulative detrended precipitation anomaly (cdPA) compared with the detrended storage anomaly (dTWSA). b) The negative residuals from the GRACE climatology considered as drought. c) Surplus required-precipitation is estimated based on the linear relationship between dTWSA and cdPA, to fill the storage deficit (middle panel)).*

Jing, W., Zhang, P. and Zhao, X.: A comparison of different GRACE solutions in terrestrial water storage trend estimation over Tibetan Plateau, Sci Rep, 9, doi:10.1038/s41598-018-38337-1, 2019.

Sakumura, C., Bettadpur, S. and Bruinsma, S.: Ensemble prediction and intercomparison analysis of GRACE time-variable gravity field models, Geophysical Research Letters, 41(5), 1389–1397, doi:10.1002/2013GL058632, 2014.

Yes, we agree, there are many precipitation products like CRU, GPCC, etc. However, GPCP is a widely used global precipitation data. GPCP combines the strength offered by in situ as well as satellite data. In many regions of the world in situ data are sparse, so using a product that only utilizes in situ data may not be the best choice. GPCP applies gauge under catch correction to in situ precipitation measurement, which has been found important to improve snowfall measurement (Behrangi et al. 2018). Besides, in section 3.3 historical analysis of the data is done using 1979-2017 precipitation data. For this period GPCP is the best available data.

Following sentences are added in the GPCP section
"Global Precipitation Climatology Project (GPCP) is a widely used global precipitation data. Most of the other observational products don't produce precipitation estimates beyond 60deg S/N for longer historical period (1979 – present). Besides, GPCP applies gauge under catch correction to in situ precipitation measurement, which has been found important to improve snowfall measurement (Behrangi et al., 2018)
Behrangi, A., A. Gardner, J. T. Reager, J. B. Fisher, D. Yang, G. J. Huffman, and R. F. Adler (2018), Using GRACE to Estimate Snowfall Accumulation and Assess Gauge Under catch Corrections in High Latitudes, Journal of Climate, 31(21), 8689-8704, doi: 10.1175/jcli-d-18-0163.1."

5. The GRACE and the GPCP datasets are represented on $3°$ spherical cap and $2.5° \times 2.5°$ equi-angular grid. After indicating that the area of the unit representations are comparable, they represent the two datasets on a $0.5° \times 0.5°$ grid to perform the analyses. There a couple of issues here. Firstly, the difference between the areas of the unit representations are at best $\approx [10, 000]\text{km}^2$ (at the equator) and at worst $\approx [80, 000]\text{km}^2$ (close to the poles). Secondly, by regridding them to a smaller grid size, they are only making map a bit smooth, but there is no change in the information content. The best way to bring them to a commensu- rate resolution to perform the data analysis would have been to filter them with a common filter either a Gaussian or any other contrast preserving filter, and then regrid them to any other grid size they wanted. It is essential that the authors discuss the impact of these data processing choices on the final results.

Author's response: Thanks for bringing it so precisely. The mascon solution in the study is re-gridded by multiplying it with a scaling factor, to improve the interpretation of signals at sub-mascon resolution. This is essential as the shape and size of mascon changes with latitude. We agree that there are significant differences between the mascon (3x3 grid) and GPCP (2.5) area at different locations. The Following sentence is added in the GPCP section, thanks for the comment with numbers.

"Nevertheless, areas of the unit representations are different in tens of thousands km$^2$ at different locations which get worst towards pole. We also acknowledge the possible caveat due to different methods of re-gridding of both the datasets, which can be improved in future work. However, as drought is a smooth process the impact of neighboring pixels should not affect the analysis significantly. "

Based on these comments I recommend a major revision.

**3 Technical comments**

30 Please provide references for the events you have described

Author's response: Reference added

example the 2011 East African drought (Lyon and DeWitt, 2012) or the 2014-16 dry corridors of central America (Guevara-Murua et al., 2018)

32 Please provide standard references for the drought definitions, for e.g., Wilhite and Glanz (1985). Water International

Author's response: Added (Wilhite and Glantz, 1985)

33 It is not clear what you want to convey by indicating the different indices.

Author's response: In this study GRACE TWS is used as a drought index, therefore it is essential to describe some other common drought indices and their limitations. The paragraph has been restructured to make it clearer.

38 Similar is the case for remote sensing data based drought indicators. Please clarify to the reader what their benefits and shortcomings are in order to get a perspective.

Author's response: Thanks, we added the following line in the introduction

"With the sparse availability of in-situ groundwater observations and limited soil moisture observations (up to top 5cm of the soil), a complete profile of the water stored in a column can only be obtained from the GRACE-based terrestrial water storage."

51 "This method can improve ..." until end of line 55. Please corroborate the state- ment, if it is not a conclusion of Thomas et al (2014).

Author's response: The lines are moved to a paragraph below to separate it from Thomas et al. paper discussion and a line added to it.

… This quantification of total required storage for drought recovery can only be estimated using GRACE observation.

59 "... are still a few" Please cite some of those studies

Author's response: Reference added (Gerdener et al., 2020; Li et al., 2019)

63 successive –> next

Author's response:  Changed

74 "However, above average ..." until end of line 77. Please clarify whether it is your opinion or a conclusion of Pan et al (2013)

Author's response: Following line is added to separate it from Pan et al paper.

Pan et-al., approach is exclusively precipitation based, however,  …

84 In general, the introduction lacks a cogent narrative. It is hard to identify what issue you are trying to address

Author's response: Added a line "The intellectual contribution of this paper is in the estimation drought recovery and conceptually bringing a framework for drought recovery forecast based on precipitation deficit. "

88 "... global and regional water cycle." Please provide a reference for the same.

Author's response: Added global (Eicker et al., 2016; Fasullo et al., 2016) and regional water cycle (Singh et al., 2018; Springer et al., 2017)

104 When you say comparable, please indicate the numbers.

Author's response: we discussed that 3 degree Mascon and 2.5 degree GPCP data products are comparable in spatial terms. Additionally, as per your suggestion area details are added in the GPCP section.

135 Please clarify to the reader why you need to integrate the precipitation time- series.

Modified the line as follows:

The smoothed and detrended precipitation anomaly is then integrated in time to get storage anomaly, which is termed as cumulative detrended smoothed precipitation anomaly (cdPA).

142 The variability of precipitation intensity can be checked. It is unclear why this needs to be assumed.

Author's response: This assumption is for the estimation of required precipitation to consider the relationship between precipitation and storage variability stable. For example, a region having mostly slow rain has one kind of storage-precipitation relationship and if it gets unusual heavy

rain then the relationship changes. Therefore, we assume here, that there is less variability in the precipitation intensity of a region.

189 The paragraph reads like the caption of Figure 5. Please interpret the figure for the reader as to what you want to convey through that figure.

Author's response: Thanks, a couple of sentences added.

"This precipitation reconstruction skill is used for a simplistic normal forecast. Further, two additional precipitation scenarios are simulated by adding respectively one and two standard deviations of precipitation to the normal forecast, which is used in probability recovery analysis."

199 Is the NSE performed on the full signals or after removing the climatology sig- nal? It is well known that the climatology will dominate the metric if it is retained. Please clarify.

Author's response: Many thanks for the correction, NSE section is removed.

204 In Figure 6, please indicate the regions of weak association. Also, instead of a continuous scale, it would be better to use a discrete scale colorbar, i.e., one colour for a range of values. It is more convenient for the human eye to interpret such images.

Author's response: Regions dominated by sub-seasonal signal has a weak association. Modified the figure

265 stimulated –> simulated?

Author's response: Corrected

299 "hydrological compartments" – Do you mean storage compartments?

Author's response: Modified: hydrological storage compartments

342 "independency from other drought indices" – Do you mean to say that SPI de- pends on other drought indices? Please clarify the "independence" argument.

Author's response: Modified: The GRACE-based drought index is independent of the meteorological estimates and their combined uncertainties

343 "spatial coverage" – Indices based on NDVI also cover much of the globe. How is this an advantage specific to the GRACE method?

Author's response: Modified: The GRACE-based drought index is independent of the meteorological estimates and their combined uncertainties

Apart from the specific comments, I would like to indicate that it was rather frustrating to read such a methodology-heavy manuscript devoid of any equations. Even if the equations involved are simple and straight-forward I believe they will provide clarity for the reader. Please consider incorporating equations.

Author's response: Equation and it description added

$$dS/dt = P - ET - R \qquad \text{Eq. 1}$$

The water balance equation based on hydrological fluxes ( Eq. 1) shows that the change in terrestrial water storage (dS) in a region for a given month (dt) depends on  is the monthly precipitation (P, mm/month); evapotranspiration (ET, mm/month) and streamflow (R, which includes both surface water and subsurface water) (Swenson and Wahr, 2006). Assuming the relationship between precipitation and ET + R remains constant for a region, the variability in precipitation gives an idea of possible variation in the storage.

Swenson, S. and Wahr, J.: Estimating Large-Scale Precipitation Minus Evapotranspi- ration from GRACE Satellite Gravity Measurements, J. Hydrometeor., 7(2), 252–270, doi:10.1175/JHM478.1, 2006.

Your results largely fall into the sequential and diverging types of data for which col- orbrewer2.org provides very good advice on choosing colorbars. Typically, sequential data require only one colour with varying intensity to indicate the sequences and di- verging data requires two colours of varying intensities. Furthermore, the standard colorbars are not color-blind friendly. I strongly recommend that you follow the rules indicated in the website to improve the graphics in the manuscript.

Author's response: All of the maps are modified with new color bars, please check the attachment.

a. Correlation coefficients

b. Regression Coefficients

[revised manuscript text omitted]

---

## Author Response (AR2)

Rebuttal

1. The mascons are estimated as 3-degree spherical caps, where 3-degree indicates the radius of the spherical cap. The 3 degree spherical cap mascon estimates are then represented on a 0.5 degree x 0.5 degree grid. Please correct the text in the manuscript accordingly.

Ans. Modification made in the manuscript at line 120. Thanks for the suggestion.

2. The supplementary material requires a proper motivation and an objective. Currently, the authors show that the involved datasets lack consistency between them in terms of amplitude. However, they point out that the phase of the datasets match each other, and therefore, they argue that it is not required to bring the datasets to a consistent resolution, which can alleviate the issue. While the argumentation can be accepted, it still requires to clarify whether the result depicted in the supplementary material is not a chance occurrence. I suggest the authors to add similar plots for a few more locations, maybe for different climatic zones, to see if the phase coherence between the datasets is consistent globally. Furthermore, I urge the authors to clearly specify their goal for the supplementary material.

Ans. Thanks for the advice. The supplementary material is modified by adding a small introduction and additional locations to demonstrate the global consistency of the method.